

# A time-varying parameter estimation approach using split-sample calibration based on dynamic programming

**Xiaojing Zhang[a,b], Pan Liu[a,b*]**

[a]State Key Laboratory of Water Resources and Hydropower Engineering Science, Wuhan University, Wuhan 430072, China

[b]Hubei Provincial Key Lab of Water System Science for Sponge City Construction, Wuhan University

*Corresponding author. Email: liupan@whu.edu.cn;

Tel: +86-27-68775788; Fax: +86-27-68773568





**Abstract:** Although the parameters of hydrological models are usually regarded as
constant, temporal variations can occur in a changing environment. Thus, effectively
estimating time-varying parameters becomes a significant challenge. Following a
survey of existing estimation methodologies, this paper describes a new method that
combines (1) the basic concept of split-sample calibration (SSC), whereby parameters
are assumed to be stable for one sub-period, and (2) the parameter continuity
assumption, i.e., the differences between parameters in consecutive time steps are small.
Dynamic programming is then used to determine the optimal parameter trajectory by
considering two objective functions: maximization of simulation accuracy and
maximization of parameter continuity. The efficiency of the proposed method is
evaluated by two synthetic experiments, one with a simple two-parameter monthly
model and the second using a more complex 15-parameter daily model. The results
show that the proposed method is superior to SSC alone, and outperforms the ensemble
Kalman filter if the proper sub-period length is used. An application to the Wuding
River basin indicates that the soil water capacity parameter varies before and after 1972,
which can be interpreted according to land use and land cover changes. Further
application to the Xun River basin shows that parameters are generally stationary on an
annual scale, but exhibit significant changes over seasonal scales. These results
demonstrate that the proposed method is an effective tool for identifying time-varying
parameters in a changing environment.
**Keywords:** hydrological model; time-varying parameter; calibration; dynamic
programming



## 1. Introduction

Conceptual models describe the physical processes that occur in the real world by means of certain assumptions and empirically determined functions (Toth and Brath, 2007). In spite of their simplicity, conceptual models are effective in providing reliable runoff predictions for widespread applications (Quoc Quan et al., 2018; Refsgaard and Knudsen, 1996), such as real-time flood forecasting, climate change impact assessments (Dakhlaoui et al., 2017), and water resources management. Conceptual hydrological models typically have several inputs, a moderate number of parameters, state variables, and outputs. Among these, the parameters play an important role in accurate simulation and should be related to the catchment properties. However, parameter values often cannot be obtained by field measurements (Merz et al., 2011). An alternative approach is to calibrate parameters based on historical data.

Parameters are usually regarded as constants, because of the general idea that catchment conditions are stable. Constant parameters become inaccurate in differential split sample test (DSST) conditions (Klemes, 1986). For example, parameters calibrated based on data from a wet (or dry) period may fail to simulate runoff in a dry (or wet) period for the same catchment. Broderick et al. (2016) used DSST to assess the transferability of six conceptual models under contrasting climate conditions. They found that performance declines most when models are calibrated during wet periods but validated in dry ones. Fowler et al. (2016) pointed out that the parameter set obtained by mathematical optimization based on one climate condition may not be robust when applied in different conditions. Additionally, the catchment properties can

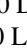



change over time, such as in the case of afforestation and deforestation (Guzha et al.,
2018; Siriwardena et al., 2006). These changes need to be taken into account through
model parameters (Bronstert, 2004; Hundecha and Bardossy, 2004). Hence, temporal
variations in parameters should reflect the changing environment.

One challenge here is the methodology used to identify time-varying parameters.

In the literature, three approaches have been discussed. The first is split-sample
calibration (SSC), whereby available data are split into a moderate number of sub-
periods and the parameters are calibrated individually for each period (Thirel et al.,
2015). The second method is data assimilation (Deng et al., 2016; Pathiraja et al., 2018).
This method assimilates observational data to enable errors, states, and parameters to
be updated (Li et al., 2013), making it possible to identify time-varying parameters. The
third approach is to construct a functional form or empirical equation according to the
correlation between parameters and some climatic variates such as precipitation and
potential evapotranspiration (Deng et al., 2019; Jeremiah et al., 2013; Westra et al.,
2014). Note that this study focuses on methods to identify time-varying parameters
rather than modelling them; hence, only comparisons between SSC and data
assimilation are discussed.

SSC is the most commonly used method (Coron et al., 2012; Fowler et al., 2018;

Paik et al., 2005; Xie et al., 2018). Merz et al. (2011) investigated the time stability of
parameters by estimating six parameter sets based on six consecutive five-year periods.
Lan et al. (2018) clustered calibration data into 24 sub-annual periods to detect the
seasonal hydrological dynamic behavior. Despite broad application, it remains



debatable whether a particular mathematical optimum gives the parameter value during
one period. Many equivalent optima can exist simultaneously for one dataset when
calibrating the model against observations (Poulin et al., 2011). Several studies
addressed this question by adding more constraints to the objective function over the
respective period. For example, Gharari et al. (2013) emphasized consistent
performance in different climatic conditions, while Xie et al. (2018) modified SSC by
selecting parameters with good simulation ability for both the current sub-period and
the whole period. However, few reports have considered the continuity of parameters
in the SSC method.

Continuity requires differences between the parameters in consecutive time steps

to be small, because changes in the watershed characteristics occur over a prolonged
period. This assumption is the basic idea behind data assimilation methods. For
example, the a priori parameters in ensemble Kalman filter (EnKF) methods are
commonly derived from updated values from the previous time step (Moradkhani et al.,
2005; Xiong et al., 2019). From this, a trade-off between simulation accuracy and
parameter continuity is established, and parameters that enable greater continuity are
more likely to be selected. Deng et al. (2016) validated the ability of the EnKF to
identify changes in two-parameter monthly water balance (TMWB) model parameters.
Pathiraja et al. (2016) proposed two-parameter evolution models for improving
conventional dual EnKF, and obtained superior results for diagnosing the non-
stationarity in a system. EnKF and its variants are relatively advanced approaches for
identifying time-varying parameters (Lu et al., 2013). However, for a hydrological



model, the states may change over every time step, whereas the parameters may not, in
particular for hourly time scales. This can be offset by SSC, which assumes that the
parameters retain stable for a pre-determined period (such as decades, years, or months).
Compared to EnKF, the simplicity of SSC is another advantage, as it has a less complex
mechanism and reduced redundancy (Chen and Zhang, 2006).
The aim of this study is to present a new method for time-varying parameter
estimation by combining the strengths of the basic concept of SSC and the continuity
assumption of data assimilation, which is a useful tool for diagnosing the non-
stationarity caused by a changing environment. Compared with data assimilation, the
proposed split-sample calibration based on dynamic programming (SSC-DP) avoids
overly frequent changes of parameters, such as hourly or daily variations. Compared
with SSC, the distinctive element is that SSC-DP considers the parameters to be related
over adjacent sub-periods, and selects parameter sets with good performance for each
period and small differences between adjacent time steps. In this study, three aspects of
the proposed method are evaluated: (1) The performance of SSC-DP is compared with
that of existing methods in terms of the estimation of time-varying parameters; (2) The
applicability of SSC-DP to more complex hydrological models with a considerable
number of parameters; (3) The ability of SSC-DP to provide additional insights on
parameter variations and their correlations with the properties of real catchments. To
investigate the above issues, the proposed method is compared with SSC and EnKF in
two synthetic experiments (one with a two-parameter monthly model, the other with a
15-parameter daily model). SSC-DP is also applied to two real catchments for





parameter estimation under different environmental conditions.
The remainder of this paper is organized as follows. Section 2 describes the
proposed method, reference methods, and performance evaluation indices. Section 3
describes two synthetic experiments and two real catchment case studies for
comparison among different time-varying parameter estimation methods. Sections 4
and 5 present the results and discussion, respectively, before the conclusions to this
study are drawn in Sect. 6.

## 118 2. Methodology

The two hydrological models considered in this study are the TMWB and
Xinanjiang models. Their concepts and differences are presented in Sect. 2.1. To avoid
the prohibitive computational cost of the Xinanjiang model's calibration procedure,
sensitivity analysis is employed to select behavioral parameters with less uncertainty,
as outlined in Sect. 2.2. Three time-varying parameter estimation methods (SSC, SSC-
DP, and data assimilation) are then used to determine the variations in these behavioral
parameters, as described in Sect. 2.3. Finally, to evaluate the performance of the time-
varying parameter estimation methods, four evaluation criteria are selected and
formulated in Sect. 2.4.

### 128 2.1 Hydrological models

### 129 2.1.1 Two-parameter monthly water balance model

The TMWB model developed by Xiong and Guo (1999) is efficient for monthly
runoff simulations and forecasts (Dai et al., 2018; Guo et al., 2002; Kim et al., 2016;





Yang et al., 2017). The model requires monthly precipitation and potential
evapotranspiration as inputs. Its simplicity and efficiency of performance mean that
TMWB can easily be used to investigate the impacts of climate change (Deng et al.,
2016; Luo et al., 2019). Its outputs include monthly streamflow, actual
evapotranspiration, and soil moisture content index. The model has only two
parameters (Table 1), $C$ and $SC$. The parameter $C$ takes account of the effect of the
change of time scale when simulating actual evapotranspiration. The parameter $SC$
represents the field capacity.
**2.1.2 Xinanjiang model**
The Xinanjiang model (Zhao, 1992) is widely used in China (Li and Zhang, 2017;
Si et al., 2015; Yin et al., 2018). It takes precipitation and pan-evaporation data as inputs
and estimates the actual evapotranspiration, soil moisture storage, surface runoff,
interflow, and groundwater runoff from the watershed. The simulated streamflow is
calculated by summing the routing results of the surface, interflow, and groundwater
runoff (Sun et al., 2018). In this study, the surface runoff is routed by the instantaneous
unit hydrograph (Lin et al., 2014), while the interflow and groundwater runoff are
routed by the linear reservoir method (Jayawardena and Zhou, 2000). A schematic
overview of the model is presented in Fig. 1. The 15 parameters in the Xinanjiang model
are defined in Table 2.
There are two important differences between the TMWB and Xinanjiang models:
(1) TMWB is a monthly rainfall-runoff model, whereas the Xinanjiang model can run
on hourly or daily step sizes; (2) the TMWB model is much simpler and has fewer



parameters than the Xinanjiang model.
**2.2 Parameter sensitivity analysis method**
Sensitivity analysis is used to identify which parameters significantly affect the
performance of the Xinanjiang model and reduce the number of parameters to be
calibrated. Numerous sensitivity analysis methods are available, such as the Morris
method (Morris, 1991) and Sobol analysis (Sobol, 1993). The Morris method provides
similar results to Sobol analysis with a reduced computational burden (Rebolho et al.,
2018; Teweldebrhan et al., 2018; Yang et al., 2018).
The Morris method assumes that if parameters change by the same relative amount,
the parameter that causes the larger elementary effect is the more sensitive (King and
Perera, 2013). The elementary effect is calculated as follows:
$$EE_p(\theta_1, \theta_2, ..., \theta_{Np}, \Delta) = \frac{y(\theta_1, \theta_2, ..., \theta_{p-1}, \theta_p + \Delta, \theta_{p+1}, ..., \theta_{Np}) - y(\theta_1, \theta_2, ..., \theta_{Np})}{\Delta} \qquad (1)$$
where $\theta_p$ represents the $p$-th parameter; $\Delta$ is the relative amount; $Np$ is the total
number of parameters, and $y$ is the model output based on a particular parameter set.
Each parameter is changed in turn and every parameter set produces an elementary
effect. The parameter sensitivity is evaluated using the mean value $\mu$ of the
elementary effects. If a parameter has a higher value of $\mu$, it is more sensitive. In fact,
interactions between parameters should be taken into account (Jie et al., 2018). Hence,
the standard deviation $\sigma$ can be calculated. A higher value of $\sigma$ indicates a
stronger nonlinear correlation between parameters (Pappenberger et al., 2008).
**2.3 Time-varying parameter estimation method**



### 2.3.1 Split-sample calibration


SSC provides a simple way of diagnosing parameter non-stationarity under a
changing environment (Merz et al., 2011). As illustrated in Fig. 2(a), the method usually
has two steps (Hughes, 2015; Kim et al., 2015): (1) Available data are divided into
several consecutive periods, which can be arbitrarily chosen as hours, days, months,
seasons, or years; (2) Parameters are calibrated separately for the respective period.
This procedure gives better simulation performance than using constant parameters, but
leads to the estimated parameters fluctuating strongly over adjacent sub-periods,
producing false temporal variants.

### 2.3.2 Split-sample calibration based on dynamic programming


To overcome this problem, the SSC-DP method identifies time-varying parameters
with consideration of temporal continuity. SSC-DP has five steps (Fig. 2(b)):
(1) Split-sample periods. This process is the same as the first step of the SSC.
(2) Feasible parameter space generation. An ensemble of nearly optimal parameter
sets for each sub-period is obtained using Markov chain Monte Carlo (MCMC)
sampling (Chib and Greenberg, 1995). The likelihood measure of the $i$-th sub-period
links the parameter to observations using the Nash–Sutcliffe efficiency (NSE) (Nash
and Sutcliffe, 1970) as follows:
$$L_i(\theta) = 1 - \frac{\sum\limits_{t=(i-1)\times l+1}^{i\times l} (Q_t - \hat{Q}_t)^2}{\sum\limits_{t=(i-1)\times l+1}^{i\times l} (Q_t - \overline{Q}_t)^2} \tag{2}$$

where $Q_t$ and $\hat{Q}_t$ are the observed and simulated runoff at time step $t$, respectively,





and $l$ is the length of the sub-period.
(3) Dynamic programming optimization. The goal is to find parameters that
provide both good model performance and continuity. The continuity condition aims to
minimize the difference between the estimated parameters for sub-periods $i$ and $i+1$.
For $N$ sub-periods, the objective function can be expressed as follows:
$$\text{Max } F = \sum_{i=1}^{N}\left[(\text{NSE}_i + \text{NSE}_{ln,i} + \text{NSE}_{abs,i}) - \alpha \times \sum_{p=1}^{N_P} \frac{|\theta_{i+1,p} - \theta_{i,p}|}{\theta_{max,p} - \theta_{min,p}}\right] \qquad (3)$$
$$NSE_{\ln,i} = 1 - \frac{\sum_{t=(i-1)\times l+1}^{i\times l}(\ln(Q_t) - \ln(\hat{Q}_t))^2}{\sum_{t=(i-1)\times l+1}^{i\times l}(\ln(Q_t) - \ln(\overline{Q_t}))^2} \qquad (4)$$
$$NSE_{abs,i} = 1 - \frac{\sum_{t=(i-1)\times l+1}^{i\times l}\left|Q_t - \hat{Q}_t\right|}{\sum_{t=(i-1)\times l+1}^{i\times l}\left|Q_t - \overline{Q_t}\right|} \qquad (5)$$
where $\theta_{i,p}$ is the $p$-th estimated parameter over the $i$-th sub-period; $\theta_{max,p}$ and
$\theta_{min,p}$ are its maximum and minimum values, respectively; $N_P$ is the number of the
parameters; and $\alpha$ is the weight, reflecting parameter continuity. The weights of
$\text{NSE}_i$, $\text{NSE}_{ln,i}$, and $\text{NSE}_{abs,i}$ are set to 1 following the work of Merz et al. (2011), who
used equal weights for the NSE and its variants.
As the decision-making process during the current sub-period is related to that of
the previous sub-period, the parameter estimation over $N$ periods becomes a multi-stage
optimization problem. To solve this, a dynamic programming technique (Bellman, 1957)
is employed to decompose the optimization into a number of single-stage problems and
determine the optimal trajectory of the time-varying parameters. Dynamic
programming is a useful method for handling sequential operation decisions. It allows


the problem to be solved using a backward recursive procedure, whereby the decision-
making for each sub-period maximizes the sum of current and future benefits (Li et al.,
2018; Ming et al., 2017). In this study, the objective function is formulated as the
following recursive equation:
$$\begin{cases} F_i^* = max\{f_i[\vartheta_{i,1}, \vartheta_{i,2}, \vartheta_{i,3}, \cdots, \vartheta_{i,p}] + F_{i+1}^*\} \\ F_N^* = 0 \end{cases} \qquad (6)$$

where $F_i^*$ is the evaluation index using the optimal time-varying parameters from the
$N$-th to the $i$-th sub-periods, and Eq. (6) calculates the objective function from the $N$-th
sub-period to the first sub-period.

(4) Update initial states. The initial states, such as that of the soil water content,

are important in model simulation and calibration. As the final states for sub-period $i$
are not used as the initial states for sub-period $i$+1 during steps (1)–(3), the time-varying
parameter set obtained from step (3) is applied to the hydrological model to update the
initial states of each sub-period for the next iteration.

(5) Steps (1)–(4) are repeated until the initial states of each sub-period are

generally stable.
**2.3.3 Data assimilation**

Another approach for diagnosing variations in parameters is data assimilation,

using methods such as the EnKF and ensemble Kalman smoother (EnKS). These are
used here as reference methods. The EnKF has been widely applied to conceptual
models, including TMWB (Deng et al., 2016). Li et al. (2013) noted that the EnKF
struggles to handle the time-lag in routing processes. However, the routing component





is vital to the Xinanjiang model. EnKS can efficiently determine the states of the
Xinanjiang model (Meng et al., 2017), but the estimation of routing parameters deserves
discussion. Most previous studies have used a fixed distribution of the routing
hydrograph in data assimilation (Lu et al., 2013), i.e., the parameters are constant for
routing processes. With respect to these issues, a modified EnKF (named SSC-EnKF)
is established as a third data assimilation reference method in the synthetic experiment
with the Xinanjiang model (described in Sect. 3.1).

The EnKF includes two main steps: model prediction and assimilation. The state

vector is augmented with parameter variables so that time-varying parameters can be
estimated simultaneously with model states. For model prediction, the augmented
vector is derived by adding noise on that from the previous time step through the
following equation:
$$\begin{pmatrix} \vartheta_{t+1}^{k-} \\ x_{t+1}^{k-} \end{pmatrix} = \begin{pmatrix} \vartheta_t^{k+} \\ f\left(x_t^{k+}, \theta_{t+1}^{k-}, u_{t+1}\right) \end{pmatrix} + \begin{pmatrix} \delta_t^k \\ \varepsilon_t^k \end{pmatrix}, \; \delta_t^k \sim N\left(0, R_t\right), \varepsilon_t^k \sim N\left(0, G_t\right) \qquad (7)$$

where $\vartheta_t$ is the parameter vector at time step $t$, represented as $\left(\theta_{t,1}, \theta_{t,2}, ..., \theta_{t,Np}\right)$;
$x_t$ is the state vector; $\vartheta_{t+1}^{k-}$ and $x_{t+1}^{k-}$ are the $k$-th ensemble member forecasts at time
step $t$+1; $\vartheta_t^{k+}$ and $x_t^{k+}$ are the updated values of the $k$-th ensemble member forecasts
at time step $t$; $u_{t+1}$ denotes the forcing data (e.g., precipitation) at time step $t$+1; $\delta_t^k$
and $\varepsilon_t^k$ are the white noise for the $k$-th ensemble member, which follow a Gaussian
distribution with zero mean and specified covariance of $R_t$ and $G_t$, respectively.

In the assimilation process, the augmented vector is updated using the following

equations if suitable observations are available:





$$\begin{pmatrix} x_{t+1}^{k+} \\ \vartheta_{t+1}^{k+} \end{pmatrix} = \begin{pmatrix} x_{t+1}^{k-} \\ \vartheta_{t+1}^{k-} \end{pmatrix} + \begin{pmatrix} K_{t+1}^{x} \left[ y_{t+1}^{k} - \widehat{y}_{t+1}^{k} \right] \\ K_{t+1}^{\vartheta} \left[ y_{t+1}^{k} - \widehat{y}_{t+1}^{k} \right] \end{pmatrix}$$ (8)

$$y_{t+1}^{k} = y_{t+1} + \xi_{t+1}^{k}, \ \xi_{t+1}^{k} \sim N\left(0, W_{t}\right),$$ (9)

$$\widehat{y}_{t+1}^{k} = h(x_{t+1}^{k-}, \vartheta_{t+1}^{k-})$$ (10)

where $y_{t+1}$ is the observation vector at time $t$+1; $y_{t+1}^{k}$ is the $k$-th observation ensemble
member at time step $t$+1; $\widehat{y}_{t+1}$ is the simulation vector at time $t$+1; $h$ is the
observational operator that converts the model states to observations; $\xi_{t+1}^{k}$ is the
measurement error, which follows a Gaussian distribution with a covariance of $W_{t}$ ;
and $K_{t+1}^{k}$ is the Kalman gain matrix (for details, see Feng et al., 2017).

The EnKS is based on the EnKF. Whereas the EnKF updates the model states and

parameters at the current time step, the EnKS takes account of those values over the
past time steps. The main steps of the EnKS are identical to those of the EnKF, but the
equation of the assimilation process is formulated as follows:
$$\begin{pmatrix} x_{t+1 \to t-n+2}^{k+} \\ \vartheta_{t+1 \to t-n+2}^{k+} \end{pmatrix} = \begin{pmatrix} x_{t+1 \to t-n+2}^{k-} \\ \vartheta_{t+1 \to t-n+2}^{k-} \end{pmatrix} + \begin{pmatrix} K_{t+1}^{x*} \left[ y_{t+1}^{k} - \widehat{y}_{t+1}^{k} \right] \\ K_{t+1}^{\vartheta*} \left[ y_{t+1}^{k} - \widehat{y}_{t+1}^{k} \right] \end{pmatrix}$$ (11)

$$\widehat{y}_{t+1}^{k} = h(x_{t+1 \to t-n+2}^{k-}, \vartheta_{t+1 \to t-n+2}^{k-})$$ (12)

where $K_{t+1}^{*}$ is the Kalman gain matrix of EnKS. The fixed time window $n$ of EnKS
is pre-determined based on the response function or unit hydrograph. Meng et al.
(2017) suggested that the time window should be set as half of the recession time of
a flood.

A third data assimilation approach is constructed based on the SSC. Instead of

assimilating one observed variable, it assimilates the observed variables during a given
period in one assimilation process. Assuming that the parameters are constant in the





given period, the equation of the assimilation process for the $i$-th sub-period is
expressed as follows:
$$\begin{pmatrix} x_{i+1}^{k+} \\ \vartheta_{i+1}^{k+} \end{pmatrix} = \begin{pmatrix} x_{i+1}^{k-} \\ \vartheta_{i+1}^{k-} \end{pmatrix} + \begin{pmatrix} K_{i+1}^{x*}\left[ y_{i\times l+1\to(i+1)\times l,}^{k} - \widehat{y}_{i\times l+1\to(i+1)\times l}^{k} \right] \\ K_{i+1}^{\vartheta*}\left[ y_{i\times l+1\to(i+1)\times l}^{k} - \widehat{y}_{i\times l+1\to(i+1)\times l}^{k} \right] \end{pmatrix} \quad (13)$$

$$\widehat{y}_{i\times l+1\to(i+1)\times l}^{k} = h(x_{i+1}^{k-}, \vartheta_{i+1}^{k-}) \quad (14)$$

where $\vartheta_i$ is the parameter vector for sub-period $i$, represented as $\left(\theta_{i,1}, \theta_{i,2}, ...., \theta_{i,Np}\right)$;
$x_i$ is the initial state vector for sub-period $i$; and $l$ is the length of the sub-period.

This approach addresses the routing-lag issue by allowing parameters of the

routing processes, such as the instantaneous unit hydrograph, to remain constant for
each sub-period and to be time-varying over the whole period.
**2.4 Model evaluation criteria**

The streamflow simulations and parameter estimations given by the proposed

time-varying parameter estimation approach are verified using the NSE, root mean
square error (RMSE), and Pearson correlation coefficient ($R^2$). The simulated
streamflow is evaluated using the NSE. A higher NSE value indicates a better
simulation.

The estimated parameters are evaluated by the RMSE (Alvisi et al., 2006) and $R^2$

(Kim et al., 2007). For the $p$-th parameter, the formulations are as follows:
$$RMSE_p = \sqrt{\frac{1}{m}\sum_{t=1}^{m}(\theta_{t,p} - \widehat{\theta}_p)^2} \quad (15)$$





$$R^2{}_p = \frac{\sum_{t=1}^{m}\left(\widehat{\theta}_{t,p} - \overline{\widehat{\theta}}_p\right)\left(\theta_{t,p} - \overline{\theta}_p\right)}{\sqrt{\sum_{t=1}^{m}\left(\theta_{t,p} - \overline{\widehat{\theta}}_p\right)^2\left(\theta_{t,p} - \overline{\theta}_p\right)^2}}$$
(17)

where $\theta_t$ and $\widehat{\theta}_t$ are the true parameter and its estimated value at the $t$-th time step,
respectively; $\overline{\theta}_p$ and $\overline{\widehat{\theta}}_p$ are the mean value of the true parameters and its estimated
values, respectively; and $m$ is the length of the data during the whole period. RMSE
quantifies the accuracy of the estimated parameters, and $R^2$ records the overall
agreement between the true and estimated parameters. Smaller values of RMSE and
higher values of $R^2$ indicate stronger parameter identification ability. A Taylor diagram
is used to summarize the standard deviation, RMSE, and $R^2$ in a polar plot, providing a
graphical representation of the performance of SSC-DP.
## 3.   Data and study area

Two synthetic experiments and two real catchment case studies were designed to

assess the performance of SSC-DP. The experiments are described in Table 3.
### 3.1 Synthetic experiments

The two synthetic experiments examine the ability of SSC-DP to identify the time-

varying parameters of the TMWB and Xinanjiang hydrological models. The merit of
synthetic experiments is that the parameters can be synthetically generated to be either
constant or time varying. Hence, it is convenient to compare the estimated values with
the a priori known parameters to evaluate different parameter estimation methods. Note
that synthetic experiments have been successfully used in several time-varying

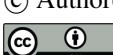



parameter identification studies (Deng et al., 2016; Pathiraja et al., 2016; Xiong et al.,

2019).

**3.1.1 Synthetic experiment with the TMWB model**
Synthetic data of monthly precipitation and potential evapotranspiration were
collected from the 03451500 catchment of the Model Parameter Estimation Experiment
(MOPEX) (Duan et al., 2006). The data cover 252 months. Runoff was derived by the
TMWB model using synthetic precipitation, potential evapotranspiration, and the
known parameters. Gaussian noise was added to the simulated runoff to represent
uncertainties. The mean of the noise was set to zero, and the standard deviation was
assumed to be 3 % of the magnitude of the values (Deng et al., 2016).
Eight scenarios with different known parameters were investigated (Table 4). The
first scenario considered constant parameters. Scenarios 2 and 3 considered month-by-
month variations in TMWB model parameters, i.e., the parameters remain constant
during each month, but change from month to month. Scenarios 4 and 5 considered
parameters that change every six months. Scenarios 6–8 considered year-by-year
variations. The changes in both $C$ and $SC$ were considered to be linear in scenarios 2,
4, and 6 (Trend) and sinusoidal in scenarios 3, 5 and 7 (periodicity), reflecting the
impacts of climate change and human activities (Pathiraja et al., 2016). Scenario 8
considered a periodic variation with an increasing trend for parameter $C$ and only the
linear variation in $SC$.





**3.1.2 Synthetic experiment with the Xinanjiang model**
Hourly precipitation and pan evaporation data were collected from the Baiyunshan
Reservoir basin in China. The data cover a period of 18000 h. The Xinanjiang model
has 15 parameters, which can lead to a significant computational burden. To reduce the
total number of model runs, only the sensitive parameters were considered to be free.
The Morris method was used to detect the free parameters (Fig. 3), with the results
showing that *KE*, *CI*, *CG*, *KI*, *KG*, and *NK* are sensitive parameters. Thus, the other
parameters were held constant for the whole period.
Similar to the experiment with the TMWB model, synthetic runoff was derived
from the Xinanjiang model with added Gaussian noise. The mean of the noise was set
to zero, and the standard deviation was assumed to be 5 % of the magnitude of the
values. As presented in Table 5, all 15 parameters were set to be constant in the first
scenario. The known sensitive parameters were considered to vary with a certain trend
and periodicity in scenarios 2 and 3, respectively. Scenario 4 considered a combined
variation of trend and periodicity for the parameter *KE*, with the other free parameters
set to vary linearly. The parameter variations in scenarios 2–4 were assumed to occur
once a year.
**3.2 Study area: Wuding River basin**
The Wuding River basin (Fig. 4(a)) examined in the first case study is a large sub-
basin of the Yellow River basin located on the Loess Plateau (Xu, 2011). The Wuding
River has a drainage area of 30261 km$^2$ and a total length of 491 km. The average slope





is 0.2 %, and the elevation varies from 600–1800 m above sea level. The area is a semi-
arid region with mean annual precipitation of ~401 mm. The annual potential
evapotranspiration is 1077 mm, and the mean annual runoff is 39 mm. The data for this
basin were collected over the period 1958–2000. The daily precipitation was obtained
from Thiessen polygons using records from 122 rain gauges. Based on meteorological
data from the China Meteorological Data Sharing Service System (http://data.cma.cn),
areal pan evaporation data were obtained. As illustrated in Fig. 4(a), the station furthest
downstream, Baijiachuan, drains an area of 29,662 km$^2$ (98 % of the total basin) and
records the daily runoff data.
The erosion of loess, vegetable degradation, and human activities mean that the
Wuding River basin suffers severe soil erosion. Soil and water conservation measures,
such as reservoir construction and afforestation, have been undertaken since the 1960s.
Several studies have reported the anthropogenic impacts of this area and demonstrated
the changing relationship between precipitation and runoff (Gao et al., 2017; Jiao et al.,

2017).

**3.3 Study area: Xun River basin**
The proposed method was also applied to the Xun River basin, China (Fig. 4(b)).
Located between 108°24'–109°26' E and 32°52'–33°55' N, the study area covers
approximately 6448 km$^2$. The Xun River is ~218 km long and has an average annual
flow of 73 m$^3$/s (Li et al., 2016). The basin has a subtropical monsoon climate. The
weather is wet and moderate with an annual average temperature of 15–17 ℃. The daily





hydrological data from 1991–2001 include precipitation from 28 rainfall stations, pan
evaporation from three hydrological gauged stations, and discharge at the outlet of the
Xun River basin. Areal precipitation was obtained using the Thiessen polygon method,
and areal pan evaporation was computed using the average value of the data from
gauged stations.
As a tributary of the Han River, climatic impacts are important factors in the South-
to-North water diversion project. Given that the majority of rainfall (approximately 70–
80 % of the total) occurs in the summer, seasonal variations should also be considered.

## 4.  Results

### 4.1 Synthetic experiment

### 4.1.1 Results of synthetic experiment with the TMWB model

When using SSC-DP, the first task is to define how the hydrological data series
should be split into the $k$ sub-periods within which the parameters are assumed to be
constant. As climate change can induce seasonal or half-annual variations while human
activities usually influence the watershed annually, lengths of three months, six months,
and 12 months were arbitrarily chosen. Thus, this experiment compared the following
four methods: (1) EnKF; (2) 3-SSC-DP; (3) 6-SSC-DP, and (4) 12-SSC-DP.
Table 6 presents the runoff simulation performance for various scenarios. There is
little difference among the four methods in terms of NSE, with all NSE values higher
than 99 % in scenarios 1, 2, 4, 5, 6, 7, and 8. In scenario 3, the NSE values of 6-SSC-
DP and 12-SSC-DP decrease significantly, because the assumed sub-period length is

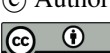



longer than the time-scale of actual variations. This issue does not appear in scenario 2,
where the NSEs of 6-SSC-DP and 12-SSC-DP are greater than those of 3-SSC-DP and
EnKF. All of the SSC-DP methods exhibit superior simulation performance compared
with ENKF in scenarios 1, 4, 5, 6, and 8, and both 6-SSC-DP and 12-SSC-DP have
higher NSEs than EnKF in scenario 7. SSC-DP offers improved accuracy if the proper
length is chosen.
Figure 5 focuses on the ability of the four methods to identify time-varying
parameters. In scenario 1, 12-SSC-DP gives the best performance (see Fig. 6(b)).
Although 6-SSC-DP and EnKF give similar estimates for *SC* in this scenario, 6-SSC-
DP gives a better estimation of parameter *C* with a lower RMSE. When the synthetic
true parameters vary linearly (scenarios 2, 4, and 6), 12-SSC-DP produces a lower
RMSE and higher $R^2$ for *C* and *SC* in comparison with EnKF, 3-SSC-DP, and 6-SSC-
DP, regardless of the difference between assumed and actual sub-period lengths. When
the synthetic true parameters vary sinusoidally (scenarios 3, 5, and 7), the results are
more complex. EnKF gives the best performance in scenario 3, where the parameters
vary sinusoidally from month to month. The poor performance of 6-SSC-DP and 12-
SSC-DP can be explained by the assumed length being much longer than the actual one,
preventing parameter variations over short timescales from being identified. When the
parameters vary periodically at six-month intervals (scenario 5), 6-SSC-DP yields the
best performance (Fig. 6(a)). For the annual variation in parameters (scenario 7), 12-
SSC-DP produces the optimal results. It can be inferred that a proper period length is
important for identifying sinusoidal variations, but does not limit the ability of SSC-DP


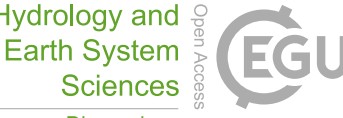

to identify linear variations. Note that in scenario 5, where the parameters vary
sinusoidally at six-month intervals, 12-SSC-DP and EnKF provide similar performance.
This indicates that, even when the lengths are unsuitable, SSC has the potential to
achieve comparable results to EnKF. Further evidence is that both 12-SSC-DP and 6-
SSC-DP are generally better than EnKF in scenario 8, where $C$ has a combined variation
from year to year.

The 3-SSC-DP model fails to detect parameter variations among different

scenarios, because the assumed length that serves as a calibration period for MCMC is
too short (i.e., three months). Thus, the estimated parameters are associated with higher
uncertainties.
**4.1.2 Results of synthetic experiment with the Xinanjiang model**

The Xinanjiang model is more complex than TMWB, and so some sensitivity

analysis is necessary. As stated in Sect. 3.1.2, the sensitive parameters are $KE$, $CI$, $CG$,
$KI$, $KG$, and $NK$. The 18000 hourly hydrological data points were divided into 25 sub-
periods (monthly time scale) and 12 sub-periods (bimonthly time scale). It is considered
that a monthly time scale helps diagnose seasonal variations, whereas a two-monthly
time scale provides data for longer calibration lengths.

Three data assimilation methods (see Sect. 2.3.2 for details) were applied to the

synthetic data: (1) EnKF; (2) EnKS, and (3) SSC-EnKF. The results in Table 7 indicate
that EnKS is superior to EnKF, as previously observed (Li et al., 2013), although SSC-
EnKF gives the best results. This is probably because SSC-EnKF is based on the





assumption that the parameters remain constant during each sub-period.
The simulated streamflow and identification of time-varying parameters was
compared across four methods: 1-SSC, SSC-EnKF, 1-SSC-DP, and 2-SSC-DP. The
simulation performance is summarized in Table 8. The NSE value of 1-SSC-DP is lower
than that of 1-SSC for scenarios 2 and 4. Overall, the difference between the simulation
ability of 1-SSC-DP and 1-SSC is slight, because 1-SSC-DP is based on 1-SSC with
more continuous parameters at a cost of some accuracy. However, the NSE value of 1-
SSC-DP is greater than that of 1-SSC for scenario 3. This may be because the
parameters estimated by 1-SSC introduce more uncertainty than SSC-DP when the
"true" parameters fluctuate sinusoidally. The simulation performance of 1-SSC-DP is
superior to that of SSC-EnKF, but 2-SSC-DP is inferior to SSC-EnKF. That is because,
when there are more sub-periods (but the length of each sub-period is not too short),
the performance tends to be better.
Figure 7 compares the time-varying parameter estimation performance among the
four methods. There is a slight difference between 1-SSC and 1-SSC-DP for both
scenarios 1 (constant) and 2 (trend). However, 1-SSC-DP offers a significant
improvement over 1-SSC in scenarios 3 (period) (Fig. 8) and 4 (combination). This
indicates that SSC-DP selects more continuous parameters and provides lower RMSE
and higher $R^2$ values when identifying time-varying parameters.
Model 1-SSC-DP significantly outperforms SSC-EnKF under any scenario for
parameter $KE$ in terms of RMSE and $R^2$. For the other parameters, SSC-EnKF gives a
slightly better estimation than 1-SSC-DP. However, the difference is small in terms of



RMSE and $R^2$. In particular, 1-SSC-DP generates better estimations than SSC-EnKF in
scenario 3 for parameters *CG* and *KI*. The results indicate that SSC-DP achieves high-
quality, robust parameter estimations when a proper period length is specified. Figure
7 provides further evidence that 2-SSC-DP performs better than 1-SSC-DP and SSC-
EnKF in scenarios 1 (constant) and 2 (trend), giving a lower RMSE and higher $R^2$.
However, it performs worse than 1-SSC-DP and SSC-EnKF in scenarios 3 (period) and
4 (combination).

**4.2 Case study: Wuding River basin**

Figures 9(a) and 9(b) show the double mass curves between daily runoff and
precipitation for the Wuding River basin. Similar to the work of Deng et al. (2016), the
two linear slopes of the curves are different before and after 1972, demonstrating the
relationship between precipitation and runoff changes under the soil and water
conservation measures. This suggests that there are annual variations in the watershed
characteristics. Hence, the length of each sub-period was set to 12 months, and the time-
varying parameters were identified using 12-SSC-DP. Based on daily Wuding data from
1958–2000, sensitivity analysis showed that nine parameters of the Xinanjiang model
are relatively sensitive: *WM*, *WUM*, *WLM*, *KE*, *IMP*, *KI*, *KG*, *N*, and *NK*.
The simulation results given by 12-SSC-DP were benchmarked against those from
12-SSC, data assimilation, and the conventional method in which all Xinanjiang model
parameters remain constant. The simulation performance of the conventional method
is presented in Fig. 10(a) (NSE = 40.7 %). Model 12-SSC-EnKF gives the best

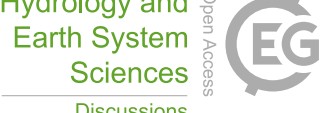

simulation results among the data assimilation methods described in Sect. 2.3.2, with
an NSE value of 38.0 % (Fig.10 (b)). The inferior performance compared with the
conventional method is probably because 12-SSC-EnKF is inapplicable to such a semi-
arid catchment and there are too many parameters to be updated. The performance of
12-SSC (NSE = 48.5 %) and that of 12-SSC-DP (NSE = 51.3 %) are illustrated in Figs.
10(c) and 10(d), respectively. It is evident that 12-SSC and 12-SSC-DP can significantly
improve the simulation performance of the Xinanjiang model in this semi-arid region.

Although the objective function of 12-SSC-DP considers the trade-off between

simulation accuracy and parameter continuity, 12-SSC-DP gives a higher NSE value.
This may be because 12-SSC locates a local peak over one sub-period, resulting in
unreasonable model states for the beginning of the next sub-period, whereas 12-SSC-
DP uses dynamic programming to explore more reasonable parameter values and model
states. Figure 10 shows the quantile-quantile plots, from which it can be seen that if the
parameters are assumed to be constant, streamflow is highly underestimated. Models
12-SSC and 12-SSC-DP reduce this underestimation by using time-varying parameters.
Additionally, 12-SSC-DP is slightly inferior to 12-SSC in terms of peak flows, but is
superior in terms of simulating streamflow lower than 100 m$^3$/s, which accounts for 80 %
of the whole streamflow time series. It can be inferred the 12-SSC-DP is more
applicable to the simulation of streamflow in semi-arid regions.

The estimated time-varying parameters estimated by 12-SSC-DP are plotted in Fig.

11. The results show that *WM* remains constant before and after 1972, but *WUM* varies
significantly over this period, indicating that the distribution of soil water capacity may





change, i.e., *WUM* decreases but *WLM* increases. It can be inferred that less severe soil
erosion occurred, because the upper layers became thinner while the lower layer, where
vegetation roots dominate, became thicker (Jayawardena and Zhou, 2000). *KE* changes
slightly, suggesting reduced impacts on the ratio of potential evapotranspiration to pan
evaporation. Similarly, the differences in *KI*, *KG*, *N*, and *NK* before and after 1972 are
not significant. However, *IMP* decreases significantly, indicating a reduction in
impervious areas of the basin. This can be attributed to the soil and water conservation
measures, especially the implementation of tree and grass plantations and land terracing.
The variations in *WLM* and *IMP* slowed down after the turning point, similar to the
results of Deng et al. (2016).
**4.3 Case study: Xun River basin**

Figures 9(c) and 9(d) show the double mass curves between runoff and

precipitation for the Xun River basin. The linear slope of the curve is generally
stationary for the whole ten-year period shown in Fig. 9(c), with a correlation
coefficient of 99.6 %. In contrast, the linear slope for an intra-annual timescale is non-
stationary (Fig. 9(d)). Based on these results, it can be inferred that the relationship
between precipitation and runoff is stable from 1990–2000, but varies over the intra-
annual timescale. Hence, sub-periods of three and 12 months were examined in the Xun
River basin using models 3-SSC-DP and 12-SSC-DP. From the Xun River basin data
from 1991–2000, sensitivity analysis suggested that five parameters of the Xinanjiang
model are relatively sensitive, namely *KE*, *B*, *KI*, *KG*, and *NK*.





Similar to the case study of the Wuding River basin, the simulation performance
of 3-SSC-DP was benchmarked against that of 3-SSC, data assimilation, and the
conventional calibration method. Among the data assimilation methods described in
Sect. 2.3.2, 3-SSC-EnKF gives the highest simulation accuracy. All methods performed
well, with NSE values of 92.5 %, 93.0 %, 95.0 %, and 94.8 % for the conventional
method, 3-SSC-EnKF, 3-SSC, and 3-SSC-DP, respectively. This shows that 3-SSC-DP
outperforms 3-SSC-EnKF, while 3-SSC gives slightly better performance than 3-SSC-
DP, which can be intuitively attributed to the multi-objective function used in the SSC-
DP method. This result is different from that in the Wuding case study. Here, the reason
can be attributed to the fact that streamflow is easier to model in wet regions, i.e., the
Xun River basin, in which the parameters and model states of each sub-period have less
uncertainty. However, the uncertainty increases in dry regions, i.e., the Wuding River
basin. The superior performance in the Wuding River basin suggests that SSC-DP is
more useful when simulating streamflow in dry regions (or periods).
The estimated parameters using 3-SSC-DP are presented in Fig. 12(a). Some
parameters vary significantly over an intra-annual time scale. Among them, the
parameter *KE*, representing the ratio of potential evapotranspiration to pan evaporation,
exhibits the most distinct seasonal variations. A fast Fourier transform was used to
calculate the spectral power of the *KE* time series to explore its periodic characteristics.
As can be observed from Fig. 12(b), 3-SSC-DP had the greatest spectral power, for a
period of 4.0 cycles per year, somewhat higher than the power obtained by 3-SSC and
3-SSC-EnKF. This means a stronger periodic pattern is captured by 12-SSC-DP. Given





that the estimated *KE* varies at three-monthly intervals, it has a one-year periodicity.
The other parameters do not exhibit significant one-year periodic patterns. This may be
because only *KE*, linking potential evapotranspiration and pan evaporation, is directly
impacted by seasonal climate variations, such as temperature.

In this experiment, 12-SSC-DP was also applied to the Xun River basin and

benchmarked against 12-SSC and the conventional method. The resulting NSE values
of 93.2 % for 12-SSC and 93.1 % for 12-SSC-DP are similar to those in the above
analysis. The simulation performance decreases slightly from 12-SSC to 12-SSC-DP
because of the tradeoff between simulation accuracy and parameter continuity. The
estimated time-varying parameters using 12-SSC are plotted in Fig. 12(c). As can be
seen, five sensitive parameters vary over a relatively small extent compared to the
parameter ranges listed in Table 2, indicating that the associated watershed
characteristics of Xun River basin lacked a strong temporal pattern over an annual scale
during 1990 to 2000. Hence, seasonal climate variability is the main cause of the non-
stationarity in the hydrological processes of the Xun River basin. This finding is in
agreement with the work of Lan et al. (2018), who also recognized and analyzed the
seasonal hydrological dynamics of this study region.

## 565   5.   Discussion

As noted in the methodology section, the performance of the proposed method is

influenced by several factors, such as the weights in the objective function and the
choice of lengths. Some suggestions regarding the improvement of the proposed
approach are now discussed in detail.





## 5.1 Objective function of dynamic programming in SSC-DP

In the conventional method, a parameter set is identified as optimal for providing the best simulation over the calibration period. However, other parameter sets with slightly worse (but still good) performance can also be candidates. Allowing for input data uncertainty and local optima, SSC-DP identifies parameter sets that perform near-optimally and display less fluctuations over sub-periods. This can be adjusted by weights in the objective function of the dynamic programming approach (see Eq. (3)). As the weighting for accuracy increases, parameters providing more accurate simulations are chosen, but parameter continuity is less important. If too much importance is given to continuity, the variations in real world processes may be underestimated. Here, the influence of different weights has been assessed for simulation accuracy and parameter continuity based on synthetic experiments with the TMWB and Xinanjiang models, respectively. Specifically, the weight for simulation accuracy was set to 1, and the weight for parameter continuity $\alpha$ varied from zero to a small positive value (e.g., 1). When $\alpha = 0$, only simulation accuracy was considered.

Figure 13(a) shows the $R^2$ value of 12-SSC-DP with various continuity weights for scenario 4 in the synthetic experiment with the TMWB model. It can be seen that $R^2$ is lowest when $\alpha = 0$ for both $C$ and $SC$. There is some improvement when a nonzero weight is applied. As $\alpha$ increases, the performance of 12-SSC-DP improves, and then worsens; the differences among schemes with nonzero weights are not distinct. Similar results can be observed in Fig. 13(b), which presents the $R^2$ value of 12-SSC-DP with various $\alpha$ for scenario 2 in the synthetic experiment with the Xinanjiang model.



Therefore, nonzero continuity weights can significantly improve the parameter
estimation performance compared with the zero-weight case. It is suggested that
weights of 1 (accuracy) and 0.005 (continuity) be used with the TMWB model and
weights of 1 (accuracy) and 0.2 (continuity) be applied with the Xinanjiang model, as
in this study.
**5.2 Choice of sub-period length in SSC-DP**

As mentioned by Gharari et al. (2013), there are different ways of determining the

sub-period lengths. The sub-periods can be non-continuous hydrological years (Seiller
et al., 2012), months or seasons (Deng et al., 2018; Paik et al., 2005), and discharge or
precipitation events (Singh and Bardossy, 2012). This introduces a controversial issue
whereby parameters are impacted by the length of the calibration period. Merz et al.
(2009) suggested that 3–5 years is an acceptable calibration period, whereas Singh and
Bardossy (2012) indicated that a small number of events may be sufficient for
parameter identification. As reported in Sect. 4, the length should be neither too long
nor too short so as to increase the reliability of the calibration while guaranteeing that
variations in real processes are captured. Thus, given that the time scale of variations is
unknown, the proposed SSC-DP can be used with different split-sample lengths. It is
suggested that the length should be as long as possible without degrading the simulation
performance. For example, in the synthetic experiment with the TMWB model, if the
difference between the NSE values of 6-SSC-DP and 3-SSC-DP are small, the preferred
length is six months.





However, many studies are based on the conventional assumption that parameters
of different sub-periods are independent. Hence, the sub-period lengths should be long
enough to reduce the degree of uncertainty. In this study, the assumption of parameter
continuity is introduced to give another constraint that considers correlations between
parameters of adjacent sub-periods. It appears that the determination of sub-period
lengths deserves further investigation.
## 6. Conclusions
This paper has described a time-varying parameter estimation approach based on
dynamic programming. The proposed SSC-DP combines the basic concept of SSC and
the continuity assumption of data assimilation to estimate more continuous parameters
while providing comparably good streamflow simulations. Two synthetic experiments
were designed to evaluate its applicability and efficiency for time-varying parameter
identification. Furthermore, two case studies were conducted to explore the advantages
of SSC-DP in real catchments. From the results, the following conclusions can be drawn:
1. One synthetic experiment used the TMWB model with two parameters and eight
scenarios, and the results indicate that the impact of sub-period lengths on the
performance of SSC-DP is significant when the known parameters vary sinusoidally.
Using a suitable length not only produces better simulation performance, but also
ensures that the parameter estimates are more accurate than with SSC and EnKF.
2. The second experiment involved the Xinanjiang model with 15 parameters and
four scenarios. A sensitivity analysis was performed to reduce the probable
computational cost and improve the efficiency of identifying the time-varying





parameters. The results were similar to those in the first experiment, demonstrating that
SSC-DP has the potential to deal with more complex models.
3. In a case study applied to the Wuding River basin, SSC-DP produced the best
simulation performance. Additionally, it detected that parameters reflecting soil water
capacity and impervious areas changed significantly after 1972, reflecting the soil and
water conservation projects carried out from 1958–2000. A second case study focused
on the Xun River basin. The results from a fast Fourier transform suggest that SSC-DP
detects stronger one-year periodicity than other methods, indicating the distinct impacts
of seasonal climate variability. Thus, SSC-DP can be used to detect the relationship
between the temporal variations of parameters and the changing environment in real
catchments.
This study has demonstrated that the proposed method is an effective approach for
identifying time-varying parameters under changing environments. Further work is still
needed, such as to determine an objective method for choosing the sub-period lengths.

## Acknowledgements

This study was supported by the Natural Science Foundation of Hubei Province
(2017CFA015), the National Natural Science Foundation of China (51861125102), and
Innovation Team in Key Field of the Ministry of Science and Technology
(2018RA4014). The authors would like to thank the editor and anonymous reviewers
for their comments that helped improve the quality of the paper.



**Code/Data availability**
The data and codes that support the findings of this study are available from the
corresponding author upon request.

**Author contribution**
All of the authors helped to develop the method, designed the experiments, analyzed
the results and wrote the paper.

**Compliance with Ethical Standards**
**Conflict of Interest** The authors declare that they have no conflict of interest.

Alvisi, S., Mascellani, G., Franchini, M., Bardossy, A., 2006. Water level forecasting through fuzzy logic
and artificial neural network approaches. Hydrology and Earth System Sciences 10(1), 1-17.
Bellman, R., 1957. Dynamic programming. Princeton University Press, Princeton.
Broderick, C., Matthews, T., Wilby, R.L., Bastola, S., Murphy, C., 2016. Transferability of hydrological
models and ensemble averaging methods between contrasting climatic periods. Water
Resources Research 52(10), 8343-8373.
Bronstert, A., 2004. Rainfall-runoff modelling for assessing impacts of climate and land-use change.
Hydrological Processes 18(3), 567-570.
Chen, Y., Zhang, D., 2006. Data assimilation for transient flow in geologic formations via ensemble
kalman filter. Advances in Water Resources 29(8), 1107-1122.
Chib, S., Greenberg, E., 1995. Understanding the metropolis-hastings algorithm. American Statistician
49(4), 327-335.
Coron, L. et al., 2012. Crash testing hydrological models in contrasted climate conditions: An experiment
on 216 australian catchments. Water Resources Research 48.
Dai, C., Qin, X.S., Chen, Y., Guo, H.C., 2018. Dealing with equality and benefit for water allocation in a
lake watershed: A gini-coefficient based stochastic optimization approach. Journal of
Hydrology 561, 322-334.
Dakhlaoui, H., Ruelland, D., Tramblay, Y., Bargaoui, Z., 2017. Evaluating the robustness of conceptual
rainfall-runoff models under climate variability in northern tunisia. Journal of Hydrology 550,
201-217.



Deng, C., Liu, P., Guo, S., Li, Z., Wang, D., 2016. Identification of hydrological model parameter variation
using ensemble kalman filter. Hydrology and Earth System Sciences 20(12), 4949-4961.
Deng, C., Liu, P., Wang, D., Wang, W., 2018. Temporal variation and scaling of parameters for a monthly
hydrologic model. Journal of Hydrology 558, 290-300.
Deng, C., Liu, P., Wang, W., Shao, Q., Wang, D., 2019. Modelling time-variant parameters of a two-
parameter monthly water balance model. Journal of Hydrology 573, 918-936.
Duan, Q. et al., 2006. Model parameter estimation experiment (mopex): An overview of science strategy
and major results from the second and third workshops. Journal of Hydrology 320(1-2), 3-17.
Feng, M. et al., 2017. Deriving adaptive operating rules of hydropower reservoirs using time-varying
parameters generated by the enkf. Water Resources Research 53(8), 6885-6907.
Fowler, K., Peel, M., Western, A., Zhang, L., 2018. Improved rainfall-runoff calibration for drying climate:
Choice of objective function. Water Resources Research 54(5), 3392-3408.
Fowler, K.J.A., Peel, M.C., Western, A.W., Zhang, L., Peterson, T.J., 2016. Simulating runoff under
changing climatic conditions: Revisiting an apparent deficiency of conceptual rainfall-runoff
models. Water Resources Research 52(3), 1820-1846.
Gao, S. et al., 2017. Derivation of low flow frequency distributions under human activities and its
implications. Journal of Hydrology 549, 294-300.
Gharari, S., Hrachowitz, M., Fenicia, F., Savenije, H.H.G., 2013. An approach to identify time consistent
model parameters: Sub-period calibration. Hydrology and Earth System Sciences 17(1), 149-
161.

Guo, S.L., Wang, J.X., Xiong, L.H., Ying, A.W., Li, D.F., 2002. A macro-scale and semi-distributed monthly
water balance model to predict climate change impacts in china. Journal of Hydrology 268(1-
4), 1-15.

Guzha, A.C., Rufino, M.C., Okoth, S., Jacobs, S., Nobrega, R.L.B., 2018. Impacts of land use and land cover
change on surface runoff, discharge and low flows: Evidence from east africa. Journal of
Hydrology-Regional Studies 15, 49-67.
Hughes, D.A., 2015. Simulating temporal variability in catchment response using a monthly rainfall-
runoff model. Hydrological Sciences Journal-Journal Des Sciences Hydrologiques 60(7-8), 1286-
1298.

Hundecha, Y., Bardossy, A., 2004. Modeling of the effect of land use changes on the runoff generation
of a river basin through parameter regionalization of a watershed model. Journal of Hydrology
292(1-4), 281-295.

Jayawardena, A.W., Zhou, M.C., 2000. A modified spatial soil moisture storage capacity distribution
curve for the xinanjiang model. Journal of Hydrology 227(1-4), 93-113.
Jeremiah, E., Marshall, L., Sisson, S.A., Sharma, A., 2013. Specifying a hierarchical mixture of experts for
hydrologic modeling: Gating function variable selection. Water Resources Research 49(5),
2926-2939.

Jiao, Y. et al., 2017. Impact of vegetation dynamics on hydrological processes in a semi-arid basin by
using a land surface-hydrology coupled model. Journal of Hydrology 551, 116-131.
Jie, M.X. et al., 2018. Transferability of conceptual hydrological models across temporal resolutions:
Approach and application. Water Resources Management 32(4), 1367-1381.
Kim, S., Hong, S.J., Kang, N., Noh, H.S., Kim, H.S., 2016. A comparative study on a simple two-parameter
monthly water balance model and the kajiyama formula for monthly runoff estimation.
Hydrological Sciences Journal-Journal Des Sciences Hydrologiques 61(7), 1244-1252.





Kim, S.M., Benham, B.L., Brannan, K.M., Zeckoski, R.W., Doherty, J., 2007. Comparison of hydrologic
calibration of hspf using automatic and manual methods. Water Resources Research 43(1).
Kim, S.S.H., Hughes, J.D., Chen, J., Dutta, D., Vaze, J., 2015. Determining probability distributions of
parameter performances for time-series model calibration: A river system trial. Journal of
Hydrology 530, 361-371.
King, D.M., Perera, B.J.C., 2013. Morris method of sensitivity analysis applied to assess the importance
of input variables on urban water supply yield - a case study. Journal of Hydrology 477, 17-32.
Klemes, V., 1986. Operational testing of hydrological simulation-models. Hydrological Sciences Journal-
Journal Des Sciences Hydrologiques 31(1), 13-24.
Lan, T. et al., 2018. A clustering preprocessing framework for the subannual calibration of a hydrological
model considering climate-land surface variations. Water Resources Research 54(0).
Li, H. et al., 2018. Hybrid two-stage stochastic methods using scenario-based forecasts for reservoir refill
operations. Journal of Water Resources Planning and Management 144(12).
Li, H., Zhang, Y., 2017. Regionalising' rainfall-runoff modelling for predicting daily runoff: Comparing
gridded spatial proximity and gridded integrated similarity approaches against their lumped
counterparts. Journal of Hydrology 550, 279-293.
Li, Y., Ryu, D., Western, A.W., Wang, Q.J., 2013. Assimilation of stream discharge for flood forecasting:
The benefits of accounting for routing time lags. Water Resources Research 49(4), 1887-1900.
Li, Z. et al., 2016. Evaluation of estimation of distribution algorithm to calibrate computationally
intensive hydrologic model. Journal of Hydrologic Engineering 21(6).
Lin, K. et al., 2014. Xinanjiang model combined with curve number to simulate the effect of land use
change on environmental flow. Journal of Hydrology 519, 3142-3152.
Lu, H. et al., 2013. The streamflow estimation using the xinanjiang rainfall runoff model and dual state-
parameter estimation method. Journal of Hydrology 480, 102-114.
Luo, M., Pan, C., Zhan, C., 2019. Diagnosis of change in structural characteristics of streamflow series
based on selection of complexity measurement methods: Fenhe river basin, china. Journal of
Hydrologic Engineering 24(2).
Meng, S., Xie, X., Liang, S., 2017. Assimilation of soil moisture and streamflow observations to improve
flood forecasting with considering runoff routing lags. Journal of Hydrology 550, 568-579.
Merz, R., Parajka, J., Bloeschl, G., 2009. Scale effects in conceptual hydrological modeling. Water
Resources Research 45.
Merz, R., Parajka, J., Bloeschl, G., 2011. Time stability of catchment model parameters: Implications for
climate impact analyses. Water resources research 47(W02531).
Ming, B., Liu, P., Bai, T., Tang, R., Feng, M., 2017. Improving optimization efficiency for reservoir
operation using a search space reduction method. Water Resources Management 31(4), 1173-
1190.
Moradkhani, H., Sorooshian, S., Gupta, H.V., Houser, P.R., 2005. Dual state-parameter estimation of
hydrological models using ensemble kalman filter. Advances in Water Resources 28(2), 135-
147.
Morris, M.D., 1991. Factorial sampling plans for preliminary computational experiments. Technometrics
33(2), 161-174.
Nash, J.E., Sutcliffe, J.V., 1970. River flow forecasting through conceptual models part i — a discussion
of principles. Journal of Hydrology 10(3), 282-290.
Paik, K., Kim, J.H., Kim, H.S., Lee, D.R., 2005. A conceptual rainfall-runoff model considering seasonal



variation. Hydrological Processes 19(19), 3837-3850.
Pappenberger, F., Beven, K.J., Ratto, M., Matgen, P., 2008. Multi-method global sensitivity analysis of
flood inundation models. Advances in Water Resources 31(1), 1-14.

Pathiraja, S. et al., 2018. Time-varying parameter models for catchments with land use change: The
importance of model structure. Hydrology and Earth System Sciences 22(5), 2903-2919.

Pathiraja, S., Marshall, L., Sharma, A., Moradkhani, H., 2016. Hydrologic modeling in dynamic
catchments: A data assimilation approach. Water resources research 52(5), 3350-3372.

Poulin, A., Brissette, F., Leconte, R., Arsenault, R., Malo, J.-S., 2011. Uncertainty of hydrological
modelling in climate change impact studies in a canadian, snow-dominated river basin. Journal
of Hydrology 409(3-4), 626-636.

Quoc Quan, T., De Niel, J., Willems, P., 2018. Spatially distributed conceptual hydrological model building:
A genetic top-down approach starting from lumped models. Water Resources Research 54(10),
8064-8085.

Rebolho, C., Andreassian, V., Le Moine, N., 2018. Inundation mapping based on reach-scale effective
geometry. Hydrology and Earth System Sciences 22(11), 5967-5985.

Refsgaard, J.C., Knudsen, J., 1996. Operational validation and intercomparison of different types of
hydrological models. Water Resources Research 32(7), 2189-2202.

Seiller, G., Anctil, F., Perrin, C., 2012. Multimodel evaluation of twenty lumped hydrological models
under contrasted climate conditions. Hydrology and Earth System Sciences 16(4), 1171-1189.

Si, W., Bao, W., Gupta, H.V., 2015. Updating real-time flood forecasts via the dynamic system response
curve method. Water Resources Research 51(7), 5128-5144.

Singh, S.K., Bardossy, A., 2012. Calibration of hydrological models on hydrologically unusual events.
Advances in Water Resources 38, 81-91.

Siriwardena, L., Finlayson, B.L., McMahon, T.A., 2006. The impact of land use change on catchment
hydrology in large catchments: The comet river, central queensland, australia. Journal of
Hydrology 326(1-4), 199-214.

Sobol, I.M., 1993. Sensitivity estimates for nonlinear mathematical models. Mathematical modelling
and computational experiments 1(4), 407-414.

Sun, Y. et al., 2018. Development of multivariable dynamic system response curve method for real-time
flood forecasting correction. Water Resources Research 54(7), 4730-4749.

Teweldebrhan, A.T., Burkhart, J.F., Schuler, T.V., 2018. Parameter uncertainty analysis for an operational
hydrological model using residual-based and limits of acceptability approaches. Hydrology and
Earth System Sciences 22(9), 5021-5039.

Thirel, G. et al., 2015. Hydrology under change: An evaluation protocol to investigate how hydrological
models deal with changing catchments. Hydrological Sciences Journal-Journal Des Sciences
Hydrologiques 60(7-8), 1184-1199.

Toth, E., Brath, A., 2007. Multistep ahead streamflow forecasting: Role of calibration data in conceptual
and neural network modeling. Water Resources Research 43(11).

Westra, S., Thyer, M., Leonard, M., Kavetski, D., Lambert, M., 2014. A strategy for diagnosing and
interpreting hydrological model nonstationarity. Water Resources Research 50(6), 5090-5113.

Xie, S. et al., 2018. A progressive segmented optimization algorithm for calibrating time-variant
parameters of the snowmelt runoff model (srm). Journal of Hydrology 566, 470-483.

Xiong, L.H., Guo, S.L., 1999. A two-parameter monthly water balance model and its application. Journal
of Hydrology 216(1-2), 111-123.



Xiong, M. et al., 2019. Identifying time-varying hydrological model parameters to improve simulation
efficiency by the ensemble kalman filter: A joint assimilation of streamflow and actual
evapotranspiration. Journal of Hydrology 568, 758-768.
Xu, J., 2011. Variation in annual runoff of the wudinghe river as influenced by climate change and human
activity. Quaternary International 244(2), 230-237.
Yang, N. et al., 2017. Evaluation of the trmm multisatellite precipitation analysis and its applicability in
supporting reservoir operation and water resources management in hanjiang basin, china.
Journal of Hydrology 549, 313-325.
Yang, X. et al., 2018. A new fully distributed model of nitrate transport and removal at catchment scale.
Water Resources Research 54(8), 5856-5877.
Yin, J. et al., 2018. A copula-based analysis of projected climate changes to bivariate flood quantiles.
Journal of Hydrology 566, 23-42.
Zhao, R.J., 1992. The xinanjiang model applied in china. Journal of Hydrology 135(1-4), 371-381.





Table 1 Parameters of the TMWB model

| Parameter | Physical meaning | Range |
|---|---|---|
| C | Evapotranspiration parameter | 0.2-2.0 |
| SC | Catchment water storage capacity | 100-2000 |





Table 2 Parameters of the Xinanjiang model

| Category | Parameter | Physical meaning | Range |
|---|---|---|---|
| Evapotranspiration | WM | Tension water capacity | 80-400 |
| | X | WUM=X×WM, WUM is the tension water capacity of lower layer | 0.01-0.8 |
| | Y | WLM=Y×WM, WLM is the tension water capacity of deeper layer | 0.01-0.8 |
| | K | Ratio of potential evapotranspiration to pan evaporation | 0.4-1.5 |
| | C | The coefficient of deep evapotranspiration | 0.01-0.4 |
| Runoff production | B | The exponent of the tension water capacity curve | 0.1-10 |
| | IMP | The ratio of the impervious to the total area of the basin | 0.01-0.15 |
| Runoff separation | SM | The areal mean of the free water capacity of the surface soil layer | 10-80 |
| | EX | The exponent of the free water capacity curve | 0.6-6 |
| | KG | The outflow coefficients of the free water storage to groundwater | 0.01-0.45 |
| | KSS | The outflow coefficients of the free water storage to interflow | 0.01-0.45 |
| Flow concentration | N | Number of reservoirs in the instantaneous unit hydrograph | 0.5-10 |
| | NK | Common storage coefficient in the instantaneous unit hydrograph | 1-20 |
| | KKG | The recession constant of groundwater storage | 0.6-1 |
| | KKSS | The recession constant of the lower interflow storage | 0.9-1 |



Table 3 Different cases of synthetic experiments and real catchment case studies for comparison and evaluation

| | Data | Hydrological model | Time-varying parameter estimation methods | | |
|---|---|---|---|---|---|
| | | | SSC | SSC-DP | Data assimilation |
| Synthetic experiment | Monthly synthetic data | TMWB model | | ✓ | ✓ |
| | Hourly synthetic data | Xinanjiang model | ✓ | ✓ | ✓ |
| Real catchment case study | Daily data from Wuding River basin | Xinanjiang model | ✓ | ✓ | ✓ |
| | Daily data from Xun River basin | Xinanjiang model | ✓ | ✓ | ✓ |



Table 4 True parameters of different scenarios in the synthetic experiment with the TMWB model

| Scenario | Description |
|---|---|
| 1 | Both $C$ and $SC$ are constant |
| 2 | Both $C$ and $SC$ have increasing linear trends and change every month |
| 3 | Both $C$ and $SC$ have periodic variations and change every month |
| 4 | Both $C$ and $SC$ have increasing linear trends and change every six months |
| 5 | Both $C$ and $SC$ have periodic variations and change every six months |
| 6 | Both $C$ and $SC$ have increasing linear trends and change every year |
| 7 | Both $C$ and $SC$ have periodic variations and change every year |
| 8 | $C$ has a periodic variation with an increasing linear trend, whereas $SC$ only has an increasing linear trend. The parameters change every year |



Table 5 True parameters of different scenarios in the synthetic experiment with the Xinanjiang model

| Scenario | Description |
|---|---|
| 1 | *KE*, *CI*, *CG*, *KI*, *KG*, and *NK* remain constant |
| 2 | *KE*, *CI*, *CG*, *KI*, *KG*, and *NK* have linear trends and change every year |
| 3 | *KE*, *CI*, *CG*, *KI*, *KG*, and *NK* have periodic variations and change every year |
| 4 | *KE* has a periodic variation with an increasing linear trend, whereas *CI*, *CG*, *KI*, *KG*, and *NK* only have periodic variations. The parameters change every year |


Table 6 Simulation performance for streamflow in the synthetic experiment with the TMWB model

|  | scenario1 | scenario2 | scenario3 | scenario4 | scenario5 | scenario6 | scenario7 | scenario8 |
|---|---|---|---|---|---|---|---|---|
| ENKF | 99.71% | 99.83% | 99.84% | 99.72% | 99.67% | 99.67% | 99.66% | 99.73% |
| 3-SSC-DP | 99.88% | 99.78% | 99.44% | 99.79% | 99.96% | 99.95% | 99.63% | 99.94% |
| 6-SSC-DP | 99.88% | 99.94% | 98.11% | 99.94% | 99.93% | 99.92% | 99.93% | 99.94% |
| 12-SSC-DP | 99.86% | 99.91% | 94.51% | 99.91% | 99.38% | 99.92% | 99.91% | 99.94% |





Table 7 Comparison among EnKF, SSC-EnKF, and EnKS in the synthetic experiment
with the Xinanjiang model

| | | Scenario 1 (trend) | | | Scenario 3 (combination) | | | Scenario 4 (constant) | | |
|---|---|---|---|---|---|---|---|---|---|---|
| | | EnKF | SSC-EnKF | EnKS | EnKF | SSC-EnKF | EnKS | EnKF | SSC-EnKF | EnKS |
| RMSE | KE | 0.097 | 0.058 | 0.068 | 0.135 | 0.071 | 0.127 | 0.112 | 0.124 | 0.107 |
| | CI | 0.065 | 0.012 | 0.031 | 0.051 | 0.01 | 0.044 | 0.035 | 0.674 | 0.035 |
| | CG | 0.093 | 0.015 | 0.036 | 0.061 | 0.013 | 0.056 | 0.026 | 0.22 | 0.039 |
| | KI | 0.141 | 0.004 | 0.023 | 0.094 | 0.004 | 0.083 | 0.06 | 0.057 | 0.065 |
| | KG | 0.012 | 0.002 | 0.017 | 0.015 | 0.001 | 0.016 | 0.011 | 0.009 | 0.011 |
| | NK | 2.273 | 0.249 | 2.459 | 3.241 | 0.279 | 2.084 | 1.502 | 0.195 | 1.978 |
| Mean RMSE | | 0.300 | 0.053 | 0.156 | 0.256 | 0.050 | 0.215 | 0.236 | 0.138 | 0.271 |
| $R^2$ | KE | 57.60% | 66.00% | 73.30% | 25.70% | 50.30% | 26.40% | | | |
| | CI | 86.70% | 97.90% | 54.10% | 71.40% | 98.10% | 82.50% | | | |
| | CG | -18.80% | 96.40% | 91.10% | 60.70% | 96.40% | 88.20% | | | |
| | KI | 72.40% | 99.50% | 89.30% | 38.30% | 99.50% | 27.30% | | | |
| | KG | 97.20% | 98.60% | 97.20% | 96.70% | 98.30% | 97.60% | | | |
| | NK | 82.30% | 98.90% | 84.30% | 79.40% | 98.50% | 69.80% | | | |
| Mean $R^2$ | | 62.90% | 92.88% | 81.55% | 62.03% | 90.18% | 65.30% | | | |

*Note*. The mean RMSE is the average value of the normalized RMSE so that the
identification results for the parameters with different ranges can be summarized. Mean
$R^2$ is calculated in the same manner.



Table 8 Simulation performance for streamflow in the synthetic experiment with the Xinanjiang model

|  | Scenario1 | Scenario2 | Scenario3 | Scenario4 |
|---|---|---|---|---|
| SSC-EnKF | 99.72% | 99.80% | 99.72% | 99.78% |
| 12-SSC | 99.93% | 99.74% | 99.75% | 99.72% |
| 12-SSC-DP | 99.92% | 99.74% | 99.61% | 99.73% |
| 24-SSC-DP | 95.66% | 95.98% | 94.89% | 95.47% |



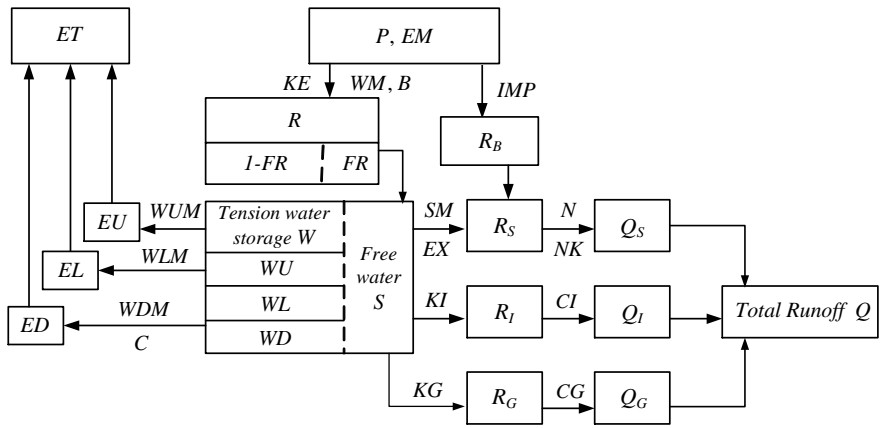

Figure 1 Flowchart of the Xinanjiang model.





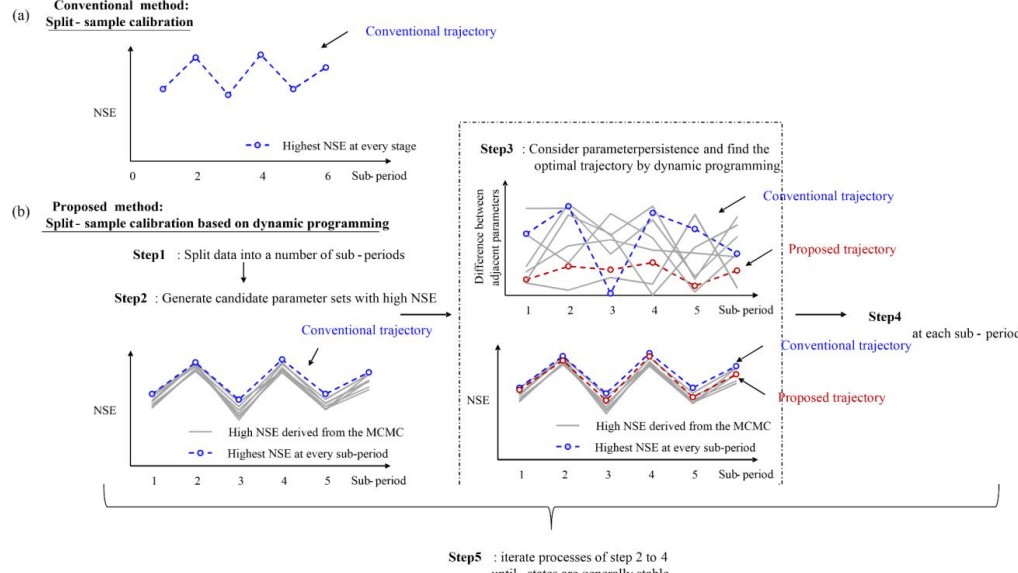

Figure 2 Flowchart of SSC-DP.

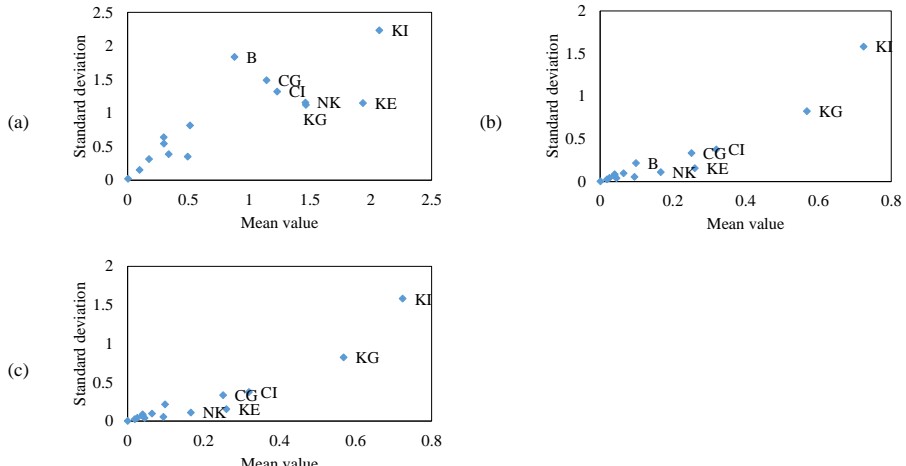

Figure 3 Results of the Morris method for the synthetic experiment with the
Xinanjiang model. The sensitivity analysis is based on three different kinds of model
responses: (a) NSE; (b) $NSE_{abs}$; (c) $NSE_{ln}$. Only the most sensitive parameters are
labeled.



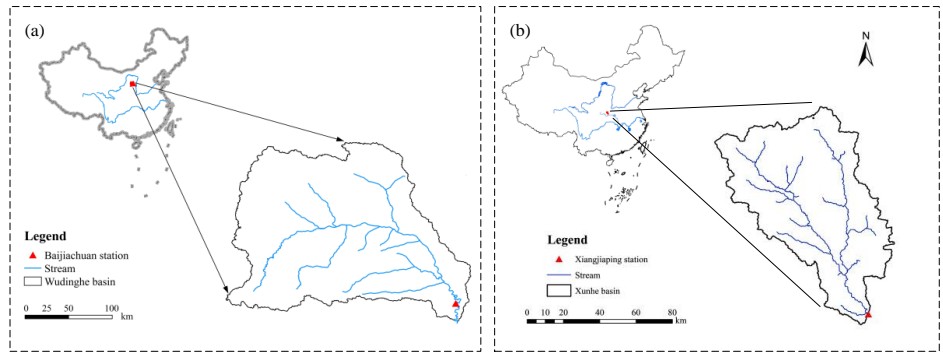

Figure 4 Location of (a) Wuding River basin and (b) Xun River basin.



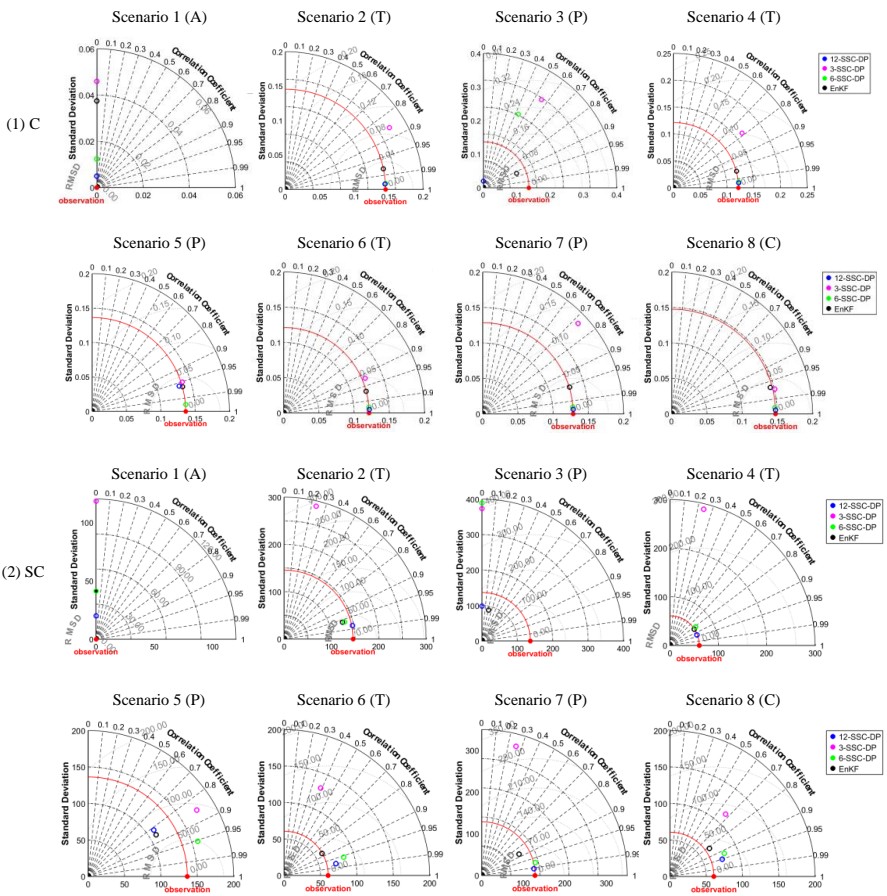

Figure 5 Taylor diagram showing a comparison between the EnKF and SSC-DP methods for parameter identification in the synthetic experiment with the TMWB model. Note the radial distance, linear distance from the observation, and angle represent the standard deviation, RMSE, and $R^2$, respectively. A = Constant; T = trend; P = Periodicity; C = Combination.





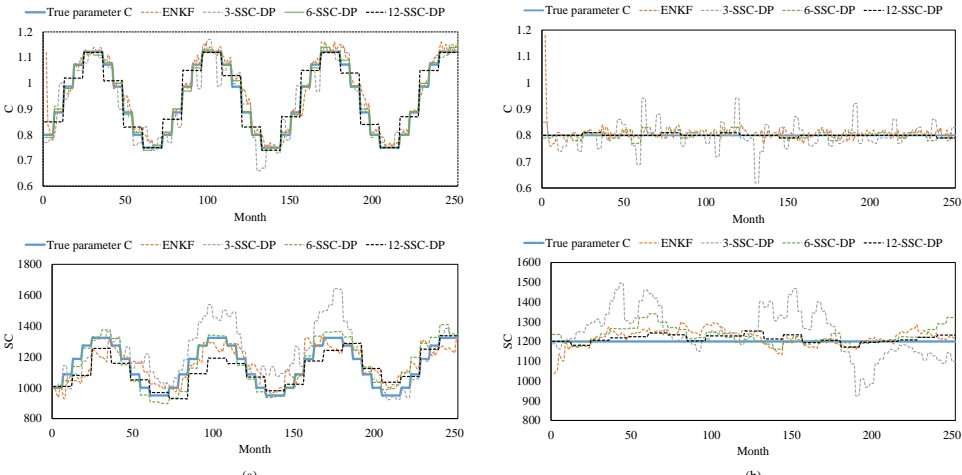

Figure 6 Comparison among different methods for (a) scenario 5 and (b) scenario 1 of

the synthetic experiment with the TMWB model.

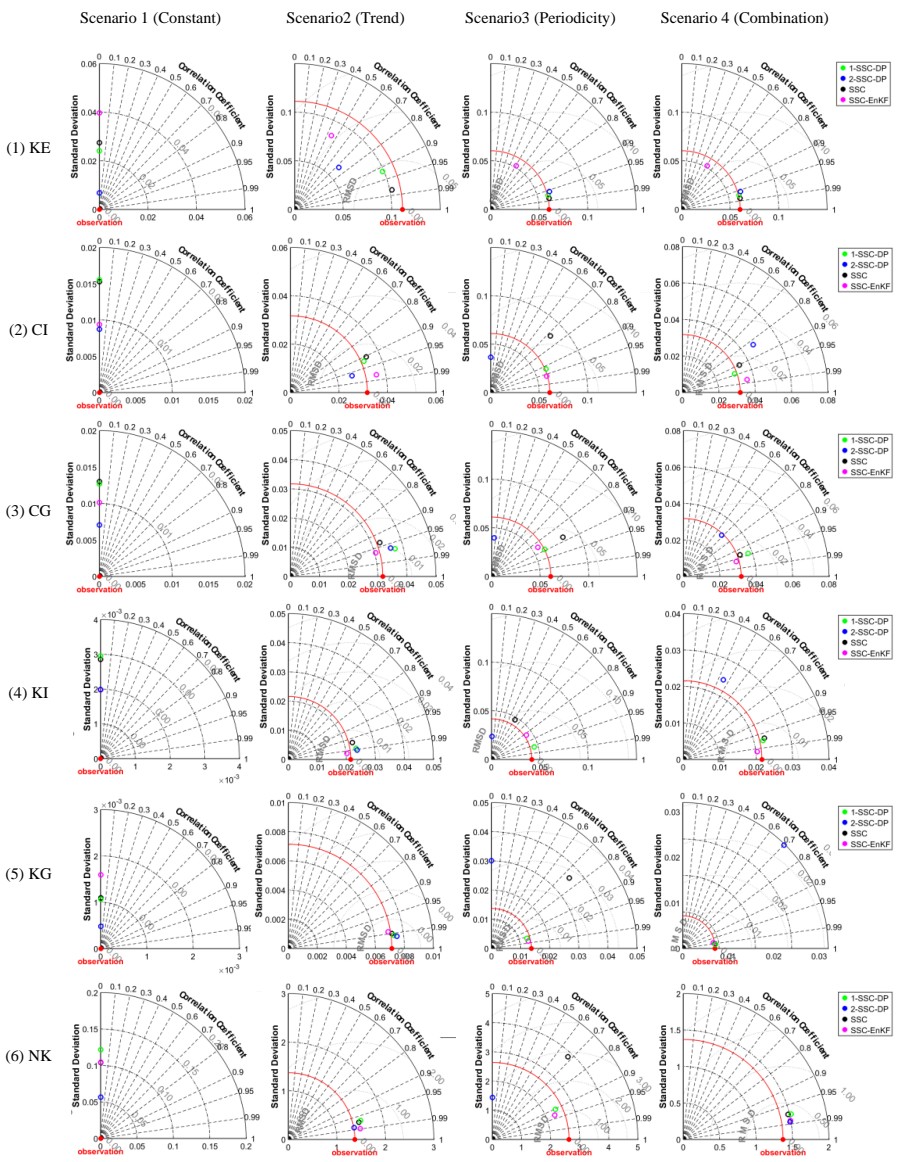

Figure 7 Taylor diagram showing a comparison between the SSC, SSC-EnKF, and
SSC-DP methods for parameter identification in the synthetic experiment with the
Xinanjiang model. The radial distance, linear distance from the observation, and angle
represent the standard deviation, RMSE, and $R^2$, respectively.





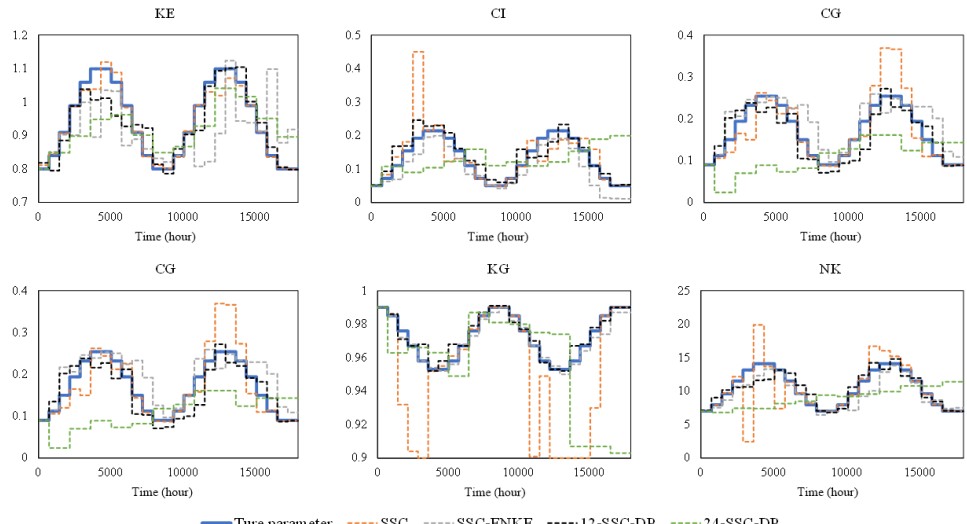

Figure 8 Comparison between estimated parameters and their true values for scenario

3 of the synthetic experiment with the Xinanjiang model.

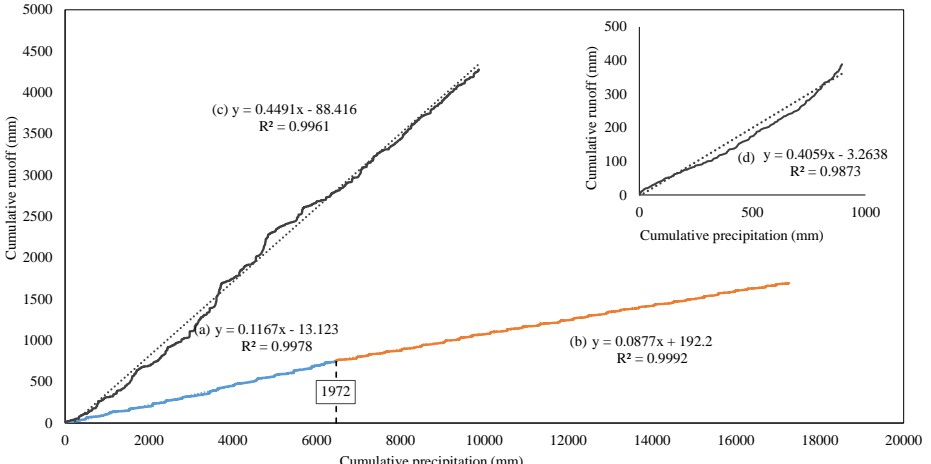

Figure 9 Double mass curves between daily runoff and precipitation for (a) Wuding
River basin from 1958–1972; (b) Wuding River basin from 1973–2000; (c) Xun River
basin from 1991–2001. Subgraph (d) represents the double mass curve between the
mean daily runoff and precipitation from 1991–2001.

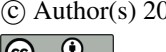



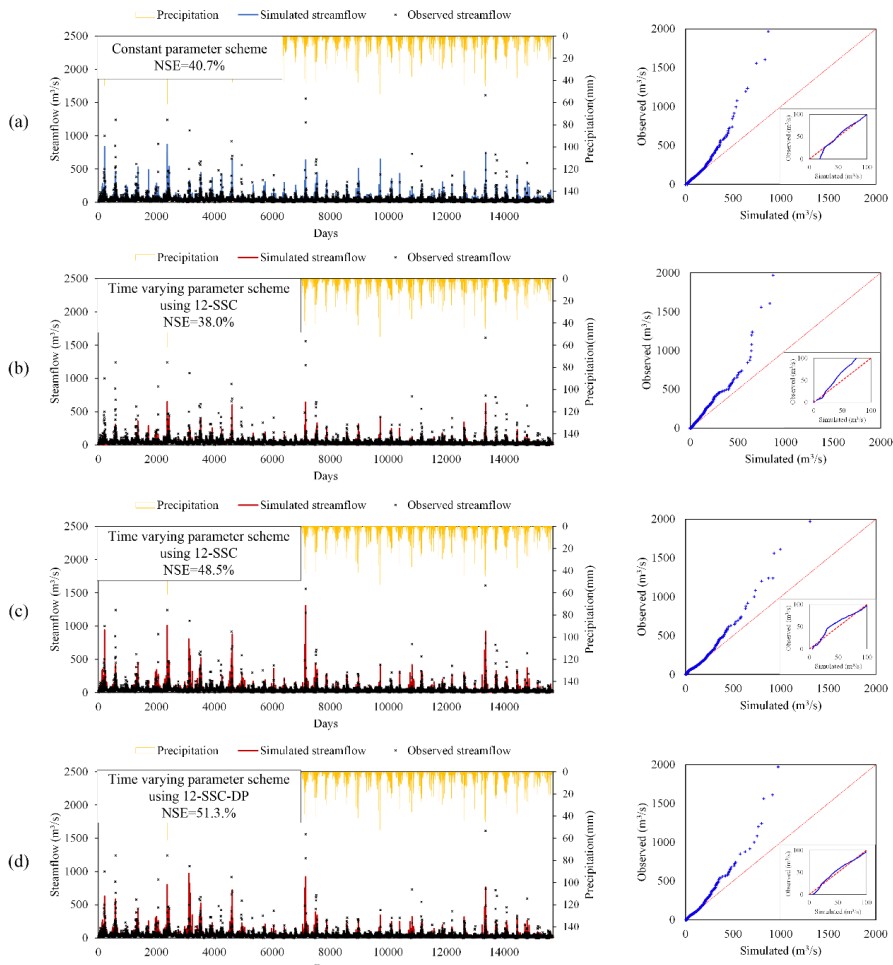

Figure 10 Streamflow simulation hydrographs (left panels) and quantile-quantile plots (right panels) using (a) conventional method, (b) 12-SSC-EnKF, (c) SSC, and (d) SSC-DP for the Wuding River basin. The right subgraphs represent the quantile-quantile plots for streamflow lower than 100 m³/s.



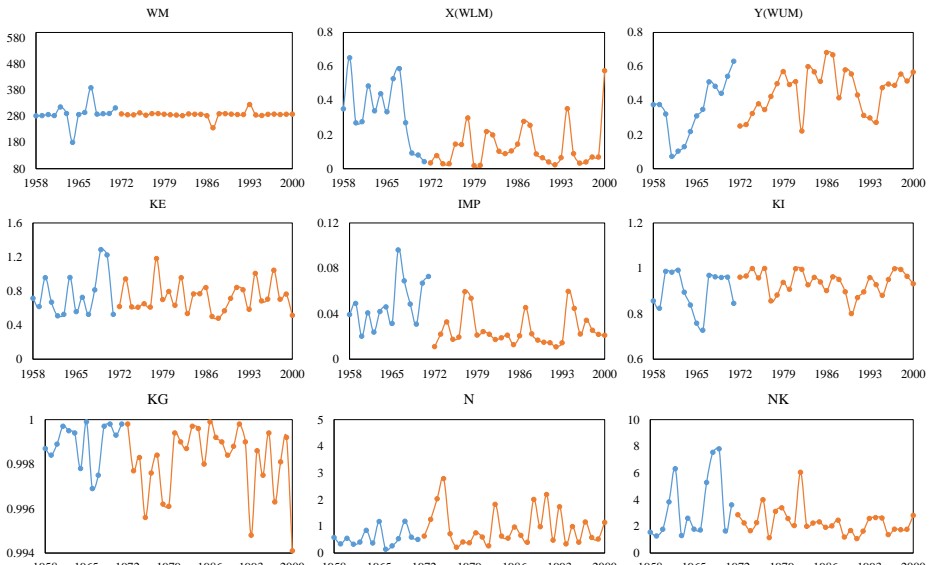

Figure 11 Estimated sensitive parameters of the Xinanjiang model for the Wuding River basin. The blue and orange solid lines represent the estimated parameters pre- and post-1972, respectively.



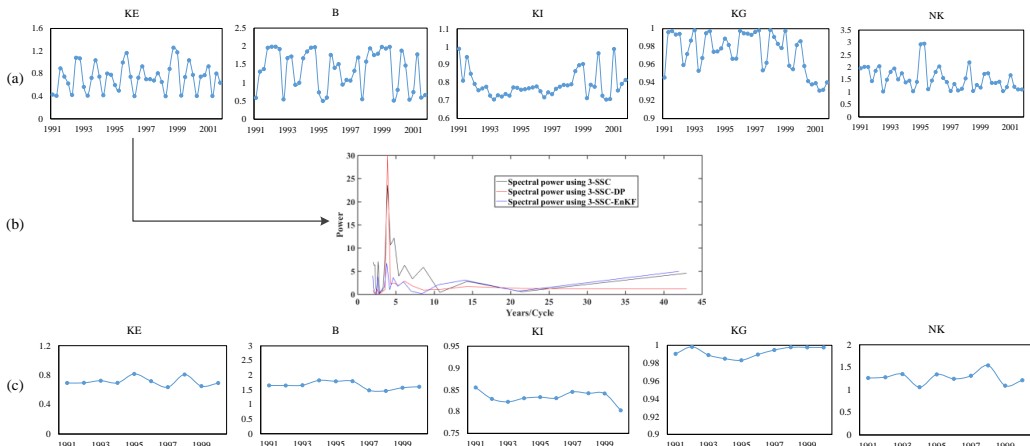

Figure 12 Estimated sensitive parameters of the Xinanjiang model for the Xun River basin over (a) seasonal time scale and (c) annual time scale. Plot (b) illustrates the spectral power of parameter KE using different methods.





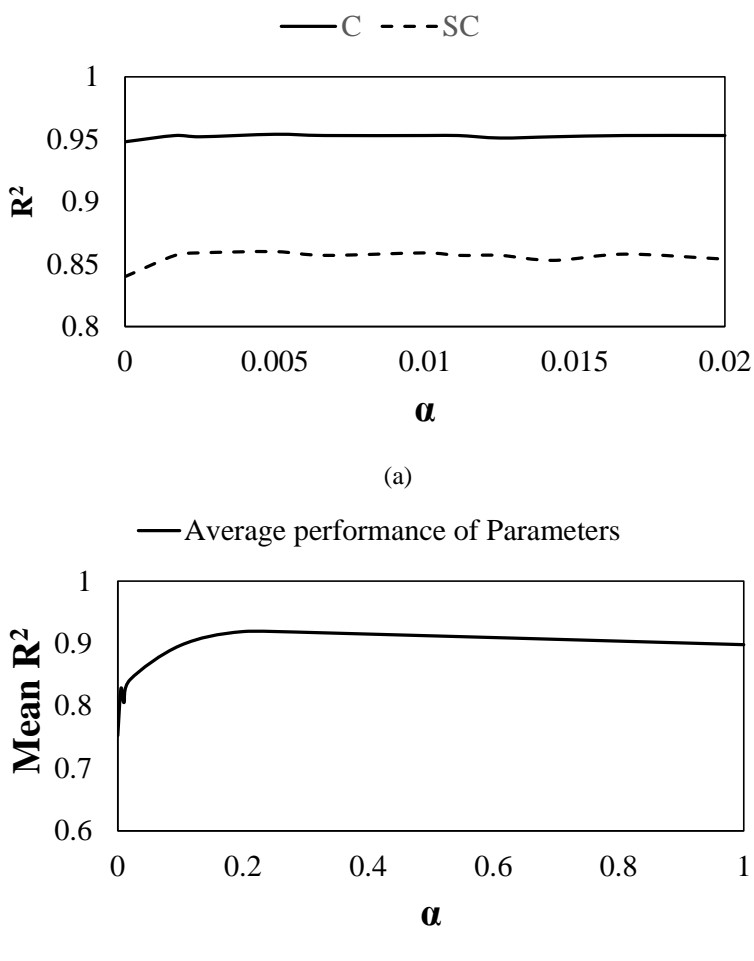

(a)

(b)

Figure 13 Correlation efficiency results of SSC-DP using different weights of parameter continuity for synthetic experiments with (a) TMWB model and (b) Xinanjiang model. The mean $R^2$ is the average value of the $R^2$ such that the identification results for parameters with different ranges can be summarized.