# Peer review of "A time-varying parameter estimation approach using split-sample calibration based on dynamic programming"

_Hydrology and Earth System Sciences, 2019_

## Referee Comment (RC1) · Anonymous Referee #1 · 14 Feb 2020

General comments: The main objective of this paper is to propose a new method to estimate time-varying parameters. The study is interesting, by considering the combination of the basic concept of split-sample calibration (SSC) and the parameter continuity assumption. And dynamic programming is used to determine the optimal parameter trajectory. Two synthetic experiments were designed to evaluate its applicability and efficiency for time-varying parameter identification. However, the assumption that the response of individual parameter variations to changes in the climatic conditions should be further discussed. Therefore, I think the manuscript requires major revision before publication. I will detail my process here through general comments that the author could use to rework the paper in order to improve it. Special comments: 1. This paper

presents methods to estimate the time-varying parameter based on dynamic programming. The authors attempt to combine multiple methods including SSC and ENKF. However, the highlight of this paper is no very clear, which should be refined. 2. The fundamental assumption that the individual parameters may not response to the catchment dynamics due to the linear or nonlinear correlations between parameters (Bardossy, 2007). The effects of identifiability of parameters on this research are suggested to be investigated. 3. The non-stationary change in catchment characteristics may not be predicted. Lots of uncertainty factors would prevent the estimation of future scenarios in catchments. 4. How to generally estimate the stable period, such as decades, years or months, considering catchment characteristics? It is vital for the method in this study. The impact of sub-period lengths on the performance of SSC-DP is significant. 5. The two lumped models were chosen in this study. The number of parameters is different. The sensitivity analysis was further performed to reduce the dimension of parameters in the Xinanjiang model. Hence, the purpose of choosing two different lumped models should be discussed. 6. The titles cannot show the logic framework of the research. The flowchart is suggested to used to illustrate the framework in this study. The introduction of the manuscript is suggested to present in the appendix. 7. The sensitive hydrograph phases of model performance criteria, i.e., RMSE, R2 and NSE are peaks and discharge dynamics, flood peak, and discharge dynamics (Pfannerstill et al., 2014). Three metrics have strong correlations. The results as shown in Figure 5 needs furthermore discussion. 8. The streamflow, climate and underlying surface conditions in the two study areas were not analyzed in this study. However, it is critical to the estimation of time-varying parameters. 9. In lines 175-176, the assumption that the continuity condition aims to minimize the difference between the estimated parameters for sub-periods i and i+1 unreasonable. The differences between two consecutive sub-periods represent the time-varying changes of the catchment. The continuity conditions for enhancing the model performance should focus on the model structure, such as state variables. 10. Minor comment. The resolution of Figure 5 is low and information is not presented. References: Bardossy, A., 2007. Calibration of

hydrological model parameters for ungauged catchments. Hydrol Earth Syst Sc, 11(2): 703-710. DOI:DOI 10.5194/hess-11-703-2007 Pfannerstill, M., Guse, B., Fohrer, N., 2014. Smart low flow signature metrics for an improved overall performance evaluation of hydrological models. J Hydrol, 510: 447-458. DOI:10.1016/j.jhydrol.2013.12.044

---

## Referee Comment (RC2) · Anonymous Referee #2 · 30 Aug 2020

**1. OVERALL RECOMMENDATION**

The manuscript addresses the important topic of calibration of rainfall-runoff model parameters, and presents results obtained on two different catchments with two models. Even if the introduction includes relevant references and the methods are well presented, the paper lacks important discussions on the rainfall-runoff model performances, observed time series quality, attribution of observed/simulated changes, consideration of only two catchments, and several obtained results are over-interpreted. Finally, several figures and tables must be significantly improved. Therefore, I think the manuscript requires major revision before publication.

[Figure]

**2. GENERAL COMMENTS**

The tables 6 to 8 might be presented as figures to be more easily interpreted. Figure 5, 7 and 10 are very difficult to read, and must be significantly improved.

Line 28 to 29: several studies highlighted the difficulty of conceptual rainfall-runoff models in the context of climate change impact studies.

Line 35 to 36: the terms "constants" and "stable" must be defined: constant/stable in space and/or in time?

Line 43 to 44: in this context, it may be needed to define what is called "climate conditions".

Line 122: the terms "behavioural" must be clearly defined or not used in this context.

Line 137, line 150 and Table 1 and 2: please presents parameter units.

Section 2.2: the need to reduce the number of Xinanjian free parameters using a sensitivity analysis must be investigated more deeply in the paper. In the current version of the paper, this model is considered with different number of free parameters depending on the modeling experiments. Why not calibrating the 15 free parameters of this rainfall-runoff model for all experiments?

Section 2.3.1: one of the main hypotheses of this paper is the important "fluctuations" of the model parameter values over adjacent sub-periods, hypothesis that is not justified by the literature review, and that is not illustrated with the obtained results. This point must be discussed more deeply in the paper.

Evaluation criteria: Why only use the NSE criterion as only evaluation criteria, and no other criteria, such as KGE and its components? NSE appears to be non-discriminating between considered calibration methods. Using other calibration criteria - looking at different time step and/or different error characteristics such as bias on the highest streamflow values – might be interesting in this context.

Section 3.2 (Wuding river basin), lines 364 to 369: the changes of the studied catchment characteristics seem to be decisive for the interpretation of the results obtained on this watershed. Nevertheless, no quantitative results / analysis of these changes are given in the paper: what is the percentage of the catchment that has been afforested? What are the number and the capacity of the built reservoirs? When are they built? Finally, an important point not discussed in the paper is the stationary and the quality of the precipitation and streamflow time series studied and used for the model calibration. This point is crucial in this context and need to be discussed.

Section 3.3 (Xun River basin): same remarks as the Wuding river basin: what about potential changes on this basin? Are precipitation and streamflow time series of good quality?

Section 4.1: the seasonal signal of the parameter values (cf. figures 6 and 8) must be more significantly discussed in the paper.

Line 456 to 458: this conclusion must be significantly moderated: the "SSC-DP" calibration method is by definition better to select more continuous parameter values.

Section 4.2: this data analysis is crucial in this context. It might be relevant to present it in the data section. Moreover, this analysis must be significantly improved: what about potential errors (random or systematic) in the observed precipitation and streamflow series? What about potential break in the streamflow series due to rating curve changes? What is the statistical significance of this analysis? The analysis of only one catchment requires to look carefully the studied time series in the context of attribution of changes. The relative bad performance of the rainfall-runoff model on this catchment (NSE=0.41) must be discussed. In particular, the systematic streamflow underestimation for the different calibration methods must be discussed.

Line 492 to 495: the "unreasonable model states" between sub-periods might be illustrated in the paper.

Line 500 to 501: this conclusion must be moderated, since results have been obtained on one catchment only.

Line 511 to 514: this attribution analysis must be moderated (see previous remarks on attribution analysis).

Line 520 to 522: again, what about potential error in the rating curve in this context?

Line 526 to 539: why not presenting a Figure such as Figure 10 to illustrate rainfall-runoff simulations on this catchment?

Line 531 to 532: is this out-performance significant?

Line 534 to 539 and line 637 to 648: again, these attribution conclusions must be moderated, because they are drawn from only two basins, without any investigation of potential systematic errors in the observed time series.

―――――――――――――――――

---

## Author Comment (AC1) · 26 Sep 2020

Responses to Reviewer #1:

1. This paper presents methods to estimate the time-varying parameter based on dynamic programming. The authors attempt to combine multiple methods including SSC and ENKF. However, the highlight of this paper is no very clear, which should be refined. Reply: Thank you for reviewing our manuscript and for the professional comments. The highlights of this paper are refined as follows: 1. The proposed method combines split-sample calibration (SSC) and ensemble Kalman filter (EnKF) for time-varying parameter estimation. Compared to SSC, the proposed method can find a more continuous

parameter trajectory; compared to EnKF, the proposed method allows parameters to retain stable for a pre-determined period, instead of varying at every time-step. 2. The effectiveness of the proposed method is validated with two hydrological models and two real catchment case studies of different conditions. 3. For the case study of Xun River basin, the proposed method detects the strongest seasonal signal. The highlights are elaborated on in the abstract as follows: Although the parameters of hydrological models are usually regarded as constant, temporal variations can occur in a changing environment. Thus, effectively estimating time-varying parameters becomes a significant challenge. Two methods, include split-sample calibration (SSC) and Data assimilation, have been used to estimate time-varying parameters. However, SSC is unable to consider the parameter temporal continuity, while Data assimilation assumes parameters vary at every time-step. This study proposed a new method that combines (1) the basic concept of SSC, whereby parameters are assumed to be stable for one sub-period, and (2) the parameter continuity assumption, i.e., the differences between parameters in consecutive time steps are small. (Pages 2, Lines 3-7) The highlights are also elaborated in the conclusions as follows: 1. The proposed method with a suitable length not only produces better simulation performance, but also ensures more accurate parameter estimates than SSC and EnKF in the synthetic experiment using the TMWB model with two parameters. The impact of sub-period lengths on the performance of SSC-DP is significant when the known parameters vary sinusoidally. 2. The proposed method can be used to deal with complex hydrological models involving a large number of parameters, demonstrated by the synthetic experiment using the Xinanjiang model with 15 parameters. A sensitivity analysis was performed to reduce the probable computational cost and improve the efficiency of identifying the time-varying parameters. 3. The proposed method has the potential to detect the relationship between the time-varying parameters and dynamic catchment characteristics. For example, SSC-DP produced the best simulation performance in the case study of the Wuding River basin and detects that parameters reflecting soil water capacity and impervious areas changed significantly after 1972, reflecting the soil and water

conservation projects carried out from 1958–2000. Additionally, SSC-DP detects the strongest seasonal signal in the case study of Xun River basin, indicating the distinct impacts of seasonal climate variability. (Pages 33, Lines 925-941)

2. The fundamental assumption that the individual parameters may not response to the catchment dynamics due to the linear or nonlinear correlations between parameters (Bardossy, 2007). The effects of identifiability of parameters on this research are suggested to be investigated. Reply: We agree with the reviewer that the hydrological model parameters should be treated as parameter vectors instead of independent individual values (Bardossy 2007). The identifiability of parameters is considered in this study: (1) Parameters are not treated as individuals, but multiple parameters are identified simultaneously. For the two-parameter monthly water balance (TMWB) model, parameters C and SC are estimated simultaneously. While for the Xinanjiang model, the sensitive parameters are calibrated at the same time. (2) By generating a large number of parameter sets as candidates in each sub-period, the proposed method takes into account the parameter equifinality, while the traditional SSC method only takes the optimal parameter set.

3. The non-stationary change in catchment characteristics may not be predicted. Lots of uncertainty factors would prevent the estimation of future scenarios in catchments. Reply: This study focuses on methods to identify time-varying parameters, and the future research is considered to relate time-varying parameters and available information, such as number of dams and population. Then the time-varying parameters' function can be derived to predict future streamflow under the changing environment.

4. How to generally estimate the stable period, such as decades, years or months, considering catchment characteristics? It is vital for the method in this study. The impact of sub-period lengths on the performance of SSC-DP is significant. Reply: Determination of the stable period considers 3 factors: 1. Temporal scale of climate change or human activities. The Wudinghe River basin is taken as a case study. Since 1960s, the soil and water conservation measures were carried out in this basin to reduce the highly

erodible loess, such as tree plantation, reservoir construction and land terracing. The human activities lead to a durative and long-term change in the catchment characteristic. Hence, the yearly sub-period is considered. 2. Seasonality. The Xun River basin is taken as a case study. Contrary to the Wudinghe River basin, the relationship between precipitation and runoff of the Xun River basin is rarely affected by human activities during 1991-2001. However, its significant seasonal dynamics can be observed and has been studied in literature (Lan et al. 2020, Lan et al. 2018). In order to diagnose the seasonality, the stable period of 3-month is considered. 3. The simulation accuracy. The length should not be too long to capture the variations in physical processes, while it should be long enough to reduce the uncertainty of calibration. Based on the results of the synthetic experiments, it is suggested that the length should be as long as possible without degrading the simulation performance significantly. For example, in the synthetic experiment with the TMWB model, if the difference between the NSE values of 6-SSC-DP and 3-SSC-DP is small, the preferred length is six months. The determination of the sub-period length has been described in discussion as follows: It is suggested that the determination of the sub-period length considers three factors: (1) The temporal scale of climate change or human activities. For example, the Wudinghe River basin is taken as a case study. The soil and water conservation measures lead to a durative and long-term change in the catchment characteristic since 1960s. Due to this, the yearly sub-period is preferred. (2) The seasonality. Contrary to the Wudinghe River basin, the relationship between precipitation and runoff of the Xun River basin is rarely affected by human activities during 1991-2001. However, its significant seasonal dynamics can be observed and has been studied in literature (Lan et al. 2020, Lan et al. 2018). In order to diagnose the seasonality, the stable period of 3-month is adopted. (3) The simulation accuracy. The length should be neither too long nor too short so as to increase the reliability of the calibration while guaranteeing that variations in real processes are captured. Thus, given that the time scale of the variations is unknown, the proposed SSC-DP can be used with different split-sample lengths. It is suggested that the length should be as long as possible without degrading the simulation performance significantly. For example, in the synthetic experiment with the TMWB model, if the difference between the NSE values of 6-SSC-DP and 3-SSC-DP is small, the preferred length is 6-month. (Page 31~32, Lines 881~905)

5. The two lumped models were chosen in this study. The number of parameters is different. The sensitivity analysis was further performed to reduce the dimension of parameters in the Xinanjiang model. Hence, the purpose of choosing two different lumped models should be discussed. Reply: Two lumped models are chosen to evaluate the applicability of the proposed method to hydrological models with different number of parameters. Furthermore, the parameters of the TMWB model have been identified by EnKF in the work of Deng et al. (2016), but the parameters of the Xinanjiang model are scarcely recognized as time-variant. Hence, the use of the TMWB model is benefit for comparison. The purpose of choosing two different lumped models has been added as follows: There are two important differences between the TMWB and Xinanjiang models: (1) the TMWB model has two parameters, while the Xinanjiang model has fifteen parameters; (2) TMWB is a monthly rainfall-runoff model, whereas the Xinanjiang model can run on hourly or daily step sizes. (Page 9, Lines 178~181)

6. The titles cannot show the logic framework of the research. The flowchart is suggested to be used to illustrate the framework in this study. The introduction of the manuscript is suggested to present in the appendix. Reply: To avoid confusion, the title of the Section 3, i.e., "Data and study area", is replaced by "Synthetic experiment and real catchment case study". A flowchart describing the framework of the research is added in Fig.1. The introduction of the methodologies is presented as follows: In this section, a SSC-DP method is proposed to identify the time-varying parameters of hydrological models. The two hydrological models considered in this study are the TMWB and Xinanjiang models. Their concepts and differences are presented in Sect. 2.1. A sensitivity analysis is employed to focus efforts on parameters important to calibration and avoid prohibitive computational cost, as outlined in Sect. 2.2. Three time-varying parameter estimation methods (SSC, SSC-DP, and data assimilation) are presented

in Sect. 2.3. The SSC and data assimilation are provided for comparisons with the SSC-DP. Finally, to evaluate the performance of the time-varying parameter estimation methods, six evaluation criteria are selected and formulated in Sect. 2.4. The flowchart of the methodologies is shown in Fig. 1. (Pages 7, Lines 136-145)

7. The sensitive hydrograph phases of model performance criteria, i.e., RMSE, R2 and NSE are peaks and discharge dynamics, flood peak, and discharge dynamics (Pfannerstill et al., 2014). Three metrics have strong correlations. The results as shown in Figure 5 needs furthermore discussion. Reply: Thanks for the comment. This comment involves three aspects: (1) Three metrics are used to evaluate the streamflow simulations. NSE coefficient, and two evaluation metrics have been added: relative error (RE) and the NSE on logarithm of streamflow (NSEln). In the revised paper, these evaluation metrics are described as follows: The streamflow simulations given by the proposed method are verified using the NSE, relative error (RE) and NSE on logarithm of streamflow (NSEln) (Hock, 1999). RE evaluates the error of the total volume of streamflow, while NSE and NSEln evaluate the agreement between the hydrograph of observations and simulations. NSE is more sensitive to high flows, but NSEln focuses more on low flows. Higher values of NSE, NSEln and lower values of RE indicate better streamflow simulations. To see the equations of NSE, RE and NSEln, please refer to the supplement.(Pages 15~16, Lines 324-333) Description of the evaluation results has been added in Revised Manuscript as follows: ïČŸ For results of the synthetic experiment with the TMWB model Figure 6(a) presents the runoff simulation performance for various scenarios. In scenario 1, the NSE values of the three SSC-DP methods are all higher than that of EnKF. The results of NSEln show no significant differences among various methods. For scenarios 2, 4, and 6, where true parameters have linear trends, the 6-SSC-DP and 12-SSC-DP are superior to the EnKF and 3-SSC-DP in terms of NSE and NSEln. In scenario3, where the true parameters have periodic variations and change every month, the NSE and NSEln values of 6-SSC-DP and 12-SSC-DP decrease significantly, because the assumed sub-period length is longer than the time-scale of actual variations. Similarly, in scenario 5, 12-SSC-DP performs

worst for NSE and NSEln, but 6-SSC-DP performs best. In scenario 7 and 8, both 6-SSC-DP and 12-SSC-DP perform better than EnKF. According to the evaluations of NSE and NSEln, the SSC-DP offers improved accuracy than the EnKF if the proper length is chosen. Another advantage of the SSC-DP is the low RE. For all scenarios, the SSC-DP methods significantly outperform for RE compared with EnKF. Among the SSC-DP methods, the RE of 3-SSC-DP is the smallest. (Page 22, Lines 492~506) ïČŸ For results of the synthetic experiment with the Xinanjiang model The simulated streamflow and identification of time-varying parameters was compared across four methods: 1-SSC, SSC-EnKF, 1-SSC-DP, and 2-SSC-DP. The simulation performance is summarized in Figure 9(a). For all scenarios, the NSE of 2-SSC-DP is the lowest, but it performs better for low flows. The SSC-EnKF produces the highest RE in scenarios 2, 3 and 4, indicating the problem of simulating water balance. The SSC and 1-SSC-DP perform well for all scenarios in terms of NSE, RE and NSEln. Wherein, the SSC performs better than the 1-SSC-DP with regard to RE, while 1-SSC-DP is slightly superior to SSC in scenario 3 with higher NSEln. (Page 24, Lines 560~566) ïČŸ For results of case study in Wuding River basin The simulation performance is presented in Figure 12. The values of the NSEs are relatively low, it is because the streamflow in dry regions is difficult to simulate. It can be seen that the 12-SSC-DP gives the best simulation results among different methods with the highest NSE, NSEln and low RE. Although the 12-SSC produces relatively high NSE, but it performs worst simulations for low flows. The SSC-EnKF has relative high NSEln, but the RE of it is the largest. Overall, the 12-SSC-DP significantly improve the simulation performance of the Xinanjiang model in the Wuding River basin. (Page 26, Lines 706~713) ïČŸ For results of case study in Xun River basin The simulation performance is presented in Figure 15. All methods performed well, with NSE values of 92.5 %, 93.0 %, 95.0 %, and 94.8 % for the conventional method, 3-SSC-EnKF, 3-SSC, and 3-SSC-DP, respectively. 3-SSC and 3-SSC-DP also perform well for NSEln compared with 3-SSC-EnKF and the conventional method. However, as regards to RE, the values are 0.0007 and 0.0324 for 3-SSC-DP and 3-SSC-DP, respectively. It indicated that the 3-SSC-DP can better

simulate water balance than the 3-SSC in the Xun River basin. (Page 28~29, Lines 785~806) (2) Three metrics are used to evaluate the parameter estimations. The estimated parameters are evaluated by the RMSE (Alvisi et al., 2006), MARE (Khalil et al., 2001) and R2 (Kim et al., 2007). RMSE and MARE quantifies the accuracy of the estimated parameters, but RMSE is more sensitive to high values than MARE. R2 records the overall agreement between the true and estimated parameters. Smaller values of RMSE, MARE and higher values of R2 indicate stronger parameter identification ability. To see the equations of RMSE, MARE and R2, please refer to the supplement. (Pages 16-17, Lines 334-345) Description of the evaluation results has been added in Revised Manuscript as follows: ïČŸ For results of the synthetic experiment with the TMWB model Figures 6 (b) and (c) focuses on the ability of the four methods to identify time-varying parameters. It can be seen that the RMSE and MARE values of the 3-SSC-DP are larger than those of other methods in most cases. That is because the sub-period length that serves as a calibration period for MCMC is too short (i.e., three months) that the estimated parameters are associated with higher uncertainties. Regarding the synthetic true parameters are a constant value (scenario 1), 12-SSC-DP gives the best performance with the lowest RMSE, MARE and highest R2. The observations and estimated parameters are presented in Figure 7 (b). It shows that the estimated parameters obtained by EnKF vary at every time step, resulting in larger deviations from the observations than 6-SSC-DP and 12-SSC-DP. When the synthetic true parameters vary linearly (scenarios 2, 4, and 6), 12-SSC-DP produces best estimations in comparison with EnKF, 3-SSC-DP, and 6-SSC-DP. The performances of 6-SSC-DP and EnKF are similar. When the synthetic true parameters vary sinusoidally from month to month, EnKF gives the best estimations in scenario 3. The poor performances of 6-SSC-DP and 12-SSC-DP can be explained by the sub-period length being much longer than the actual one. When the parameters vary periodically at six-month intervals (scenario 5), 6-SSC-DP yields the best performance with the lowest RMSE, MARE and highest R2. The differences of estimation performances among 3-SSC-DP, 12-SSC-DP and EnKF are small. The estimated parameters for scenario 5 have been plotted in Fig. 7(a).

Although 3-SSC-DP and 12-SSC-DP have different lengths of sub-periods, they can also detect the correct seasonal signal of the parameters. For the annual variation in parameters (scenario 7), 12-SSC-DP and 6-SSC-DP produce better results than EnKF. Similar results can be seen in scenario 8 where C has a combined variation from year to year. In summary, the results indicate that the SSC-DP with a suitable length can estimate more accurate parameters than EnKF. (Pages 22∼24, Lines 507-546) ïČŸ For results of the synthetic experiment with the Xinanjiang model Figures 9(b) and (c) compares the time-varying parameter estimation performance among the four methods. In scenarios 1 and 2, 2-SSC-DP produces the lowst RMSE, MARE and R2, followed by the 1-SSC-DP. The 1-SSC-DP is slightly superior to the 1-SSC and significantly outperforms the SSC-EnKF for the two scenarios. When the synthetic true parameters vary sinusoidally from month to month (scenario 3), the estimated parameters are plotted in Fig. 10. It can be seen that 1-SSC-DP successfully detects seasonal signal in every parameter. The SSC-EnKF performs well for R2, but it has high MARE. Although the average MARE of the SSC and 2-SSC-DP are lower than that of SSC-EnKF, the R2 of them are relatively low. Therein, form Fig. 10, the estimated parameters by the 1-SSC fluctuate generally periodically, but the variations are dramatic, resulting in lowest R2 for CI, KI, KG and NK. The estimated parameters of the 2-SSC-DP fluctuate more slowly, but the sub-period length is too long. In scenario 4, 1-SSC performs better than the SSC-EnKF and 2-SSC-DP, but is still slightly inferior to the 1-SSC-DP. Overall, the 1-SSC-DP achieves higher-quality and more robust parameter estimations performances than the other methods. (Pages 24, Lines 652-666) (3) The figure 5 is replaced by Figure 6 in the Revised Manuscript. The results as shown in Figure 6 have been presented in the reply (2) of R1-C7.

8. The streamflow, climate and underlying surface conditions in the two study areas were not analyzed in this study. However, it is critical to the estimation of time-varying parameters. Reply: Figure 5 has been modified. The details of the Wuding River basin have been added as follows: As illustrated in Fig. 5(a), the station furthest downstream, Baijiachuan, drains an area of 29,662 km2 (98 % of the total basin) and records the

daily runoff data. The data of the daily precipitation and streamflow in the Wuding River basin were obtained from the local Hydrology and Water Resources Bureau of China, the quality of which has been checked by the official authorities, and there are no gaps among these data for all the hydrological stations. It can be seen from Fig. 5(c) that the annual streamflow in the Wudinghe River basin has a distinct decreasing trend, while seasonal variations are not significant, but the annual precipitation and pan evaporation generally have no trend, suggesting the impacts of human activities on rainfall–runoff relationships. (Pages 19∼20, Lines 429-440)

The details of the Wuding River basin have been added as follows: As illustrated in Fig. 5(a), the station furthest downstream, Baijiachuan, drains an area of 29,662 km2 (98 % of the total basin) and records the daily runoff data. The data of the daily precipitation and streamflow in the Wuding River basin were obtained from the local Hydrology and Water Resources Bureau of China, the quality of which has been checked by the official authorities, and there are no gaps among these data for all the hydrological stations. It can be seen from Fig. 5(c) that the annual streamflow in the Wudinghe River basin has a distinct decreasing trend, while seasonal variations are not significant, but the annual precipitation and pan evaporation generally have no trend, suggesting the impacts of human activities on rainfall–runoff relationships. (Pages 19∼20, Lines 429-440) The details of the Xun River basin have been added as follows: It can be observed from Fig. 5(d) that no trend is found in annual precipitation, pan evaporation and streamflow, suggesting that the relationship between precipitation and runoff of the Xun River basin is rarely affected by human activities during 1991-2001. However, there exhibits strong seasonal patterns in these three climatic and hydrological variables, suggesting that seasonal variations in hydrological parameters should be considered. (Pages 21, Lines 472-477)

9. In lines 175-176, the assumption that the continuity condition aims to minimize the difference between the estimated parameters for sub-periods i and i+1 unreasonable. The differences between two consecutive sub-periods represent the time-varying

changes of the catchment. The continuity conditions for enhancing the model performance should focus on the model structure, such as state variables. Reply: Thanks for the comment. The main hypothesis of parameter continuity is justified as follows: 1. The hypothesis of parameter continuity can be found in the model prediction process of the ensemble Kalman filter (EnKF). Therein, the values of the parameters at the time step t+1 are forecasted by perturbing those of parameters from the time step t. To see the equation, please refer to the supplement. In the equation, there is a white noise following a Gaussian distribution with zero mean and specified covariance of ,R_t, which is very small. That is, the fluctuations between parameters of adjacent sub-periods can be little. 2. Some conceptual hydrological parameters reflect the catchment characteristics, such as soil water storage capacity in the Xinanjiang model. While climate change and human activities exert influence on catchment characteristics, the soil water storage capacity can hardly change dramatically in a very quick time, such as an hour. Hence, it is reasonable to consider parameter continuity in estimating time-varying parameters. This point has been added in the Revised Manuscript as follows: Some conceptual hydrological parameters reflect the catchment characteristics. While climate change and human activities exert influence on these catchment characteristics, they can hardly change dramatically in a very quick time, such the soil water storage capacity. (Pages 5, Lines 89-92)

10. Minor comment. The resolution of Figure 5 is low and information is not presented. Reply: The Figure 5 is replaced by Figure 6 in the Revised Manuscript to be easier to read.

Bardossy, A. (2007) Calibration of hydrological model parameters for ungauged catchments. Hydrology and Earth System Sciences 11(2), 703-710. Lan, T., Lin, K., Xu, C.-Y., Tan, X. and Chen, X. (2020) Dynamics of hydrological-model parameters: mechanisms, problems and solutions. Hydrology and Earth System Sciences 24(3), 1347-1366. Lan, T., Lin, K.R., Liu, Z.Y., He, Y.H., Xu, C.Y., Zhang, H.B. and Chen, X.H. (2018) A Clustering Preprocessing Framework for the Subannual Calibration of

a Hydrological Model Considering Climate-Land Surface Variations. Water resources research 54(0). Deng, C., Liu, P., Guo, S., Li, Z. and Wang, D. (2016) Identification of hydrological model parameter variation using ensemble Kalman filter. Hydrology and Earth System Sciences 20(12), 4949-4961.

Please also note the supplement to this comment:
https://hess.copernicus.org/preprints/hess-2019-639/hess-2019-639-AC1-supplement.pdf

[Figure]

**Fig. 1.** Figure 1 The flowchart of the methodologies

[Figure]

**Fig. 2.** Figure 6 Comparison between the EnKF and SSC-DP methods for (a) streamflow simulation and identification of (b) parameter C and (c) parameter SC.

[Figure]

**Fig. 3.** Figure 9 Comparison among the SSC, SSC-EnKF and SSC-DP methods for (a) streamflow simulation and parameter identification in terms of (b) RMSE, (c) MARE and (d) R2.

[Figure]

**Fig. 4.** Figure 12 Simulation performance for streamflow in the Wuding River basin.

[Figure]

**Fig. 5.** Figure 15 Simulation performance for streamflow in the Xun River basin.

[Figure]

**Fig. 6.** Figure 5 Location of (a) Wuding River basin and (b) Xun River basin. The plots (c) and (d) show the average yearly and monthly variations of precipitation, pan evaporation and streamflow in the Wuding

---

## Author Comment (AC2) · 26 Sep 2020

Responses to Reviewer #2:

1. OVERALL RECOMMENDATION The manuscript addresses the important topic of calibration of rainfall-runoff model parameters, and presents results obtained on two different catchments with two models. Even if the introduction includes relevant references and the methods are well presented, the paper lacks important discussions on the rainfall-runoff model performances, observed time series quality, attribution of observed/simulated changes, consideration of only two catchments, and several obtained results are over-interpreted. Finally, several figures and tables must be significantly improved. Therefore, I think the manuscript requires major revision before publication. Reply: Thank you for reviewing our manuscript and for the professional comments, which are carefully followed in making revision.

2. GENERAL COMMENTS (1) The tables 6 to 8 might be presented as figures to be more easily interpreted. Figure 5, 7 and 10 are very difficult to read, and must be significantly improved. Reply: 1. Table 6 and Figure 5 have been modified and replaced by Figure 6 in Revised Manuscript. Please refer to the supplement. 2. Table 7 has been presented as Figure 8 in Revised Manuscript. Please refer to the supplement. 3. Table 8 and Figure 7 have been modified and replaced by Figure 9 in Revised Manuscript. Please refer to the supplement. 4. Figure 10 is modified and replaced by Figure 13 in Revised Manuscript. Please refer to the supplement.

(2) Line 28 to 29: several studies highlighted the difficulty of conceptual rainfall-runoff models in the context of climate change impact studies. Reply: Thanks. We agree that conceptual rainfall-runoff models can be difficult to simulate the variations in discharge in response to climate changes in some cases (Merz et al. 2011, Fowler et al. 2020). That is, the simulation accuracy reduces when the conceptual model is applied in situations where the climatic conditions, e.g., dry periods, are not consistent with that of the calibration period, e.g., wet periods. Some literatures have made improvements to enhance parameter transferability between various climatic conditions. One approach is to allow the parameters of the conceptual model to change (Stephens et al. 2019, Deng et al. 2019), which can efficiently improve the accuracy of the conceptual model and simulate the response of runoff in a changing environment.

(3) Line 35 to 36: the terms "constants" and "stable" must be defined: constant/stable in space and/or in time? Reply: The terms are defined as "constant in time scale" and "temporally stable". The statement at line 35 to 36 will be modified in the Revised Manuscript: Parameters are usually regarded as constants in time scale, because of the general idea that catchment conditions are temporally stable.

(4) Line 43 to 44: in this context, it may be needed to define what is called "climate conditions". Reply: Here, the "climate conditions" means "wet/dry periods". To avoid confusion, the sentence at line 42 to 44 of the Revised Manuscript is modified by replacing "climate conditions" with "wet/dry periods": Fowler et al. (2016) pointed out that the parameter set obtained by mathematical optimization based on wet periods may not be robust when applied in dry periods.

(5) Line 122: the terms "behavioural" must be clearly defined or not used in this context. Reply: The "behavioural" means "important to calibration metrics and predictions". To avoid confusion, the "behavioural" is replaced by "sensitive" in the line 122 of the Revised Manuscript.

(6) Line 137, line 150 and Table 1 and 2: please presents parameter units. Reply: Thanks for reminder. The parameter units have been added at Line 137, line 150 as follows: The model has only two parameters (Table 1), C and SC. The parameter C takes account of the effect of the change of time scale when simulating actual evapotranspiration. The parameter SC represents the field capacity (mm). (Pages 8, Lines 164∼166) The meaning, range and units of all the parameters in the Xinanjiang model are listed in Table 2. (Pages 9, Lines 176∼177) The parameter units have also been added in Tables 1 and 2. Please refer to the supplement.

(7) Section 2.2: the need to reduce the number of Xinanjian free parameters using a sensitivity analysis must be investigated more deeply in the paper. In the current version of the paper, this model is considered with different number of free parameters depending on the modeling experiments. Why not calibrating the 15 free parameters of this rainfall-runoff model for all experiments? Reply: Here we add a synthetic experiment with the Xinanjiang model, where the true values of KE, CI, CG, KI, KG, and NK have periodic variations with changes every month (720h) and those of the insensitive parameters remain temporally constant. The 1-SSC-DP is applied to this experiment and all 15 free parameters are calibrated without a sensitivity analysis. The estimated parameters are plotted in Figure R1. From the figure, it can be seen that except the

estimated KE, CI, CG, KI, KG, and NK, the estimations of the insensitive parameters, such as WM, X and Y, are also recognized to vary significantly during the calibration period. This is inconsistent with the true values in the synthetic experiment, and the attribution analysis between time-varying parameters and watershed characteristics will be mistaken in practical use, which also occurs in the data assimilation method. Hence a sensitivity analysis is needed to find which parameters are really important for calibration. This point has been highlighted in the Revised Manuscript as follows: A sensitivity analysis is employed to focus efforts on parameters important to calibration and avoid prohibitive computational cost, as outlined in Sect. 2.2. (Pages 7, Lines 139-141)

(8) Section 2.3.1: one of the main hypotheses of this paper is the important "fluctuations" of the model parameter values over adjacent sub-periods, hypothesis that is not justified by the literature review, and that is not illustrated with the obtained results. This point must be discussed more deeply in the paper. Reply: Thanks for the comment. The main hypothesis of parameter continuity is justified as follows: 1. The hypothesis of parameter continuity can be found in the model prediction process of the ensemble Kalman filter (EnKF). Therein, the values of the parameters at the time step t+1 are forecasted by perturbing those of parameters from the time step t. To see the equation, please refer to the supplement.In the equation,there is a white noise following a Gaussian distribution with zero mean and specified covariance of ,$R_t$, which is very small. That is, the fluctuations between parameters of adjacent sub-periods can be little. 2. Some conceptual hydrological parameters reflect the catchment characteristics, such as soil water storage capacity in the Xinanjiang model. While climate change and human activities exert influence on catchment characteristics, the soil water storage capacity can hardly change dramatically in a very quick time, such as an hour. Hence, it is reasonable to consider parameter continuity in estimating time-varying parameters. This point has been added in the Revised Manuscript as follows: Some conceptual hydrological parameters reflect the catchment characteristics. While climate change and human activities exert influence on these catchment characteristics, they can hardly

change dramatically in a very quick time, such the soil water storage capacity. (Pages 5, Lines 89-92)

(9) Evaluation criteria: Why only use the NSE criterion as only evaluation criteria, and no other criteria, such as KGE and its components? NSE appears to be nondiscriminating between considered calibration methods. Using other calibration criteria- looking at different time step and/or different error characteristics such as bias on the highest streamflow values – might be interesting in this context. Reply: As well as NSE coefficient, two evaluation metrics have been added in Revised Manuscript: relative error (RE) and the NSE on logarithm of streamflow (NSEln). In the revised paper, these evaluation metrics are described as follows: The streamflow simulations given by the proposed method are verified using the NSE, relative error (RE) and NSE on logarithm of streamflow (NSEln) (Hock, 1999). RE evaluates the error of the total volume of streamflow, while NSE and NSEln evaluate the agreement between the hydrograph of observations and simulations. NSE is more sensitive to high flows, but NSEln focuses more on low flows. Higher values of NSE, NSEln and lower values of RE indicate better streamflow simulations. To see the equations of NSE, RE and NSEln, please refer to the supplement.(Pages 15∼16, Lines 324-333) A description of the evaluation results has been added as follows: ïČŸ For results of the synthetic experiment with the TMWB model Figure 6(a) presents the runoff simulation performance for various scenarios. In scenario 1, the NSE values of the three SSC-DP methods are all higher than that of EnKF. The results of NSEln show no significant differences among various methods. For scenarios 2, 4, and 6, where true parameters have linear trends, the 6-SSC-DP and 12-SSC-DP are superior to the EnKF and 3-SSC-DP in terms of NSE and NSEln. In scenario3, where the true parameters have periodic variations and change every month, the NSE and NSEln values of 6-SSC-DP and 12-SSC-DP decrease significantly, because the assumed sub-period length is longer than the time-scale of actual variations. Similarly, in scenario 5, 12-SSC-DP performs worst for NSE and NSEln, but 6-SSC-DP performs best. In scenario 7 and 8, both 6-SSC-DP and 12-SSC-DP perform better than EnKF. According to the evaluations of NSE and NSEln, the SSC-DP

offers improved accuracy than the EnKF if the proper length is chosen. Another advantage of the SSC-DP is the low RE. For all scenarios, the SSC-DP methods significantly outperform for RE compared with EnKF. Among the SSC-DP methods, the RE of 3-SSC-DP is the smallest. (Page 22, Lines 492∼506) ïČŸ For results of the synthetic experiment with the Xinanjiang model The simulated streamflow and identification of time-varying parameters was compared across four methods: 1-SSC, SSC-EnKF, 1-SSC-DP, and 2-SSC-DP. The simulation performance is summarized in Figure 9(a). For all scenarios, the NSE of 2-SSC-DP is the lowest, but it performs better for low flows. The SSC-EnKF produces the highest RE in scenarios 2, 3 and 4, indicating the problem of simulating water balance. The SSC and 1-SSC-DP perform well for all scenarios in terms of NSE, RE and NSEln. Wherein, the SSC performs better than the 1-SSC-DP with regard to RE, while 1-SSC-DP is slightly superior to SSC in scenario 3 with higher NSEln. (Page 24, Lines 560∼566) ïČŸ For results of case study in Wuding River basin The simulation performance is presented in Figure 12. The values of the NSEs are relatively low, it is because the streamflow in dry regions is difficult to simulate. It can be seen that the 12-SSC-DP gives the best simulation results among different methods with the highest NSE, NSEln and low RE. Although the 12-SSC produces relatively high NSE, but it performs worst simulations for low flows. The SSC-EnKF has relative high NSEln, but the RE of it is the largest. Overall, the 12-SSC-DP significantly improve the simulation performance of the Xinanjiang model in the Wuding River basin. (Page 26, Lines 706∼713) ïČŸ For results of case study in Xun River basin The simulation performance is presented in Figure 15. All methods performed well, with NSE values of 92.5 %, 93.0 %, 95.0 %, and 94.8 % for the conventional method, 3-SSC-EnKF, 3-SSC, and 3-SSC-DP, respectively. 3-SSC and 3-SSC-DP also perform well for NSEln compared with 3-SSC-EnKF and the conventional method. However, as regards to RE, the values are 0.0007 and 0.0324 for 3-SSC-DP and 3-SSC-DP, respectively. It indicated that the 3-SSC-DP can better simulate water balance than the 3-SSC in the Xun River basin. (Page 28∼29, Lines 785∼806)

(10) Section 3.2 (Wuding river basin), lines 364 to 369: the changes of the studied

catchment characteristics seem to be decisive for the interpretation of the results obtained on this watershed. Nevertheless, no quantitative results / analysis of these changes are given in the paper: what is the percentage of the catchment that has been afforested? What are the number and the capacity of the built reservoirs? When are they built? Finally, an important point not discussed in the paper is the stationary and then quality of the precipitation and streamflow time series studied and used for the model calibration. This point is crucial in this context and need to be discussed. Reply: This comment involves two aspects: 1. For the first aspect, a quantitative analysis of the changes in the Wuding river basin has been added in the Revised Manuscript, including the areas of tree planning and check dams for soil and water conservations. This point is described in the Revised Manuscript as follows: Soil and water conservation measures, such as construction of the check dams and afforestation, have been undertaken since the 1960s. The areas of two soil and water conservation measures are plotted in Fig. 5(e), the data of which were collected from Zhang et al. (2002). The areas of tree planning have an increasing trend, but the slope gets much larger after 1972. It indicates that the greater efforts have been made for afforestation since the turning point. Similarly, the areas of dammed lands also increase, but the rate gets slower after 1972. These two soil and water conservation measures had changed underlying surface of the watershed, and impacted the relationship between precipitation and runoff (Gao et al., 2017; Jiao et al., 2017). (Pages 20, Lines 441-449) 2. The reviewer concerns the quality of the precipitation and streamflow used. The data of the daily precipitation and streamflow in the Wuding River basin are obtained from the local Hydrology and Water Resources Bureau of China, the quality of which has been checked by the official authorities, and there are no gaps among these data for all the hydrological stations. This point has been clarified in the Revised Manuscript. (Page 20, Lines 433∼436)

(11) Section 3.3 (Xun River basin): same remarks as the Wuding river basin: what about potential changes on this basin? Are precipitation and streamflow time series of good quality? Reply: This comment involves two aspects: 1. The seasonal variations

of the mean monthly precipitation, pan evaporation and streamflow are shown in Fig. 5(d). It shows that Xun River basin exhibits strong seasonal patterns in these climatic and hydrological variables. This point is added in the Revised Manuscript as follows: It can be observed from Fig. 5(d) that no trend is found in annual precipitation, pan evaporation and streamflow, suggesting that the relationship between precipitation and runoff of the Xun River basin is rarely affected by human activities during 1991-2001. However, there exhibits strong seasonal patterns in these three climatic and hydrological variables, suggesting that seasonal variations in hydrological parameters should be considered. (Pages 21, Lines 472-477) (2) The data of the precipitation and streamflow in the Xun River basin are also obtained from the local Hydrology and Water Resources Bureau of China, the quality of which has been checked by the official authorities, and there are no gaps among these data for all the hydrological stations. This point has been clarified in the Revised Manuscript. (Pages 21, Lines 469-471)

(12) Section 4.1: the seasonal signal of the parameter values (cf. figures 6 and 8) must be more significantly discussed in the paper. Reply: More discussion concerning the seasonal signal of the parameter values is added in the Revised Manuscript as follows: 1. For the synthetic experiment with the TMWB model When the synthetic true parameters vary sinusoidally from month to month, EnKF gives the best estimations in scenario 3. The poor performances of 6-SSC-DP and 12-SSC-DP can be explained by the sub-period length being much longer than the actual one. When the parameters vary periodically at six-month intervals (scenario 5), 6-SSC-DP yields the best performance with the lowest RMSE, MARE and highest R2. The differences of estimation performances among 3-SSC-DP, 12-SSC-DP and EnKF are small. The estimated parameters for scenario 5 have been plotted in Fig. 7(a). Although 3-SSC-DP and 12-SSC-DP have different lengths of sub-periods, they can also detect the correct seasonal signal of the parameters. For the annual variation in parameters (scenario 7), 12-SSC-DP and 6-SSC-DP produce better results than EnKF. Similar results can be seen in scenario 8 where C has a combined variation from year to year. In summary, the results indicate that the SSC-DP with a suitable length can estimate more accurate

parameters than EnKF. (Pages 23~24, Lines 534-546) 2. For the synthetic experiment with the Xinanjiang model When the synthetic true parameters vary sinusoidally from month to month (scenario 3), the estimated parameters are plotted in Fig. 10. It can be seen that 1-SSC-DP successfully detects seasonal signal in every parameter. The SSC-EnKF performs well for R2, but it has high MARE. Although the average MARE of the SSC and 2-SSC-DP are lower than that of SSC-EnKF, the R2 of them are relatively low. Therein, form Fig. 10, the estimated parameters by the 1-SSC fluctuate generally periodically, but the variations are dramatic, resulting in lowest R2 for CI, KI, KG and NK. The estimated parameters of the 2-SSC-DP fluctuate more slowly, but the sub-period length is too long. In scenario 4, 1-SSC performs better than the SSC-EnKF and 2-SSC-DP, but is still slightly inferior to the 1-SSC-DP. Overall, the 1-SSC-DP achieves higher-quality and more robust parameter estimations performances than the other methods. (Pages 25, Lines 656-666)

(13) Line 456 to 458: this conclusion must be significantly moderated: the "SSC-DP" calibration method is by definition better to select more continuous parameter values. Reply: This conclusion has been moderated in the Revised Manuscript as follows: Overall, the 1-SSC-DP achieves higher-quality and more robust parameter estimation performances than the other methods. (Page 25, Lines 665~666)

(14) Section 4.2: this data analysis is crucial in this context. It might be relevant to present it in the data section. Moreover, this analysis must be significantly improved: what about potential errors (random or systematic) in the observed precipitation and streamflow series? What about potential break in the streamflow series due to rating curve changes? What is the statistical significance of this analysis? The analysis of only one catchment requires to look carefully the studied time series in the context of attribution of changes. The relative bad performance of the rainfall-runoff model on this catchment (NSE=0.41) must be discussed. In particular, the systematic streamflow underestimation for the different calibration methods must be discussed. Reply: This comment involves four aspects: 1. For the first aspect, the hydrological data are

collected from the local Hydrology and Water Resources Bureau of China, the systematic errors of which have been checked by the official authorities. Additionally, random errors are considered in the synthetic experiment, the results show that 5% random errors have little influence on the SSC-DP. 2. The reviewer concerns that the potential break in the streamflow series due to rating curve changes. The streamflow data used have been checked to guarantee their continuity, that is, no break (the discharge equal to zero) has been found except on two discontinuous days. Since the daily streamflow is also very low near the break, the values of the streamflow are reasonable. 3. It is found that all the analyses of the linear regression are significant. The statistical significance of the analysis has been added in the Revised Manuscript as follows. The Figure 11 is also modified. Please refer to the supplement. The two linear slopes (p-value < 0.05) of the curves are different before and after 1972, demonstrating the relationship between precipitation and runoff changes under the soil and water conservation measures. (Page 25, Lines 669~672) 4. The reviewer also concerns the bad simulation performance in the Wuding River basin. It is because the streamflow in dry regions is difficult to simulate. The main reason is the deficiencies of the model structure. This point is added in the Revised Manuscript. (Page 26, Lines 706~708; Pages 26~27, Lines 720-735)

(15) Line 492 to 495: the "unreasonable model states" between sub-periods might be illustrated in the paper. Reply: The statement about "the unreasonable model states" is an incorrect description and has been deleted in the Revised Manuscript. The parameters over each sub-period are calibrated separately using the SSC method. Several sets of parameters can lead to similar simulation performance in each sub-period, i.e., parameter equifinality. This equifinality causes uncertainty in simulating fluxes and streamflow. Here, the time-series of the estimated groundwater discharge have been plotted in Fig. R2. From the Fig. R2, the estimated groundwater discharge by the SSC fluctuates dramatically on December 27, 1977, which seems unreasonable, while the estimations by the SSC-DP have no dramatically fluctuations. Hence, the SSC-DP outperforms the SSC for the Wuding River basin.

(16) Line 500 to 501: this conclusion must be moderated, since results have been obtained on one catchment only. Reply: This sentence has been moderated in the Revised Manuscript: It can be inferred the 12-SSC-DP is more applicable to the simulation of streamflow in the Wuding River basin. (Pages 27, Lines 739-740)

(17) Line 511 to 514: this attribution analysis must be moderated (see previous remarks on attribution analysis). Reply: The attribution analysis has been moderated in the Revised Manuscript: The results show that WM remains constant before and after 1972, but WUM varies significantly over this period, indicating that the distribution of soil water capacity may change, i.e., WUM decreases but WLM increases. A Person correlation analysis is applied to investigate the relationship between the areas of tree planning and WUM as well as WLM. It is found that there is a significant negative correlation (Pearson correlation efficient =-0.38, P<0.05) between the areas of tree planning and WUM. While WLM has a nonsignificant positive correlation (=0.26, P>0.05) with the areas of tree planning. It can be inferred that less severe soil erosion occurred, because the upper layers became thinner while the lower layer, where vegetation roots dominate, became thicker (Jayawardena and Zhou, 2000). Additionally, IMP is significantly correlated with the areas of tree planning (=-0.33, P<0.05). Except for afforestation, the areas of the dammed lands are significantly correlated with WLM (=0.46, P<0.05), suggesting that the construction of the check dams also has influence on the soil water capacity of the Wuding river basin. Other parameters, KE, KI, KG, N and NK have little differences before and after 1972. The variations in WLM and IMP slowed down after the turning point, similar to the results of Deng et al. (2016). (Pages 27~28, Lines 741-770)

(18) Line 520 to 522: again, what about potential error in the rating curve in this context? Reply: The potential error in the rating curve is considered in this study from two aspects: 1. The streamflow data are managed by the local Hydrology and Water Resources Bureau. There is a strict specification for hydrometry for drawing the rating curve. Hence, the streamflow accuracy is guaranteed. 2. In the synthetic experiment,

the uncertainty of observations has been considered, and the results show that 5% random errors have little influence on the SSC-DP.

(19) Line 526 to 539: why not presenting a Figure such as Figure 10 to illustrate rainfall-runoff simulations on this catchment? Reply: Thanks for the reminder. The Figure and the description have been added in the Revised Manuscript as follows: Figure 16 illustrates the hydrograph and quantile-quantile plots for the simulations in the Xun river basin. It is evident that the peak flows estimated by the 3-SSC is higher than those of 3-SSC-DP, and 3-SSC-DP simulate better the flows ranging from 100 m3/s to 200 m3/s. (Pages 29, Lines 806-809)

(20) Line 531 to 532: is this out-performance significant? Reply: To give a more comprehensive evaluation, two metrics, relative error (RE) and the NSE on logarithm of streamflow (NSEln), are added. This sentence on line 531 to 532 has been modified as follows: As regards to RE, the values are 0.0007 and 0.0324 for 3-SSC-DP and 3-SSC-DP, respectively. It indicated that the 3-SSC-DP can better simulate water balance than the 3-SSC in the Xun River basin. (Pages 28~29, Lines 789-806)

(21) Line 534 to 539 and line 637 to 648: again, these attribution conclusions must be moderated, because they are drawn from only two basins, without any investigation of potential systematic errors in the observed time series. Reply: The statement is an incorrect description and has been deleted in the Revised Manuscript. Here, the estimations of groundwater discharge are plotted in the Fig. R3. It can be seen that the estimations are similar for SSC and SSC-DP, which is different from that in the Wuding case study. It suggests that the SSC-DP gives more robust simulation performance for both case studies.

Merz, R., Parajka, J. and Bloeschl, G. (2011) Time stability of catchment model parameters: Implications for climate impact analyses. Water resources research 47(W02531). Fowler, K., Knoben, W.J.M., Peel, M.C., Peterson, T.J., Ryu, D., Saft, M., Seo, K.-W. and Western, A. (2020) Many Commonly Used Rainfall-Runoff Models

Lack Long, Slow Dynamics: Implications for Runoff Projections. Water resources research 56(5). Stephens, C.M., Marshall, L.A. and Johnson, F.M. (2019) Investigating strategies to improve hydrologic model performance in a changing climate. Journal of Hydrology 579. Deng, C., Liu, P., Wang, W., Shao, Q. and Wang, D. (2019) Modelling time-variant parameters of a two-parameter monthly water balance model. Journal of Hydrology 573, 918-936.

Please also note the supplement to this comment:
https://hess.copernicus.org/preprints/hess-2019-639/hess-2019-639-AC2-supplement.pdf

[Figure]

**Fig. 1.** Figure 6 Comparison between the EnKF and SSC-DP methods for (a) streamflow simulation and identification of (b) parameter C and (c) parameter SC.

[Figure]

**Fig. 2.** Figure 8 Comparison among EnKF, SSC-EnKF, and EnKS in the synthetic experiment with the Xinanjiang model

[Figure]

**Fig. 3.** Figure 9 Comparison among the SSC, SSC-EnKF and SSC-DP methods for (a) stream-flow simulation and parameter identification in terms of (b) RMSE, (c) MARE and (d) R2.

[Figure]

**Fig. 4.** Figure 13 Streamflow simulation hydrograph (left panels) and quantile-quantile plots (right panels) using conventional method, SSC-EnKF, SSC, and SSC-DP for the Wuding River basin. (a) The quantile-qu

[Figure]

**Fig. 5.** Figure R1 The parameters estimated without a sensitivity analysis

[Figure]

**Fig. 6.** Figure 12 Simulation performance for streamflow in the Wuding River basin.

[Figure]

**Fig. 7.** Figure 15 Simulation performance for streamflow in the Xun River basin.

[Figure]

**Fig. 8.** Figure 5 Location of (a) Wuding River basin and (b) Xun River basin. The plots (c) and (d) show the average yearly and monthly variations of precipitation, pan evaporation and streamflow in the Wuding

[Figure]

**Fig. 9.** Fig. 11 Double mass curves between daily runoff and precipitation for (a) Wuding River basin from 1958–1972; (b) Wuding River basin from 1973–2000; (c) Xun River basin from 1991–2001. Subgraph (d) rep

[Figure]

**Fig. 10.** Fig. R2 The estimated groundwater discharge of the Wuding River basin

[Figure]

**Fig. 11.** Figure 16 (a) Streamflow simulation hydrograph (left panels) and quantile-quantile plots (right panels) using conventional method, SSC-EnKF, SSC, and SSC-DP for the Xun River basin. (b) The quantile-q

[Figure]

**Fig. 12.** Fig. R3 The estimated groundwater discharge of the Xun River basin

**Supplement:**

**Responses**

The reviewer's comments have been considered carefully, and the manuscript has been extensively revised. The following tables summarize the reviewer's comments. For convenience, we have classified them into categories. Comments regarding similar issues and their locations are summarized together. Simplified notation has been used to assist the reviewer in conveniently locating a specific comment: R-Reviewers, and C-Comment. For example, R2-C1 denotes comment 1 made by Reviewer #2. This simplified form is used in all of the following tables.

**Category I. The methodology**

| | Comment | Reviewer-Location |
|---|---|---|
| 1 | Justification of using conceptual hydrological models | R2-C2 |
| 2 | Adding units for Xinanjiang model parameters | R2-C6 |
| 3 | Justification of using the sensitivity analysis | R2-C7 |
| 4 | Adding the evaluation metrics | R2-C9 |

**Category II. The case study and results**

| | Comment | Reviewer-Location |
|---|---|---|
| 1 | Quantified analysis of the soil and water conservation measures on the Wuding River basin | R2-C10 |
| 2 | Quantified analysis of the seasonal variations on the Xun River basin | R2-C11 |
| 3 | Quality of the data used | R2-C10; R2-C11; R2-C14; R2-C18 |
| 4 | Figure modification | R2-C1; |
| 5 | Replacing tables with figures | R2-C1 |
| 6 | Modification for the results | R2-C12; R2-C20 results |
| 7 | Statistical significance of this analysis | R2-C14 |
| 8 | Explanation for the relative bad performance on the Wuding River basin | R2-C14 |
| 9 | Modification of the attribution analysis | R2-C17 |
| 10 | Adding a hydrograph plot for the Xun River basin | R2-C19 |

**Category III. Others**

| | Comment | Reviewer-Location |
|---|---|---|
| 1 | Justification of the parameter continuity assumption | R2-C8 |
| 2 | Explanation and modification of words and phrasing | R2-C3;  R2-C4;  R2-C5; R2-C13;  R2-C15;  R2-C16; R2-C21 |

**Reviewer #2:**

1. OVERALL RECOMMENDATION

The manuscript addresses the important topic of calibration of rainfall-runoff model parameters, and presents results obtained on two different catchments with two models. Even if the introduction includes relevant references and the methods are well presented, the paper lacks important discussions on the rainfall-runoff model performances, observed time series quality, attribution of observed/simulated changes, consideration of only two catchments, and several obtained results are over-interpreted. Finally, several figures and tables must be significantly improved. Therefore, I think the manuscript requires major revision before publication.

**Reply:**

Thank you for reviewing our manuscript and for the professional comments, which are carefully followed in making revision.

2. GENERAL COMMENTS

(1) The tables 6 to 8 might be presented as figures to be more easily interpreted. Figure 5, 7 and 10 are very difficult to read, and must be significantly improved.

**Reply:**

(1) Table 6 and Figure 5 have been modified and replaced by Figure 6 in Revised Manuscript as follows:

(a) Simulation performance for streamflow

[Figure]

(b) Estimation performance for parameter C

(c) Estimation performance for parameter SC

□ ENKF    ○ 3-SSC-DP    △ 6-SSC-DP    ▽ 12-SSC-DP

Figure 6 Comparison between the EnKF and SSC-DP methods for (a) streamflow simulation and identification of (b) parameter C and (c) parameter SC.

(2) Table 7 has been presented as Figure 8 in Revised Manuscript as follows:

[Figure]

Figure 8 Comparison among EnKF, SSC-EnKF, and EnKS in the synthetic experiment with the Xinanjiang model

(3) Table 8 and Figure 7 have been modified and replaced by Figure 9 in Revised Manuscript as follows:

[Figure]

Figure 9 Comparison among the SSC, SSC-EnKF and SSC-DP methods for (a) streamflow simulation and parameter identification in terms of (b) RMSE, (c) MARE and (d) $R^2$.

(4) Figure 10 is modified and replaced by Figure 13 in Revised Manuscript as follows:

[Figure]

Figure 13 Streamflow simulation hydrograph (left panels) and quantile-quantile plots (right panels) using conventional method, SSC-EnKF, SSC, and SSC-DP for the Wuding River basin. (a) The quantile-quantile plot for all streamflow; (b) The quantile-quantile plot for streamflow lower than 100 m³/s.

(2) Line 28 to 29: several studies highlighted the difficulty of conceptual rainfall-runoff models in the context of climate change impact studies.

**Reply:**

Thanks. We agree that conceptual rainfall-runoff models can be difficult to simulate the variations in discharge in response to climate changes in some cases (Merz et al. 2011, Fowler et al. 2020). That is, the simulation accuracy reduces when the conceptual model is applied in situations where the climatic conditions, e.g., dry periods, are not consistent with that of the calibration period, e.g., wet periods. Some literatures have made improvements to enhance parameter transferability between various climatic conditions. One approach is to allow the parameters of the conceptual model to change (Stephens et al. 2019, Deng et al. 2019), which can efficiently improve the accuracy of the conceptual model and simulate the response of runoff in a changing environment.

(3) Line 35 to 36: the terms "constants" and "stable" must be defined: constant/stable in space and/or in time?

**Reply:**

The terms are defined as "constant in time scale" and "temporally stable". The statement at line 35 to 36 will be modified in the Revised Manuscript: Parameters are usually regarded as constants in time scale, because of the general idea that catchment conditions are temporally stable.

(4) Line 43 to 44: in this context, it may be needed to define what is called "climate conditions".

**Reply:**

Here, the "climate conditions" means "wet/dry periods". To avoid confusion, the sentence at line 42 to 44 of the Revised Manuscript is modified by replacing "climate conditions" with "wet/dry periods": Fowler et al. (2016) pointed out that the parameter set obtained by mathematical optimization based on wet periods may not be robust when applied in dry periods.

(5) Line 122: the terms "behavioural" must be clearly defined or not used in this context.

**Reply:**

The "behavioural" means "important to calibration metrics and predictions". To avoid confusion, the "behavioural" is replaced by "sensitive" in the line 122 of the Revised Manuscript.

(6) Line 137, line 150 and Table 1 and 2: please presents parameter units.

**Reply:**

Thanks for reminder. The parameter units have been added at Line 137, line 150 and in Tables 1 and 2 as follows:

The model has only two parameters (Table 1), $C$ and $SC$. The parameter $C$ takes account of the effect of the change of time scale when simulating actual evapotranspiration. The parameter $SC$ represents the field capacity **(mm)**. **(Pages 8, Lines 164~166)**

Table 1 Parameters of the TMWB model

| Parameter | Physical meaning | Range and units |
|-----------|------------------|-----------------|
| C | Evapotranspiration parameter | 0.2-2.0 (–) |
| SC | Catchment water storage capacity | 100-2000 (mm) |

The meaning, range and units of all the parameters in the Xinanjiang model are listed in Table 2. **(Pages 9, Lines 176~177)**

Table 2 Parameters of the Xinanjiang model

| Category | Parameter | Physical meaning | Range and units |
|----------|-----------|------------------|-----------------|
| | WM | Tension water capacity | 80-400 (mm) |
| | X | WUM=X×WM, WUM is the tension water capacity of lower layer | 0.01-0.8 (–) |
| Evapotran spiration | Y | WLM=Y×WM, WLM is the tension water capacity of deeper layer | 0.01-0.8 (–) |
| | K | Ratio of potential evapotranspiration to pan evaporation | 0.4-1.5 (–) |

| | | | |
|---|---|---|---|
| | C | The coefficient of deep evapotranspiration | 0.01-0.4 (–) |
| Runoff production | B | The exponent of the tension water capacity curve | 0.1-10 (–) |
| | IMP | The ratio of the impervious to the total area of the basin | 0.01-0.15 (–) |
| Runoff separation | SM | The areal mean of the free water capacity of the surface soil layer | 10-80 (mm) |
| | EX | The exponent of the free water capacity curve | 0.6-6 (–) |
| | CG | The outflow coefficients of the free water storage to groundwater | 0.01-0.45 (–) |
| | CI | The outflow coefficients of the free water storage to interflow | 0.01-0.45 (–) |
| Flow concentration | N | Number of reservoirs in the instantaneous unit hydrograph | 0.5-10 (–) |
| | NK | Common storage coefficient in the instantaneous unit hydrograph | 1-20 (–) |
| | KG | The recession constant of groundwater storage | 0.6-1 (–) |
| | KI | The recession constant of the lower interflow storage | 0.9-1 (–) |

(7) Section 2.2: the need to reduce the number of Xinanjian free parameters using a sensitivity analysis must be investigated more deeply in the paper. In the current version of the paper, this model is considered with different number of free parameters depending on the modeling experiments. Why not calibrating the 15 free parameters of this rainfall-runoff model for all experiments?

**Reply:**

Here we add a synthetic experiment with the Xinanjiang model, where the true values of KE, CI, CG, KI, KG, and NK have periodic variations with changes every month (720h) and those of the insensitive parameters remain temporally constant. The 1-SSC-DP is applied to this experiment and all 15 free parameters are calibrated without a sensitivity analysis. The estimated parameters are plotted in Figure R1.

[Figure]

Figure R1 The parameters estimated without a sensitivity analysis

From the figure, it can be seen that except the estimated KE, CI, CG, KI, KG, and NK, the estimations of the insensitive parameters, such as WM, X and Y, are also recognized to vary significantly during the calibration period. This is inconsistent with the true values in the synthetic experiment, and the attribution analysis between time-varying parameters and watershed characteristics will be mistaken in practical use, which also occurs in the data assimilation method. Hence a sensitivity analysis is needed to find which parameters are really important for calibration.

This point has been highlighted in the Revised Manuscript as follows:

A sensitivity analysis is employed to focus efforts on parameters important to calibration and avoid prohibitive computational cost, as outlined in Sect. 2.2. **(Pages 7, Lines 139-141)**

(8) Section 2.3.1: one of the main hypotheses of this paper is the important "fluctuations" of the model parameter values over adjacent sub-periods, hypothesis that is not justified by the literature review, and that is not illustrated with the obtained results. This point must be discussed more deeply in the paper.

**Reply:**

Thanks for the comment. The main hypothesis of parameter continuity is justified as follows:
1. The hypothesis of parameter continuity can be found in the model prediction process of the ensemble Kalman filter (EnKF). Therein, the values of the parameters at the time step $t$+1 are forecasted by perturbing those of parameters from the time step $t$. The equation is as follows:

$$\vartheta_{t+1}^{k-}=\vartheta_t^{k+}+\delta_t^k, \ \delta_t^k \sim N\left(0,R_t\right) \tag{1}$$

where $\vartheta_{t+1}^{k-}$ is the forecasted parameter vector at the time step $t$+1,while $\vartheta_t^{k+}$ is the well-calibrated parameter vector. $\delta_t^k$ is the white noise following a Gaussian distribution with zero mean and specified covariance of $R_t$, which is very small. That is, the fluctuations between parameters of adjacent sub-periods can be little.
2. Some conceptual hydrological parameters reflect the catchment characteristics, such as soil water storage capacity in the Xinanjiang model. While climate change and human activities exert influence on catchment characteristics, the soil water storage capacity can hardly change dramatically in a very quick time, such as an hour.
Hence, it is reasonable to consider parameter continuity in estimating time-varying parameters. This point has been added in the Revised Manuscript as follows:

Some conceptual hydrological parameters reflect the catchment characteristics. While climate change and human activities exert influence on these catchment

characteristics, they can hardly change dramatically in a very quick time, such the soil water storage capacity. **(Pages 5, Lines 89-92)**

(9) Evaluation criteria: Why only use the NSE criterion as only evaluation criteria, and no other criteria, such as KGE and its components? NSE appears to be nondiscriminating between considered calibration methods. Using other calibration criteria- looking at different time step and/or different error characteristics such as bias on the highest streamflow values – might be interesting in this context.

**Reply:**

As well as NSE coefficient, two evaluation metrics have been added in Revised Manuscript: relative error (RE) and the NSE on logarithm of streamflow ($NSE_{ln}$).

In the revised paper, these evaluation metrics are described as follows:

The streamflow simulations given by the proposed method are verified using the NSE, relative error (RE) and NSE on logarithm of streamflow ($NSE_{ln}$) (Hock, 1999). RE evaluates the error of the total volume of streamflow, while NSE and $NSE_{ln}$ evaluate the agreement between the hydrograph of observations and simulations. NSE is more sensitive to high flows, but $NSE_{ln}$ focuses more on low flows. Higher values of NSE, $NSE_{ln}$ and lower values of RE indicate better streamflow simulations. The NSE, RE and $NSE_{ln}$ are expressed as followed: **(Pages 15~16, Lines 324-333)**

$$NSE = 1 - \frac{\sum_{t=1}^{m}(Q_t - \hat{Q}_t)^2}{\sum_{t=1}^{m}(Q_t - \overline{Q}_t)^2} \tag{15}$$

$$RE = \frac{\sum_{t=1}^{m}(Q_t - \hat{Q}_t)}{\sum_{t=1}^{m}Q_t} \tag{16}$$

$$NSE_{ln} = 1 - \frac{\sum_{t=1}^{m}(\ln(Q_t) - \ln(\hat{Q}_t))^2}{\sum_{t=1}^{m}(\ln(Q_t) - \ln(\overline{Q}_t))^2} \tag{17}$$

A description of the evaluation results has been added as follows:

➤ For results of the synthetic experiment with the TMWB model

Figure 6(a) presents the runoff simulation performance for various scenarios. In scenario 1, the NSE values of the three SSC-DP methods are all higher than that of EnKF. The results of $NSE_{ln}$ show no significant differences among various methods.

For scenarios 2, 4, and 6, where true parameters have linear trends, the 6-SSC-DP and 12-SSC-DP are superior to the EnKF and 3-SSC-DP in terms of NSE and $NSE_{ln}$. In scenario3, where the true parameters have periodic variations and change every month, the NSE and $NSE_{ln}$ values of 6-SSC-DP and 12-SSC-DP decrease significantly, because the assumed sub-period length is longer than the time-scale of actual variations. Similarly, in scenario 5, 12-SSC-DP performs worst for NSE and $NSE_{ln}$, but 6-SSC-DP performs best. In scenario 7 and 8, both 6-SSC-DP and 12-SSC-DP perform better than EnKF. According to the evaluations of NSE and $NSE_{ln}$, the SSC-DP offers improved accuracy than the EnKF if the proper length is chosen. Another advantage of the SSC-DP is the low RE. For all scenarios, the SSC-DP methods significantly outperform for RE compared with EnKF. Among the SSC-DP methods, the RE of 3-SSC-DP is the smallest. **(Page 22, Lines 492~506)**

(a) Simulation performance for streamflow

(b) Estimation performance for parameter C

(c) Estimation performance for parameter SC

Figure 6 Comparison between the EnKF and SSC-DP methods for (a) streamflow simulation and identification of (b) parameter C and (c) parameter SC.

➤ For results of the synthetic experiment with the Xinanjiang model

The simulated streamflow and identification of time-varying parameters was compared across four methods: 1-SSC, SSC-EnKF, 1-SSC-DP, and 2-SSC-DP. The simulation performance is summarized in Figure 9(a). For all scenarios, the NSE of 2-SSC-DP is the lowest, but it performs better for low flows. The SSC-EnKF produces the highest RE in scenarios 2, 3 and 4, indicating the problem of simulating water balance. The SSC and 1-SSC-DP perform well for all scenarios in terms of NSE, RE and $NSE_{ln}$. Wherein, the SSC performs better than the 1-SSC-DP with regard to RE, while 1-SSC-DP is slightly superior to SSC in scenario 3 with higher $NSE_{ln}$. **(Page 24, Lines 560~566)**

[Figure]

Figure 9 Comparison among the SSC, SSC-EnKF and SSC-DP methods for (a)

streamflow simulation and parameter identification in terms of (b) RMSE, (c) MARE and (d) $R^2$.

➤ For results of case study in Wuding River basin

The simulation performance is presented in Figure 12. The values of the NSEs are relatively low, it is because the streamflow in dry regions is difficult to simulate. It can be seen that the 12-SSC-DP gives the best simulation results among different methods with the highest NSE, $NSE_{ln}$ and low RE. Although the 12-SSC produces relatively high NSE, but it performs worst simulations for low flows. The SSC-EnKF has relative high $NSE_{ln}$, but the RE of it is the largest. Overall, the 12-SSC-DP significantly improve the simulation performance of the Xinanjiang model in the Wuding River basin. **(Page 26, Lines 706~713)**

[Figure]

Figure 12 Simulation performance for streamflow in the Wuding River basin.

➤ For results of case study in Xun River basin

The simulation performance is presented in Figure 15. All methods performed well, with NSE values of 92.5 %, 93.0 %, 95.0 %, and 94.8 % for the conventional method, 3-SSC-EnKF, 3-SSC, and 3-SSC-DP, respectively. 3-SSC and 3-SSC-DP also perform well for $NSE_{ln}$ compared with 3-SSC-EnKF and the conventional method. However, as regards to RE, the values are 0.0007 and 0.0324 for 3-SSC-DP and 3-SSC-DP, respectively. It indicated that the 3-SSC-DP can better simulate water balance than the 3-SSC in the Xun River basin. **(Page 28~29, Lines 785~806)**

[Figure]

Figure 15 Simulation performance for streamflow in the Xun River basin.

(10) Section 3.2 (Wuding river basin), lines 364 to 369: the changes of the studied catchment characteristics seem to be decisive for the interpretation of the results obtained on this watershed. Nevertheless, no quantitative results / analysis of these changes are given in the paper: what is the percentage of the catchment that has been afforested? What are the number and the capacity of the built reservoirs? When are they built? Finally, an important point not discussed in the paper is the stationary and then quality of the precipitation and streamflow time series studied and used for the model calibration. This point is crucial in this context and need to be discussed.

**Reply:**

This comment involves two aspects:

(1) For the first aspect, a quantitative analysis of the changes in the Wuding river basin has been added in the Revised Manuscript, including the areas of tree planning and check dams for soil and water conservations. This point is described in the Revised Manuscript as follows:

[Figure]

Figure 5(e) Temporal variations in the soil and water conservation measures.

Soil and water conservation measures, such as construction of the check dams and afforestation, have been undertaken since the 1960s. The areas of two soil and water conservation measures are plotted in Fig. 5(e), the data of which were collected from Zhang et al. (2002). The areas of tree planning have an increasing trend, but the slope gets much larger after 1972. It indicates that the greater efforts have been made for afforestation since the turning point. Similarly, the areas of dammed lands also increase, but the rate gets slower after 1972. These two soil and water conservation measures had changed underlying surface of the watershed, and impacted the relationship between precipitation and runoff (Gao et al., 2017; Jiao et al., 2017). **(Pages 20, Lines 441-449)**

(2) The reviewer concerns the quality of the precipitation and streamflow used. The data of the daily precipitation and streamflow in the Wuding River basin are obtained from the local Hydrology and Water Resources Bureau of China, the quality of which has been checked by the official authorities, and there are no gaps among these data for all the hydrological stations. This point has been clarified in the Revised Manuscript. **(Page 20, Lines 433~436)**

(11) Section 3.3 (Xun River basin): same remarks as the Wuding river basin: what about potential changes on this basin? Are precipitation and streamflow time series of good quality?

**Reply:**

This comment involves two aspects:

(1) The seasonal variations of the mean monthly precipitation, pan evaporation and streamflow are shown in Fig. 5(d). It shows that Xun River basin exhibits strong seasonal patterns in these climatic and hydrological variables. This point is added in the Revised Manuscript as follows:

It can be observed from Fig. 5(d) that no trend is found in annual precipitation, pan evaporation and streamflow, suggesting that the relationship between precipitation and runoff of the Xun River basin is rarely affected by human activities during 1991-2001. However, there exhibits strong seasonal patterns in these three climatic and hydrological variables, suggesting that seasonal variations in hydrological parameters should be considered. **(Pages 21, Lines 472-477)**

[Figure]

Figure 5 Location of (a) Wuding River basin and (b) Xun River basin. The plots (c) and (d) show the average yearly and monthly variations of precipitation, pan evaporation and streamflow in the Wuding River basin and Xun River basin, respectively.

  (2) The data of the precipitation and streamflow in the Xun River basin are also obtained from the local Hydrology and Water Resources Bureau of China, the quality of which has been checked by the official authorities, and there are no gaps among these data for all the hydrological stations. This point has been clarified in the Revised Manuscript. **(Pages 21, Lines 469-471)**

(12) Section 4.1: the seasonal signal of the parameter values (cf. figures 6 and 8) must be more significantly discussed in the paper.

**Reply:**

More discussion concerning the seasonal signal of the parameter values is added in the Revised Manuscript as follows:

(1) For the synthetic experiment with the TMWB model

    When the synthetic true parameters vary sinusoidally from month to month, EnKF gives the best estimations in scenario 3. The poor performances of 6-SSC-DP and 12-SSC-DP can be explained by the sub-period length being much longer than the actual one. When the parameters vary periodically at six-month intervals (scenario 5), 6-SSC-DP yields the best performance with the lowest RMSE, MARE and highest R2. The differences of estimation performances among 3-SSC-DP, 12-SSC-DP and EnKF are

small. The estimated parameters for scenario 5 have been plotted in Fig. 7(a). Although 3-SSC-DP and 12-SSC-DP have different lengths of sub-periods, they can also detect the correct seasonal signal of the parameters. For the annual variation in parameters (scenario 7), 12-SSC-DP and 6-SSC-DP produce better results than EnKF. Similar results can be seen in scenario 8 where C has a combined variation from year to year. In summary, the results indicate that the SSC-DP with a suitable length can estimate more accurate parameters than EnKF. **(Pages 23~24, Lines 534-546)**

(2) For the synthetic experiment with the Xinanjiang model

When the synthetic true parameters vary sinusoidally from month to month (scenario 3), the estimated parameters are plotted in Fig. 10. It can be seen that 1-SSC-DP successfully detects seasonal signal in every parameter. The SSC-EnKF performs well for R2, but it has high MARE. Although the average MARE of the SSC and 2-SSC-DP are lower than that of SSC-EnKF, the R2 of them are relatively low. Therein, form Fig. 10, the estimated parameters by the 1-SSC fluctuate generally periodically, but the variations are dramatic, resulting in lowest R2 for CI, KI, KG and NK. The estimated parameters of the 2-SSC-DP fluctuate more slowly, but the sub-period length is too long. In scenario 4, 1-SSC performs better than the SSC-EnKF and 2-SSC-DP, but is still slightly inferior to the 1-SSC-DP. Overall, the 1-SSC-DP achieves higher-quality and more robust parameter estimations performances than the other methods. **(Pages 25, Lines 656-666)**

(13) Line 456 to 458: this conclusion must be significantly moderated: the "SSC-DP" calibration method is by definition better to select more continuous parameter values.

**Reply:**

This conclusion has been moderated in the Revised Manuscript as follows:

Overall, the 1-SSC-DP achieves higher-quality and more robust parameter estimation performances than the other methods. **(Page 25, Lines 665~666)**

(14) Section 4.2: this data analysis is crucial in this context. It might be relevant to present it in the data section. Moreover, this analysis must be significantly improved: what about potential errors (random or systematic) in the observed precipitation and streamflow series? What about potential break in the streamflow series due to rating curve changes? What is the statistical significance of this analysis? The analysis of only one catchment requires to look carefully the studied time series in the context of attribution of changes. The relative bad performance of the rainfall-runoff model on this catchment (NSE=0.41) must be discussed. In particular, the systematic streamflow underestimation for the different calibration methods must be discussed.

**Reply:**

This comment involves four aspects:

(1) For the first aspect, the hydrological data are collected from the local Hydrology and Water Resources Bureau of China, the systematic errors of which have been

checked by the official authorities. Additionally, random errors are considered in the synthetic experiment, the results show that 5% random errors have little influence on the SSC-DP.

(2) The reviewer concerns that the potential break in the streamflow series due to rating curve changes. The streamflow data used have been checked to guarantee their continuity, that is, no break (the discharge equal to zero) has been found except on two discontinuous days. Since the daily streamflow is also very low near the break, the values of the streamflow are reasonable.

(3) It is found that all the analyses of the linear regression are significant. The statistical significance of the analysis has been added in the Revised Manuscript and Fig. 11 as follows:

[Figure]

Fig. 11 Double mass curves between daily runoff and precipitation for (a) Wuding River basin from 1958–1972; (b) Wuding River basin from 1973–2000; (c) Xun River basin from 1991–2001. Subgraph (d) represents the double mass curve between the mean daily runoff and precipitation from 1991–2001.

The two linear slopes ($p$-value < 0.05) of the curves are different before and after 1972, demonstrating the relationship between precipitation and runoff changes under the soil and water conservation measures. **(Page 25, Lines 669~672)**

(4) The reviewer also concerns the bad simulation performance in the Wuding River basin. It is because the streamflow in dry regions is difficult to simulate. The main reason is the deficiencies of the model structure. This point is added in the Revised Manuscript. **(Page 26, Lines 706~708; Pages 26~27, Lines 720-735)**

(15) Line 492 to 495: the "unreasonable model states" between sub-periods might be illustrated in the paper.

**Reply:**

The statement about "the unreasonable model states" is an incorrect description and has been deleted in the Revised Manuscript. The parameters over each sub-period are calibrated separately using the SSC method. Several sets of parameters can lead to similar simulation performance in each sub-period, i.e., parameter equifinality. This equifinality causes uncertainty in simulating fluxes and streamflow.

Here, the time-series of the estimated groundwater discharge have been plotted in Fig. R2.

[Figure]

Fig. R2 The estimated groundwater discharge of the Wuding River basin

From the Fig. R2, the estimated groundwater discharge by the SSC fluctuates dramatically on December 27, 1977, which seems unreasonable, while the estimations by the SSC-DP have no dramatically fluctuations. Hence, the SSC-DP outperforms the SSC for the Wuding River basin.

(16) Line 500 to 501: this conclusion must be moderated, since results have been obtained on one catchment only.

**Reply:**

This sentence has been moderated in the Revised Manuscript:
    It can be inferred the 12-SSC-DP is more applicable to the simulation of streamflow in the Wuding River basin. **(Pages 27, Lines 739-740)**

(17) Line 511 to 514: this attribution analysis must be moderated (see previous remarks on attribution analysis).

**Reply:**

The attribution analysis has been moderated in the Revised Manuscript:

    The results show that WM remains constant before and after 1972, but WUM

varies significantly over this period, indicating that the distribution of soil water capacity may change, i.e., WUM decreases but WLM increases. A Person correlation analysis is applied to investigate the relationship between the areas of tree planning and WUM as well as WLM. It is found that there is a significant negative correlation (Pearson correlation efficient $\rho$=-0.38, P<0.05) between the areas of tree planning and WUM. While WLM has a nonsignificant positive correlation ($\rho$=0.26, P>0.05) with the areas of tree planning. It can be inferred that less severe soil erosion occurred, because the upper layers became thinner while the lower layer, where vegetation roots dominate, became thicker (Jayawardena and Zhou, 2000). Additionally, IMP is significantly correlated with the areas of tree planning ($\rho$=-0.33, P<0.05). Except for afforestation, the areas of the dammed lands are significantly correlated with WLM ($\rho$=0.46, P<0.05), suggesting that the construction of the check dams also has influence on the soil water capacity of the Wuding river basin. Other parameters, KE, KI, KG, N and NK have little differences before and after 1972. The variations in WLM and IMP slowed down after the turning point, similar to the results of Deng et al. (2016). **(Pages 27~28, Lines 741-770)**

(18) Line 520 to 522: again, what about potential error in the rating curve in this context?

**Reply:**

The potential error in the rating curve is considered in this study from two aspects:
(1) The streamflow data are managed by the local Hydrology and Water Resources Bureau. There is a strict specification for hydrometry for drawing the rating curve. Hence, the streamflow accuracy is guaranteed.
(2) In the synthetic experiment, the uncertainty of observations has been considered, and the results show that 5% random errors have little influence on the SSC-DP.

(19) Line 526 to 539: why not presenting a Figure such as Figure 10 to illustrate rainfall-runoff simulations on this catchment?

**Reply:**

Thanks for the reminder. The Figure and the description have been added in the Revised Manuscript as follows:
    Figure 16 illustrates the hydrograph and quantile-quantile plots for the simulations in the Xun river basin. It is evident that the peak flows estimated by the 3-SSC is higher than those of 3-SSC-DP, and 3-SSC-DP simulate better the flows ranging from 100 m3/s to 200 m3/s. **(Pages 29, Lines 806-809)**

[Figure]

Figure 16 (a) Streamflow simulation hydrograph (left panels) and quantile-quantile plots (right panels) using conventional method, SSC-EnKF, SSC, and SSC-DP for the Xun River basin. (b) The quantile-quantile plot for all streamflow; (c) The quantile-quantile plot for streamflow ranging from 100 m³/s to 200 m³/s.

(20) Line 531 to 532: is this out-performance significant?

**Reply:**

To give a more comprehensive evaluation, two metrics, relative error (RE) and the NSE on logarithm of streamflow (NSE$_{ln}$), are added. This sentence on line 531 to 532 has been modified as follows:

As regards to RE, the values are 0.0007 and 0.0324 for 3-SSC-DP and 3-SSC-DP, respectively. It indicated that the 3-SSC-DP can better simulate water balance than the 3-SSC in the Xun River basin. **(Pages 28~29, Lines 789-806)**

(21) Line 534 to 539 and line 637 to 648: again, these attribution conclusions must be moderated, because they are drawn from only two basins, without any investigation of potential systematic errors in the observed time series.

**Reply:**

The statement is an incorrect description and has been deleted in the Revised Manuscript. Here, the estimations of groundwater discharge are plotted in the Fig. R3. It can be seen that the estimations are similar for SSC and SSC-DP, which is different from that in the Wuding case study. It suggests that the SSC-DP gives more robust simulation performance for both case studies.

[Figure]

Fig. R3 The estimated groundwater discharge of the Xun River basin

Merz, R., Parajka, J. and Bloeschl, G. (2011) Time stability of catchment model parameters: Implications for climate impact analyses. Water resources research 47(W02531).

Fowler, K., Knoben, W.J.M., Peel, M.C., Peterson, T.J., Ryu, D., Saft, M., Seo, K.-W. and Western, A. (2020) Many Commonly Used Rainfall-Runoff Models Lack Long, Slow Dynamics: Implications for Runoff Projections. Water resources research 56(5).

Stephens, C.M., Marshall, L.A. and Johnson, F.M. (2019) Investigating strategies to improve hydrologic model performance in a changing climate. Journal of Hydrology 579.

Deng, C., Liu, P., Wang, W., Shao, Q. and Wang, D. (2019) Modelling time-variant parameters of a two-parameter monthly water balance model. Journal of Hydrology 573, 918-936.

---

## Editor Comment (EC1) · Dimitri Solomatine (Editor) · 28 Sep 2020

Referees provided comprehensive comments, and gave a numbr of very useful suggestions. From the authors' comments it is clear that they appreciate these and have a plan for the paper revision. I would ask the authors to better format their replies, clearly separating comments from replies, and to use more paragraphs, since it is quite difficult to read. Success!

---

## Author Comment (AC3) · 7 Nov 2020

**Responses to Reviewer #1**

The reviewer's comments have been considered carefully, and the manuscript has been extensively revised. The following tables summarize the reviewer's comments. For convenience, we have classified them into categories. Comments regarding similar issues and their locations are summarized together. Simplified notation has been used to assist the reviewer in conveniently locating a specific comment: R-Reviewers, and C-Comment. For example, R1-C1 denotes comment 1 made by Reviewer #1. This simplified form is used in all of the following tables.

**Category I. The abstract and introduction**

|   | Comment | Reviewer-Location | |
|---|---------|-------------------|---|
| 1 | Refining the highlight | R1-C1 | |

**Category II. The methodology**

|   | Comment | Reviewer-Location |
|---|---------|-------------------|
| 1 | Justification of using two hydrological models | R1-C5 |
| 2 | Adding a flowchart of the study | R1-C6 |
| 3 | Adding the evaluation metrics | R1-C7 |

**Category III. The case study and results**

|   | Comment | Reviewer-Location | |
|---|---------|-------------------|---|
| 1 | Adding analysis and description of data used in case study | R1-C8 | case study |
| 2 | Figure modification | R1-C10 | results |
| 3 | Modification for the results | R1-C7 | |

**Category IV. The discussion and conclusions**

|   | Comment | Reviewer-Location | |
|---|---------|-------------------|---|
| 1 | Determination of the length of sub-periods | R1-C4 | discussion |

| 2 | Refining the highlight | R1-C1 | conclusion |

**Category V. Others**

|   | Comment | Reviewer-Location |
|---|---------|-------------------|
| 1 | Identifiability of parameters | R1-C2 |
| 2 | Catchment characteristics unable to be predicted | R1-C3 |
| 3 | Justification of the parameter continuity assumption | R1-C9 |

**Responses to Reviewer #1:**

1. This paper presents methods to estimate the time-varying parameter based on dynamic programming. The authors attempt to combine multiple methods including SSC and ENKF. However, the highlight of this paper is no very clear, which should be refined.

**Reply:**

Thank you for reviewing our manuscript and for the professional comments. The highlights of this paper are refined as follows:

1. The proposed method combines split-sample calibration (SSC) and ensemble Kalman filter (EnKF) for time-varying parameter estimation. Compared to SSC, the proposed method can find a more continuous parameter trajectory; compared to EnKF, the proposed method allows parameters to retain stable for a pre-determined period, instead of varying at every time-step.

2. The effectiveness of the proposed method is validated with two hydrological models and two real catchment case studies of different conditions.

3. For the case study of the Xun River basin, the proposed method detects the strongest seasonal signal.

The highlights are elaborated on in the abstract as follows:

Although the parameters of hydrological models are usually regarded as constant, temporal variations can occur in a changing environment. Thus, effectively estimating time-varying parameters becomes a significant challenge. **Two methods, including split-sample calibration (SSC) and Data assimilation, have been used to estimate time-varying parameters. However, SSC is unable to consider the parameter temporal continuity, while Data assimilation assumes parameters vary at every time-step.** This study proposed a new method that combines (1) the basic concept of SSC, whereby parameters are assumed to be stable for one sub-period, and (2) the parameter continuity assumption, i.e., the differences between parameters in consecutive time steps are small. **(Pages 2, Lines 3-7)**

The highlights are also elaborated in the conclusions as follows:

**1. The proposed method with a suitable length not only produces better simulation performance, but also ensures more accurate parameter estimates than SSC and EnKF in the synthetic experiment using the TMWB model with two parameters.** The impact of sub-period lengths on the performance of SSC-DP is significant when the known parameters vary sinusoidally.

**2. The proposed method can be used to deal with complex hydrological models involving a large number of parameters**, demonstrated by the synthetic experiment using the Xinanjiang model with 15 parameters. A sensitivity analysis was performed to reduce the probable computational cost and improve the efficiency of identifying the time-varying parameters.

3. **The proposed method has the potential to detect the relationship between the time-varying parameters and dynamic catchment characteristics.** For example, SSC-DP produced the best simulation performance in the case study of the Wuding River basin and detects that parameters representing soil water capacity and impervious areas changed significantly after 1972, reflecting the soil and water conservation projects carried out from 1958–2000. Additionally, SSC-DP detects the strongest seasonal signal in the case study of Xun River basin, indicating the distinct impacts of seasonal climate variability. **(Page 33 Line 656~673)**

2. The fundamental assumption that the individual parameters may not response to the catchment dynamics due to the linear or nonlinear correlations between parameters (Bardossy, 2007). The effects of identifiability of parameters on this research are suggested to be investigated.

**Reply:**

We agree with the reviewer that the hydrological model parameters should be treated as parameter vectors instead of independent individual values (Bardossy 2007). The identifiability of parameters is considered in this study:

(1) Parameters are not treated as individuals, but multiple parameters are identified simultaneously. For the two-parameter monthly water balance (TMWB) model, parameters C and SC are estimated simultaneously. While for the Xinanjiang model, the sensitive parameters are calibrated at the same time.

(2) By generating a large number of parameter sets as candidates in each sub-period, the proposed method takes into account the parameter equifinality, while the traditional SSC method only takes the optimal parameter set.

3. The non-stationary change in catchment characteristics may not be predicted. Lots of uncertainty factors would prevent the estimation of future scenarios in catchments.

**Reply:**

This study focuses on methods to identify time-varying parameters, and the future research is considered to relate time-varying parameters and available information, such as the number of dams and population. Then the time-varying parameters' function can be derived to predict future streamflow under the changing environment.

4. How to generally estimate the stable period, such as decades, years or months, considering catchment characteristics? It is vital for the method in this study. The impact of sub-period lengths on the performance of SSC-DP is significant.

**Reply:**

Determination of the stable period considers 3 factors:

1. Temporal scale of climate change or human activities. The Wudinghe River basin is taken as a case study. Since 1960s, the soil and water conservation measures were carried out in this basin to reduce the highly erodible loess, such as tree plantation, reservoir construction and land terracing. The human activities lead to a durative and long-term change in the catchment characteristic. Hence, the yearly sub-period is considered.

2. Seasonality. The Xun River basin is taken as a case study. Contrary to the Wudinghe River basin, the relationship between precipitation and runoff of the Xun River basin is rarely affected by human activities during 1991-2001. However, its significant seasonal dynamics can be observed and has been studied in literature (Lan et al. 2020, Lan et al. 2018). In order to diagnose the seasonality, the stable period of 3-month is considered.

3. The simulation accuracy. The length should not be too long to capture the variations in physical processes, while it should be long enough to reduce the uncertainty of calibration. Based on the results of the synthetic experiments, it is suggested that the length should be as long as possible without degrading the simulation performance significantly. For example, in the synthetic experiment with the TMWB model, if the difference between the NSE values of 6-SSC-DP and 3-SSC-DP is small, the preferred length is six months.

The determination of the sub-period length has been described in discussion as follows:

It is suggested that the determination of the sub-period length considers three factors:

(1) The temporal scale of climate change or human activities. For example, the Wudinghe River basin is taken as a case study. The soil and water conservation measures have led to a durative and long-term change in the catchment characteristics since the 1960s. Due to this, the yearly sub-period is preferred.

(2) The seasonality. Contrary to the Wudinghe River basin, the relationship between precipitation and runoff of the Xun River basin is rarely affected by human activities during 1991-2001. However, its significant seasonal dynamics can be observed and has been studied in literature (Lan et al. 2020, Lan et al. 2018). In order to diagnose the seasonality, the stable period of 3-month is adopted.

(3) The simulation accuracy. The length should be neither too long nor too short so as to increase the reliability of the calibration while guaranteeing that variations in real processes are captured. Thus, given that the time scale of the variations is unknown, the proposed SSC-DP can be used with different split-sample lengths. It is suggested that the length should be as long as possible without degrading the simulation performance significantly. For example, in the synthetic experiment with the TMWB model, if the difference between the NSE values of 6-SSC-DP and 3-SSC-DP is small, the preferred length is 6-month. **(Page 31~32 Line 623~641)**

5. The two lumped models were chosen in this study. The number of parameters is different. The sensitivity analysis was further performed to reduce the dimension of parameters in the Xinanjiang model. Hence, the purpose of choosing two different lumped models should be discussed.

**Reply:**

Two lumped models are chosen to evaluate the applicability of the proposed method to hydrological models with different number of parameters. Furthermore, the parameters of the TMWB model have been identified by EnKF in the work of Deng et al. (2016), but the parameters of the Xinanjiang model are scarcely recognized as time-variant. Hence, the use of the TMWB model is beneficial for comparison.

The purpose of choosing two different lumped models has been added as follows:

There are two important differences between the TMWB and Xinanjiang models: (1) the TMWB model has two parameters, while the Xinanjiang model has fifteen parameters; (2) TMWB is a monthly rainfall-runoff model, whereas the Xinanjiang model can run on hourly or daily step sizes. **(Page 9 Line 157~158)**

6. The titles cannot show the logic framework of the research. The flowchart is suggested to be used to illustrate the framework in this study. The introduction of the manuscript is suggested to present in the appendix.

**Reply:**

To avoid confusion, the title of Section 3, i.e., "Data and study area", is replaced by "Synthetic experiment and real catchment case study".

A flowchart describing the framework of the research is added as follows:

[Figure]

**Modeling:**
Hydrological models and
sensitivity analysis
(Section 2.1&2.2)

**Calibration:**
Time-varying parameter
estimation method
(Section 2.3)

**Comparison:**
Model evaluation criteria
(Section 2.4)

Figure 1 The flowchart of the methodologies

The introduction of the methodologies is presented as follows:

**In this section, a SSC-DP method is proposed to identify the time-varying parameters of hydrological models.** The two hydrological models considered in this study are the TMWB and Xinanjiang models. Their concepts and differences are presented in Sect. 2.1. A sensitivity analysis is employed to focus efforts on parameters important to calibration and avoid prohibitive computational cost, as outlined in Sect. 2.2. **Three time-varying parameter estimation methods (SSC, SSC-DP, and data assimilation) are presented in Sect. 2.3. The SSC and data assimilation are provided for comparisons with the SSC-DP.** Finally, to evaluate the performance of the time-varying parameter estimation methods, six evaluation criteria are selected and formulated in Sect. 2.4. **The flowchart of the methodologies is shown in Fig. 1. (Pages 7, Lines 123-132)**

7. The sensitive hydrograph phases of model performance criteria, i.e., RMSE, $R^2$ and NSE are peaks and discharge dynamics, flood peak, and discharge dynamics (Pfannerstill et al., 2014). Three metrics have strong correlations. The results as shown in Figure 5 needs furthermore discussion.

**Reply:**

Thanks for the comment. This comment involves three aspects:

**(1) Three metrics are used to evaluate the streamflow simulations.**

NSE coefficient, and two evaluation metrics have been added: relative error (RE) and the NSE on logarithm of streamflow ($NSE_{ln}$).

In the revised paper, these evaluation metrics are described as follows:

[revised manuscript text omitted]

Description of the evaluation results has been added in Revised Manuscript as follows:

➢ For results of the synthetic experiment with the TMWB model

Figures 6 (b) and (c) focuses on the ability of the four methods to identify time-varying parameters. It can be seen that the RMSE and MARE values of the 3-SSC-DP are larger than those of other methods in most cases. That is because the sub-period length that serves as a calibration period for MCMC is too short (i.e., three months) that the estimated parameters are associated with higher uncertainties.

Regarding the synthetic true parameters are constant values (scenario 1), 12-SSC-DP gives the best performance with the lowest RMSE, MARE and highest $R^2$. The observations and estimated parameters are presented in Figure 7 (b). It shows that the estimated parameters obtained by EnKF vary at every time step, resulting in larger deviations from the observations than 6-SSC-DP and 12-SSC-DP.

When the synthetic true parameters vary linearly (scenarios 2, 4, and 6), 12-SSC-DP produces best estimations in comparison with EnKF, 3-SSC-DP, and 6-SSC-DP. The performances of 6-SSC-DP and EnKF are similar.

When the synthetic true parameters vary sinusoidally from month to month, EnKF gives the best estimations in scenario 3. The poor performances of 6-SSC-DP and 12-SSC-DP can be explained by the sub-period length being much longer than the actual one. When the parameters vary periodically at six-month intervals (scenario 5), 6-SSC-DP yields the best performance with the lowest RMSE, MARE and highest $R^2$. The differences of estimation performances among 3-SSC-DP, 12-SSC-DP and EnKF are small. The estimated parameters for scenario 5 have been plotted in Fig. 7(a). Although 3-SSC-DP and 12-SSC-DP have different lengths of sub-periods, they can also detect the correct seasonal signal of the parameters. For the annual variation in parameters (scenario 7), 12-SSC-DP and 6-SSC-DP produce better results than EnKF. Similar results can be seen in scenario 8 where C has a combined variation from year to year. In summary, the results indicate that the SSC-DP with a suitable length can estimate more accurate parameters than EnKF. **(Page 22~24 Line 434~459)**

➢ For results of the synthetic experiment with the Xinanjiang model

Figures 9(b) and (c) compare the time-varying parameter estimation performance among the four methods. In scenarios 1 and 2, 2-SSC-DP produces the lowest RMSE, MARE and R2, followed by the 1-SSC-DP. The 1-SSC-DP is slightly superior to the 1-SSC and significantly outperforms the SSC-EnKF for the two scenarios.

When the synthetic true parameters vary sinusoidally from month to month (scenario 3), the estimated parameters are plotted in Fig. 10. It can be seen that 1-SSC-DP successfully detects seasonal signal in every parameter. The SSC-EnKF performs well for R2, but it has high MARE. Although the average MARE of the SSC and 2-SSC-DP are lower than that of SSC-EnKF, the $R^2$ of them are relatively low. Therein, from Fig. 10, the estimated parameters by the 1-SSC fluctuate generally periodically, but the variations are dramatic, resulting in lowest $R^2$ for CI, KI, KG and NK. The estimated parameters of the 2-SSC-DP fluctuate more slowly, but the sub-period length is too long. In scenario 4, 1-SSC performs better than the SSC-EnKF and 2-SSC-DP, but is still slightly inferior to the 1-SSC-DP. Overall, the 1-SSC-DP achieves higher-quality and more robust parameter estimations performances than the other methods. **(Page 25 Line 480~494)**

**(3) The figure 5 is replaced by Figure 6 in the Revised Manuscript.** The results as shown in Figure 6 have been presented in the reply (2) of R1-C7.

8. The streamflow, climate and underlying surface conditions in the two study areas were not analyzed in this study. However, it is critical to the estimation of time-varying parameters.

**Reply:**

Figure 5 has been modified as follows:

[Figure]

Figure 5 Location of (a) Wuding River basin and (b) Xun River basin. The plots (c) and (d) show the average yearly and monthly variations of precipitation, pan evaporation and streamflow in the Wuding River basin and Xun River basin, respectively. The plot (e) shows the temporal variations in the soil and water conservation measures.

The details of the Wuding River basin have been added as follows:

As illustrated in Fig. 5(a), the station furthest downstream, Baijiachuan, drains an area of 29,662 km$^2$ (98 % of the total basin) and records the daily runoff data. The data of the daily precipitation and streamflow in the Wuding River basin were obtained from the local Hydrology and Water Resources Bureau of China, the quality of which has been checked by the official authorities, and there are no gaps among these data for all the hydrological stations. It can be seen from Fig. 5(c) that the annual streamflow in the Wudinghe River basin has a distinct decreasing trend, while seasonal variations are not significant, but the annual precipitation and pan evaporation generally have no trend, suggesting the impacts of human activities on rainfall–runoff relationships. **(Page 20 Line 374~381)**

The details of the Xun River basin have been added as follows:

It can be observed from Fig. 5(d) that no trend is found in annual precipitation, pan evaporation and streamflow, suggesting that the relationship between precipitation and runoff of the Xun River basin is rarely affected by human activities during 1991-2001. However, there exhibit strong seasonal patterns in these three climatic and hydrological variables, suggesting that seasonal variations in hydrological parameters should be considered. **(Page 21 Line 404~409)**

9. In lines 175-176, the assumption that the continuity condition aims to minimize the difference between the estimated parameters for sub-periods i and i+1 unreasonable. The differences between two consecutive sub-periods represent the time-varying changes of the catchment. The continuity conditions for enhancing the model performance should focus on the model structure, such as state variables.

**Reply:**

Thanks for the comment. The main hypothesis of parameter continuity is justified as follows:

1. The hypothesis of parameter continuity can be found in the model prediction process of the ensemble Kalman filter (EnKF). Therein, the values of the parameters at the time step $t$+1 are forecasted by perturbing those of parameters from the time step $t$. The equation is as follows:

$$\vartheta_{t+1}^{k-} = \vartheta_t^{k+} + \delta_t^k, \ \delta_t^k \sim N\left(0, R_t\right) \tag{1}$$

where $\vartheta_{t+1}^{k-}$ is the forecasted parameter vector at the time step $t$+1,while $\vartheta_t^{k+}$ is the well-calibrated parameter vector. $\delta_t^k$ is the white noise following a Gaussian distribution with zero mean and specified covariance of $R_t$ which is very small. That is, the fluctuations between parameters of adjacent sub-periods can be little.

2. Some conceptual hydrological parameters reflect the catchment characteristics, such as soil water storage capacity in the Xinanjiang model. While climate change and human activities exert influence on catchment characteristics, the soil water storage capacity can hardly change dramatically in a very quick time, such as an hour.

Hence, it is reasonable to consider parameter continuity in estimating time-varying parameters.

This point has been added in the Revised Manuscript as follows:

Some conceptual hydrological parameters reflect the catchment characteristics. While climate change and human activities exert influence on these catchment characteristics, they can hardly change dramatically in a very quick time, such as the soil water storage capacity. **(Page 5 Line 79~82)**

10. Minor comment. The resolution of Figure 5 is low and information is not presented.
**Reply:**

The Figure 5 is replaced by Figure 6 in the Revised Manuscript to be easier to read as follows:

[Figure]

Figure 6 Comparison between the EnKF and SSC-DP methods for (a) streamflow simulation and identification of (b) parameter C and (c) parameter SC.

Bardossy, A. (2007) Calibration of hydrological model parameters for ungauged catchments. Hydrology and Earth System Sciences 11(2), 703-710.

Lan, T., Lin, K., Xu, C.-Y., Tan, X. and Chen, X. (2020) Dynamics of hydrological-model parameters: mechanisms, problems and solutions. Hydrology and Earth System Sciences 24(3), 1347-1366.

Lan, T., Lin, K.R., Liu, Z.Y., He, Y.H., Xu, C.Y., Zhang, H.B. and Chen, X.H. (2018) A Clustering Preprocessing Framework for the Subannual Calibration of a Hydrological Model Considering Climate-Land Surface Variations. Water resources research 54(0).

Deng, C., Liu, P., Guo, S., Li, Z. and Wang, D. (2016) Identification of hydrological model parameter variation using ensemble Kalman filter. Hydrology and Earth System Sciences 20(12), 4949-4961.

**Responses to Reviewer #2**

The reviewer's comments have been considered carefully, and the manuscript has been extensively revised. The following tables summarize the reviewer's comments. For convenience, we have classified them into categories. Comments regarding similar issues and their locations are summarized together. Simplified notation has been used to assist the reviewer in conveniently locating a specific comment: R-Reviewers, and C-Comment. For example, R2-C1 denotes comment 1 made by Reviewer #2. This simplified form is used in all of the following tables.

**Category I. The methodology**

| | Comment | Reviewer-Location |
|---|---|---|
| 1 | Justification of using conceptual hydrological models | R2-C2 |
| 2 | Adding units for Xinanjiang model parameters | R2-C6 |
| 3 | Justification of using the sensitivity analysis | R2-C7 |
| 4 | Adding the evaluation metrics | R2-C9 |

**Category II. The case study and results**

| | Comment | Reviewer-Location |
|---|---|---|
| 1 | Quantified analysis of the soil and water conservation measures on the Wuding River basin | R2-C10 |
| 2 | Quantified analysis of the seasonal variations on the Xun River basin | R2-C11 |
| 3 | Quality of the data used | R2-C10; R2-C11; R2-C14; R2-C18 |
| 4 | Figure modification | R2-C1; |
| 5 | Replacing tables with figures | R2-C1 |
| 6 | Modification for the results | R2-C12; R2-C20 |
| 7 | Statistical significance of this analysis | R2-C14 |
| 8 | Explanation for the relative bad performance on the Wuding River basin | R2-C14 |
| 9 | Modification of the attribution analysis | R2-C17 |
| 10 | Adding a hydrograph plot for the Xun River basin | R2-C19 |

**Category III. Others**

| | Comment | Reviewer-Location |
|---|---|---|
| 1 | Justification of the parameter continuity assumption | R2-C8 |
| 2 | Explanation and modification of words and phrasing | R2-C3; R2-C4; R2-C5; R2-C13; R2-C15; R2-C16; R2-C21 |

**Responses to Reviewer #2:**

1. OVERALL RECOMMENDATION

The manuscript addresses the important topic of calibration of rainfall-runoff model parameters, and presents results obtained on two different catchments with two models. Even if the introduction includes relevant references and the methods are well presented, the paper lacks important discussions on the rainfall-runoff model performances, observed time series quality, attribution of observed/simulated changes, consideration of only two catchments, and several obtained results are over-interpreted. Finally, several figures and tables must be significantly improved. Therefore, I think the manuscript requires major revision before publication.

**Reply:**

Thank you for reviewing our manuscript and for the professional comments, which are carefully followed in making revisions.

2. GENERAL COMMENTS

(1) The tables 6 to 8 might be presented as figures to be more easily interpreted. Figure 5, 7 and 10 are very difficult to read, and must be significantly improved.

**Reply:**

(1) Table 6 and Figure 5 have been modified and replaced by Figure 6 in Revised Manuscript as follows:

(a) Simulation performance for streamflow

[Figure]

(b) Estimation performance for parameter C

(c) Estimation performance for parameter SC

□ ENKF   ○ 3-SSC-DP   △ 6-SSC-DP   ▽ 12-SSC-DP

Figure 6 Comparison between the EnKF and SSC-DP methods for (a) streamflow simulation and identification of (b) parameter C and (c) parameter SC.

(2) Table 7 has been presented as Figure 8 in Revised Manuscript as follows:

[Figure]

Figure 8 Comparison among EnKF, SSC-EnKF, and EnKS in the synthetic experiment with the Xinanjiang model (3) Table 8 and Figure 7 have been modified and replaced by Figure 9 in Revised Manuscript as follows:

[Figure]

(a) Simulation performance for streamflow (b) Parameter estimation performance in terms of RMSE

(c) Parameter estimation performance in terms of MARE

Average MARE for the 6 parameters (d) Parameter estimation performance in terms of R²

Average R² for the 6 parameters

Figure 9 Comparison among the SSC, SSC-EnKF and SSC-DP methods for (a)

streamflow simulation and parameter identification in terms of (b) RMSE, (c) MARE and (d) $R^2$.

(4) Figure 10 is modified and replaced by Figure 13 in Revised Manuscript as follows:

[Figure]

Figure 13 The simulated and observed streamflow using the conventional method, SSC-EnKF, SSC, and SSC-DP for the Wuding River basin. (a) Streamflow simulation hydrograph; (b) The quantile-quantile plot for all streamflow; (c) The quantile-quantile plot for streamflow lower than 100 m³/s.

(2) Line 28 to 29: several studies highlighted the difficulty of conceptual rainfall-runoff models in the context of climate change impact studies.

**Reply:**

Thanks. We agree that conceptual rainfall-runoff models can be difficult to simulate the variations in discharge in response to climate changes in some cases (Merz et al. 2011, Fowler et al. 2020). That is, the simulation accuracy reduces when the conceptual model is applied in situations where the climatic conditions, e.g., dry periods, are not consistent with that of the calibration period, e.g., wet periods. Some literatures have made improvements to enhance parameter transferability between various climatic conditions. One approach is to allow the parameters of the conceptual model to change (Stephens et al. 2019, Deng et al. 2019), which can efficiently improve the accuracy of the conceptual model and simulate the response of runoff in a changing environment.

(3) Line 35 to 36: the terms "constants" and "stable" must be defined: constant/stable in space and/or in time?

**Reply:**

The terms are defined as "constant in time scale" and "temporally stable". The statement at line 35 to 36 will be modified in the Revised Manuscript: Parameters are usually regarded as constants in time scale, because of the general idea that catchment conditions are temporally stable.

(4) Line 43 to 44: in this context, it may be needed to define what is called "climate conditions".

**Reply:**

Here, the "climate conditions" means "wet/dry periods". To avoid confusion, the sentence at line 42 to 44 of the Revised Manuscript is modified by replacing "climate conditions" with "wet/dry periods": Fowler et al. (2016) pointed out that the parameter set obtained by mathematical optimization based on wet periods may not be robust when applied in dry periods.

(5) Line 122: the terms "behavioural" must be clearly defined or not used in this context.

**Reply:**

The "behavioural" means "important to calibration metrics and predictions". To avoid confusion, the "behavioural" is replaced by "sensitive" in the line 122 of the Revised Manuscript.

(6) Line 137, line 150 and Table 1 and 2: please presents parameter units.

**Reply:**

Thanks for reminder. The parameter units have been added on Line 137, line 150 and in Tables 1 and 2 as follows:

The parameter *SC* represents the field capacity **(mm)**. **(Pages 8, Lines 143~144)**

Table 1 Parameters of the TMWB model

| Parameter | Physical meaning | Range and units |
| --- | --- | --- |
| C | Evapotranspiration parameter | 0.2-2.0 (–) |
| SC | Catchment water storage capacity | 100-2000 (mm) |

The meaning, range and units of all the parameters in the Xinanjiang model are listed in Table 2. **(Page 9 Line 154~155)**

Table 2 Parameters of the Xinanjiang model

| Category | Parameter | Physical meaning | Range and units |
| --- | --- | --- | --- |
| Evapotranspiration | WM | Tension water capacity | 80-400 (mm) |
| | X | WUM=X×WM, WUM is the tension water capacity of lower layer | 0.01-0.8 (–) |
| | Y | WLM=Y×WM, WLM is the tension water capacity of deeper layer | 0.01-0.8 (–) |
| | K | Ratio of potential evapotranspiration to pan evaporation | 0.4-1.5 (–) |

| | | | |
|---|---|---|---|
| | C | The coefficient of deep evapotranspiration | 0.01-0.4 (–) |
| Runoff production | B | The exponent of the tension water capacity curve | 0.1-10 (–) |
| | IMP | The ratio of the impervious to the total area of the basin | 0.01-0.15 (–) |
| Runoff separation | SM | The areal mean of the free water capacity of the surface soil layer | 10-80 (mm) |
| | EX | The exponent of the free water capacity curve | 0.6-6 (–) |
| | CG | The outflow coefficients of the free water storage to groundwater | 0.01-0.45 (–) |
| | CI | The outflow coefficients of the free water storage to interflow | 0.01-0.45 (–) |
| Flow concentration | N | Number of reservoirs in the instantaneous unit hydrograph | 0.5-10 (–) |
| | NK | Common storage coefficient in the instantaneous unit hydrograph | 1-20 (–) |
| | KG | The recession constant of groundwater storage | 0.6-1 (–) |
| | KI | The recession constant of the lower interflow storage | 0.9-1 (–) |

(7) Section 2.2: the need to reduce the number of Xinanjian free parameters using a sensitivity analysis must be investigated more deeply in the paper. In the current version of the paper, this model is considered with different number of free parameters depending on the modeling experiments. Why not calibrating the 15 free parameters of this rainfall-runoff model for all experiments?

**Reply:**

Here we add a synthetic experiment with the Xinanjiang model, where the true values of KE, CI, CG, KI, KG, and NK have periodic variations with changes every month (720h) and those of the insensitive parameters remain temporally constant. The 1-SSC-DP is applied to this experiment and all 15 free parameters are calibrated without a sensitivity analysis. The estimated parameters are plotted in Figure R1.

[Figure]

Figure R1 The parameters estimated without a sensitivity analysis

From the figure, it can be seen that except the estimated KE, CI, CG, KI, KG, and NK, the estimations of the insensitive parameters, such as WM, X and Y, are also recognized to vary significantly during the calibration period. This is inconsistent with the true values in the synthetic experiment, and the attribution analysis between time-varying parameters and watershed characteristics will be mistaken in practical use, which also occurs in the data assimilation method. Hence a sensitivity analysis is needed to find which parameters are really important for calibration.

This point has been highlighted in the Revised Manuscript as follows:

A sensitivity analysis is employed to focus efforts on parameters important to calibration and avoid prohibitive computational cost, as outlined in Sect. 2.2. **(Page 7 Line 126~127)**

(8) Section 2.3.1: one of the main hypotheses of this paper is the important "fluctuations" of the model parameter values over adjacent sub-periods, hypothesis that is not justified by the literature review, and that is not illustrated with the obtained results. This point must be discussed more deeply in the paper.

**Reply:**

Thanks for the comment. The main hypothesis of parameter continuity is justified as follows:
1. The hypothesis of parameter continuity can be found in the model prediction process of the ensemble Kalman filter (EnKF). Therein, the values of the parameters at the time step $t+1$ are forecasted by perturbing those of parameters from the time step $t$. The equation is as follows:

$$\vartheta_{t+1}^{k-} = \vartheta_t^{k+} + \delta_t^k, \ \delta_t^k \sim N\left(0, R_t\right) \tag{1}$$

where $\vartheta_{t+1}^{k-}$ is the forecasted parameter vector at the time step $t+1$, while $\vartheta_t^{k+}$ is the well-calibrated parameter vector. $\delta_t^k$ is the white noise following a Gaussian distribution with zero mean and specified covariance of $R_t$ which is very small. That is, the fluctuations between parameters of adjacent sub-periods can be little.

2. Some conceptual hydrological parameters reflect the catchment characteristics, such as soil water storage capacity in the Xinanjiang model. While climate change and human activities exert influence on catchment characteristics, the soil water storage capacity can hardly change dramatically in a very quick time, such as an hour.

Hence, it is reasonable to consider parameter continuity in estimating time-varying parameters. This point has been added in the Revised Manuscript as follows:

Some conceptual hydrological parameters reflect the catchment characteristics. While climate change and human activities exert influence on these catchment characteristics, they can hardly change dramatically in a very quick time, such as the soil water storage capacity. **(Page 5 Line 79~82)**

(9) Evaluation criteria: Why only use the NSE criterion as only evaluation criteria, and no other criteria, such as KGE and its components? NSE appears to be nondiscriminating between considered calibration methods. Using other calibration criteria- looking at different time step and/or different error characteristics such as bias on the highest streamflow values – might be interesting in this context.

**Reply:**

As well as NSE coefficient, two evaluation metrics have been added in Revised Manuscript: relative error (RE) and the NSE on logarithm of streamflow (NSE$_{ln}$).

In the revised paper, these evaluation metrics are described as follows:

The streamflow simulations given by the proposed method are verified using the NSE, relative error (RE) and NSE on logarithm of streamflow (NSE$_{ln}$) (Hock, 1999). RE evaluates the error of the total volume of streamflow, while NSE and NSE$_{ln}$ evaluate the agreement between the hydrograph of observations and simulations. NSE is more sensitive to high flows, but NSE$_{ln}$ focuses more on low flows. Higher values of NSE, NSE$_{ln}$ and lower absolute values of RE indicate better streamflow simulations. The NSE, RE and NSE$_{ln}$ are expressed as followed: **(Pages 15~16, Lines 292-301)**

$$NSE = 1 - \frac{\sum_{t=1}^{m}(Q_t - \widehat{Q}_t)^2}{\sum_{t=1}^{m}(Q_t - \overline{Q}_t)^2} \tag{15}$$

$$RE = \frac{\sum_{t=1}^{m}(Q_t - \widehat{Q}_t)}{\sum_{t=1}^{m}Q_t} \tag{16}$$

[revised manuscript text omitted]

Figure 15 Simulation performance for streamflow in the Xun River basin.

(10) Section 3.2 (Wuding river basin), lines 364 to 369: the changes of the studied catchment characteristics seem to be decisive for the interpretation of the results obtained on this watershed. Nevertheless, no quantitative results / analysis of these changes are given in the paper: what is the percentage of the catchment that has been afforested? What are the number and the capacity of the built reservoirs? When are they built? Finally, an important point not discussed in the paper is the stationary and then quality of the precipitation and streamflow time series studied and used for the model calibration. This point is crucial in this context and need to be discussed.

**Reply:**

This comment involves two aspects:

(1) For the first aspect, a quantitative analysis of the changes in the Wuding river basin has been added in the Revised Manuscript, including the areas of tree planning and check dams for soil and water conservations. This point is described in the Revised Manuscript as follows:

[Figure]

Figure 5(e) Temporal variations in the soil and water conservation measures.

Soil and water conservation measures, such as construction of the check dams and afforestation, have been undertaken since the 1960s. The areas of two soil and water conservation measures are plotted in Fig. 5(e), the data of which were collected from Zhang et al. (2002). The areas of tree planning have an increasing trend, but the slope gets much larger after 1972. It indicates that the greater efforts have been made for afforestation since the turning point. Similarly, the areas of dammed lands also increase, but the rate gets slower after 1972. These two soil and water conservation measures had changed the underlying surface of the watershed, and impacted the relationship between precipitation and runoff (Gao et al., 2017; Jiao et al., 2017). **(Page 20 Line 382~390)**

(2) The reviewer concerns the quality of the precipitation and streamflow used. The data of the daily precipitation and streamflow in the Wuding River basin are obtained from the local Hydrology and Water Resources Bureau of China, the quality of which has been checked by the official authorities, and there are no gaps among these data for all the hydrological stations. This point has been clarified in the Revised Manuscript. **(Page 20 Line 374~381)**

(11) Section 3.3 (Xun River basin): same remarks as the Wuding river basin: what about potential changes on this basin? Are precipitation and streamflow time series of good quality?

**Reply:**

This comment involves two aspects:

(1) The seasonal variations of the mean monthly precipitation, pan evaporation and streamflow are shown in Fig. 5(d). It shows that Xun River basin exhibits strong seasonal patterns in these climatic and hydrological variables. This point is added in the Revised Manuscript as follows:

It can be observed from Fig. 5(d) that no trend is found in annual precipitation, pan evaporation and streamflow, suggesting that the relationship between precipitation and runoff of the Xun River basin is rarely affected by human activities during 1991-2001. However, there exhibit strong seasonal patterns in these three climatic and hydrological variables, suggesting that seasonal variations in hydrological parameters should be considered. **(Page 21 Line 404~409)**

[Figure]

Figure 5 Location of (a) Wuding River basin and (b) Xun River basin. The plots (c) and (d) show the average yearly and monthly variations of precipitation, pan evaporation and streamflow in the Wuding River basin and Xun River basin, respectively.

(2) The data of the precipitation and streamflow in the Xun River basin are also obtained from the local Hydrology and Water Resources Bureau of China, the quality of which has been checked by the official authorities, and there are no gaps among these data for all the hydrological stations. This point has been clarified in the Revised Manuscript. **(Page 21 Line 401~403)**

(12) Section 4.1: the seasonal signal of the parameter values (cf. figures 6 and 8) must be more significantly discussed in the paper.

**Reply:**

More discussion concerning the seasonal signal of the parameter values is added in the Revised Manuscript as follows:

(1) For the synthetic experiment with the TMWB model

When the synthetic true parameters vary sinusoidally from month to month, EnKF gives the best estimations in scenario 3. The poor performances of 6-SSC-DP and 12-SSC-DP can be explained by the sub-period length being much longer than the actual one. When the parameters vary periodically at six-month intervals (scenario 5), 6-SSC-DP yields the best performance with the lowest RMSE, MARE and highest R2. The differences of estimation performances among 3-SSC-DP, 12-SSC-DP and EnKF are small. The estimated parameters for scenario 5 have been plotted in Fig. 7(a). Although 3-SSC-DP and 12-SSC-DP have different lengths of sub-periods, they can also detect the correct seasonal signal of the parameters. For the annual variation in parameters (scenario 7), 12-SSC-DP and 6-SSC-DP produce better results than EnKF. Similar results can be seen in scenario 8 where C has a combined variation from year to year. In summary, the results indicate that the SSC-DP with a suitable length can estimate more accurate parameters than EnKF. **(Pages 23~24, Lines 447-459)**

(2) For the synthetic experiment with the Xinanjiang model

When the synthetic true parameters vary sinusoidally from month to month (scenario 3), the estimated parameters are plotted in Fig. 10. It can be seen that 1-SSC-DP successfully detects seasonal signal in every parameter. The SSC-EnKF performs well for R2, but it has high MARE. Although the average MARE of the SSC and 2-SSC-DP are lower than that of SSC-EnKF, the R2 of them are relatively low. Therein, from Fig. 10, the estimated parameters by the 1-SSC fluctuate generally periodically, but the variations are dramatic, resulting in lowest R2 for CI, KI, KG and NK. The estimated parameters of the 2-SSC-DP fluctuate more slowly, but the sub-period length is too long. In scenario 4, 1-SSC performs better than the SSC-EnKF and 2-SSC-DP, but is still slightly inferior to the 1-SSC-DP. Overall, the 1-SSC-DP achieves higher-quality and more robust parameter estimations performances than the other methods. **(Pages 25, Lines 484-494)**

(13) Line 456 to 458: this conclusion must be significantly moderated: the "SSC-DP" calibration method is by definition better to select more continuous parameter values.

**Reply:**

This conclusion has been moderated in the Revised Manuscript as follows:
Overall, the 1-SSC-DP achieves higher-quality and more robust parameter estimation performances than the other methods. **(Page 25 Line 480~494)**

(14) Section 4.2: this data analysis is crucial in this context. It might be relevant to present it in the data section. Moreover, this analysis must be significantly improved: what about potential errors (random or systematic) in the observed precipitation and streamflow series? What about potential break in the streamflow series due to rating curve changes? What is the statistical significance of this analysis? The analysis of only one catchment requires to look carefully the studied time series in the context of attribution of changes. The relative bad performance of the rainfall-runoff model on this catchment (NSE=0.41) must be discussed. In particular, the systematic streamflow underestimation for the different calibration methods must be discussed.

**Reply:**

This comment involves four aspects:

(1) For the first aspect, the hydrological data are collected from the local Hydrology and Water Resources Bureau of China, the systematic errors of which have been checked by the official authorities. Additionally, random errors are considered in the synthetic experiment. The results show that 5% random errors have little influence on the SSC-DP.

(2) The reviewer concerns about the potential break in the streamflow series due to rating curve changes. The streamflow data used have been checked to guarantee their continuity, that is, no break (the discharge equal to zero) has been found except on two discontinuous days. Since the daily streamflow is also very low near the break, the values of the streamflow are reasonable.

(3) It is found that all the analyses of the linear regression are significant. The statistical significance of the analysis has been added in the Revised Manuscript and Fig. 11 as follows:

[Figure]

Fig. 11 Double mass curves between daily runoff and precipitation for (a) Wuding River basin from 1958–1972; (b) Wuding River basin from 1973–2000; (c) Xun River basin from 1991–2001. Subgraph (d) represents the double mass curve between the mean daily runoff and precipitation from 1991–2001.

The two linear slopes ($p$-value < 0.05) of the curves are different before and after 1972, demonstrating the relationship between precipitation and runoff changes under the soil and water conservation measures. **(Page 25, Lines 497~500)**

(4) The reviewer also concerns the bad simulation performance in the Wuding River basin. It is because the streamflow in dry regions is difficult to simulate. The main reason is the deficiencies of the model structure. This point is added to the Revised Manuscript. **(Page 26, Lines 508~510; Pages 26~27, Lines 522-523)**

(15) Line 492 to 495: the "unreasonable model states" between sub-periods might be illustrated in the paper.

**Reply:**

The statement about "the unreasonable model states" is an incorrect description and has been deleted in the Revised Manuscript. The parameters over each sub-period are calibrated separately using the SSC method. Several sets of parameters can lead to similar simulation performance in each sub-period, i.e., parameter equifinality. This equifinality causes uncertainty in simulating fluxes and streamflow.

Here, the time-series of the estimated groundwater discharge have been plotted in Fig. R2.

[Figure]

Fig. R2 The estimated groundwater discharge of the Wuding River basin

From the Fig. R2, the estimated groundwater discharge by the SSC fluctuates dramatically on December 27, 1977, which seems unreasonable, while the estimations by the SSC-DP have no dramatically fluctuations. Hence, the SSC-DP outperforms the SSC for the Wuding River basin.

(16) Line 500 to 501: this conclusion must be moderated, since results have been obtained on one catchment only.

**Reply:**

This sentence has been moderated in the Revised Manuscript:

It can be inferred that the 12-SSC-DP is more applicable to the simulation of streamflow in the Wuding River basin. **(Page 27 Line 527~528)**

(17) Line 511 to 514: this attribution analysis must be moderated (see previous remarks on attribution analysis).

**Reply:**

The attribution analysis has been moderated in the Revised Manuscript:

The results show that *WM* remains constant before and after 1972, but *WUM* varies significantly over this period, indicating that the distribution of soil water capacity may change, i.e., *WUM* decreases but *WLM* increases. A Person correlation analysis is applied to investigate the relationship between the areas of tree planning and *WUM* as well as *WLM*. It is found that there is a significant negative correlation (Pearson correlation efficient $\rho=-0.38$, P<0.05) between the areas of tree planning and *WUM*. While *WLM* has a nonsignificant positive correlation ($\rho=0.26$, P>0.05) with the areas of tree planning. It can be inferred that less severe soil erosion occurred, because the upper layers became thinner while the lower layer, where vegetation roots dominate, became thicker (Jayawardena and Zhou, 2000). Additionally, *IMP* is significantly correlated with the areas of tree planning ($\rho=-0.33$, P<0.05). Except for afforestation, the areas of the dammed lands are significantly correlated with *WLM* ($\rho=0.46$, P<0.05), suggesting that the construction of the check dams also has influence on the soil water capacity of the Wuding river basin. Other parameters, *KE*, *KI*, *KG*, *N* and *NK* have little differences before and after 1972. The variations in *WLM* and *IMP* slowed down after the turning point, similar to the results of Deng et al. (2016). **(Pages 27~28, Lines 530-545)**

(18) Line 520 to 522: again, what about potential error in the rating curve in this context?

**Reply:**

The potential error in the rating curve is considered in this study from two aspects:
(1) The streamflow data are managed by the local Hydrology and Water Resources Bureau. There is a strict specification for hydrometry for drawing the rating curve. Hence, the streamflow accuracy is guaranteed.
(2) In the synthetic experiment, the uncertainty of observations has been considered, and the results show that 5% random errors have little influence on the SSC-DP.

(19) Line 526 to 539: why not presenting a Figure such as Figure 10 to illustrate rainfall-runoff simulations on this catchment?

**Reply:**

Thanks for the reminder. The Figure and the description have been added in the Revised Manuscript as follows:
Figure 16 illustrates the hydrograph and quantile-quantile plots for the simulations in the Xun river basin. It is evident that the peak flows estimated by the 3-SSC are higher than those of 3-SSC-DP, and 3-SSC-DP simulate better the flows ranging from 100 $m^3$/s to 200 $m^3$/s. **(Page 28~29 Line 560~570)**

[Figure]

Figure 16 The simulated and observed streamflow using the conventional method, SSC-EnKF, SSC, and SSC-DP for the Xun River basin. (a) Streamflow simulation hydrograph; (b) The quantile-quantile plot for all streamflow; (c) The quantile-quantile plot for streamflow ranging from 100 m³/s to 200 m³/s.

(20) Line 531 to 532: is this out-performance significant?

**Reply:**

To give a more comprehensive evaluation, two metrics, relative error (RE) and the NSE on logarithm of streamflow ($NSE_{ln}$), are added. This sentence on line 531 to 532 has been modified as follows:

   As regards to RE, the values are 0.0007 and 0.0324 for 3-SSC-DP and 3-SSC-DP, respectively. It indicated that the 3-SSC-DP can better simulate water balance than the 3-SSC in the Xun River basin. **(Pages 28~29, Lines 564-567)**

(21) Line 534 to 539 and line 637 to 648: again, these attribution conclusions must be moderated, because they are drawn from only two basins, without any investigation of potential systematic errors in the observed time series.

**Reply:**

The statement is an incorrect description and has been deleted in the Revised Manuscript. Here, the estimations of groundwater discharge are plotted in the Fig. R3. It can be seen that the estimations are similar for SSC and SSC-DP, which is different from that in the Wuding case study. It suggests that the SSC-DP gives more robust simulation performance for both case studies.

[Figure]

Fig. R3 The estimated groundwater discharge of the Xun River basin

Merz, R., Parajka, J. and Bloeschl, G. (2011) Time stability of catchment model parameters: Implications for climate impact analyses. Water resources research 47(W02531).

Fowler, K., Knoben, W.J.M., Peel, M.C., Peterson, T.J., Ryu, D., Saft, M., Seo, K.-W. and Western, A. (2020) Many Commonly Used Rainfall-Runoff Models Lack Long, Slow Dynamics: Implications for Runoff Projections. Water resources research 56(5).

Stephens, C.M., Marshall, L.A. and Johnson, F.M. (2019) Investigating strategies to improve hydrologic model performance in a changing climate. Journal of Hydrology 579.

Deng, C., Liu, P., Wang, W., Shao, Q. and Wang, D. (2019) Modelling time-variant parameters of a two-parameter monthly water balance model. Journal of Hydrology 573, 918-936.

**A list of changes made to original manuscript in detail**

1) Page 2 Line 3~7, the statements of "Following a survey of existing estimation methodologies, this paper describes a new method that combines" were modified as "Two methods, including split-sample calibration (SSC) and Data assimilation, have been used to estimate time-varying parameters … This study proposed a new method that combines".

2) Page 3 Line 31, "(Deng et al., 2019; Stephens et al., 2019)" were added.

3) Page 3 Line 37~38, "constants" and "stable" were modified as "constants in time scale" and "temporally stable", respectively.

4) Page 3 Line 44~46, "one climate condition" was modified as "wet periods"; "different conditions" was modified as "dry periods".

5) Page 5 Line 79~82, "because changes in the watershed characteristics occur over a prolonged period" is modified as "Some conceptual hydrological parameters reflect the catchment characteristics … they can hardly change dramatically in a very quick time, such the soil water storage capacity".

6) Page 7 Line 123~124, "In this section, a SSC-DP method is proposed to identify the time-varying parameters of hydrological models" was added.

7) Page 7 Line 126~127, "To avoid the prohibitive computational cost of the Xinanjiang model's calibration procedure, sensitivity analysis is employed to select behavioral parameters with less uncertainty" was modified as "A sensitivity analysis is employed to focus efforts on parameters important to calibration and avoid prohibitive computational cost".

8) Page 7 Line 128~130, "Three time-varying parameter estimation methods (SSC, SSC-DP, and data assimilation) are then used to determine the variations in these behavioral parameters, as described in Sect. 2.3." was modified as "Three time-varying parameter estimation methods (SSC, SSC-DP, and data assimilation) are presented in Sect. 2.3. The SSC and data assimilation are provided for comparisons with the SSC-DP".

9) Page 7 Line 132, "The flowchart of the methodologies is shown in Fig. 1" was added.

10) Page 8 Line 144, the unit of parameter SC was added.

11) Page 9 Line 154~155, "The 15 parameters in the Xinanjiang model are defined in Table 2" was modified as "The meaning, range and units of all the parameters in the Xinanjiang model are listed in Table 2".

12) Page 9 Line 157~158, "the TMWB model is much simpler and has fewer parameters than the Xinanjiang model" was modified as "the TMWB model has two parameters, while the Xinanjiang model has fifteen parameters".

13) Page 15~17 Line 292~313, " The streamflow simulations and parameter estimations given by the proposed time-varying parameter estimation approach are verified using the NSE…A Taylor diagram is used to summarize the standard deviation, RMSE, and R2 in a polar plot, providing a graphical representation of the performance of SSC-DP" was changed into "The streamflow simulations given by the proposed method are verified using the NSE, relative error (RE) and NSE on logarithm of streamflow (NSEln)…and $m$ is the length of the data during the whole period".

14) Page 17 Line 315, "Data and study area" was changed into "Synthetic experiment and real catchment case study".

15) Page 20 Line 374~381, "The data of the daily precipitation and streamflow in the Wuding River basin were obtained from the local Hydrology and Water Resources Bureau of China …… suggesting the impacts of human activities on rainfall–runoff relationships" was added.

16) Page 20 Line 382~390, "The erosion of loess, vegetable degradation, and human activities mean that the Wuding River basin suffers severe soil erosion…Several studies have reported the anthropogenic impacts of this area and demonstrated the changing relationship between precipitation and runoff" was changed into "Soil and water conservation measures, such as construction of the check dams and afforestation, have been undertaken since the 1960s…These two soil and water conservation measures had changed underlying surface of the watershed, and impacted the relationship between precipitation and runoff".

17) Page 21 Line 401~403, "The data in the Xun River basin were also obtained from the local Hydrology and Water Resources Bureau of China, and there are no gaps among these data for all the hydrological stations" was added.

18) Page 21 Line 404~409, "As a tributary of the Han River…Given that the majority of rainfall (approximately 70–80 % of the total) occurs in the summer, seasonal variations should also be considered" was changed into "It can be observed from Fig. 5(d) that no trend is found in annual precipitation, pan evaporation and streamflow… suggesting that seasonal variations in hydrological parameters should be considered".

19) Page 22 Line 419~433, "Table 6 presents the runoff simulation performance for various scenarios…SSC-DP offers improved accuracy if the proper length is chosen" was changed into "Figure 6(a) presents the runoff simulation performance for various scenarios…Among the SSC-DP methods, the RE of 3-SSC-DP is the smallest".

20) Page 22~24 Line 434~459, "Figure 5 focuses on the ability of the four methods to identify time-varying parameters…Thus, the estimated parameters are associated with higher uncertainties" was changed into "Figures 6 (b) and (c) focuses on the ability of the four methods to identify time-varying parameters…In summary, the results indicate that the SSC-DP with a suitable length can estimate more accurate parameters than EnKF".

21) Page 24 Line 468, "Table 7" was changed into "Fig.8".

22) Page 24 Line 473~479, "The simulation performance is summarized in Table 8…That is because, when there are more sub-periods (but the length of each sub-period is not too short), the performance tends to be better" was changed into "The simulation performance is summarized in Figure 9(a)…Wherein, the SSC performs better than the 1-SSC-DP with regard to RE, while 1-SSC-DP is slightly superior to SSC in scenario 3 with higher NSEln".

23) Page 25 Line 480~494, "Figure 7 compares the time-varying parameter estimation performance among the four methods…However, it performs worse than 1-SSC-DP and SSC-EnKF in scenarios 3 (period) and 4 (combination)" was changed into "Figures 9(b) and (c) compares the time-varying parameter estimation performance among the four methods.…Overall, the 1-SSC-DP achieves higher quality and more robust parameter estimations performances than the other methods".

24) Page 25 Line 496, "Figures 9(a) and (b)" was changed into "Figures 11(a) and (b)".

25) Page 25 Line 498, "(p-value < 0.05)" was added.

26) Page 26 Line 506~515, "The simulation performance of the conventional method is presented in Fig. 10(a) (NSE = 40.7 %)…It is evident that 12-SSC and 12-SSC-DP can significantly improve the simulation performance of the Xinanjiang model in this semi-arid region" was changed into "The simulation performance is presented in Figure 12…Overall, the 12-SSC-DP significantly improve the simulation performance of the Xinanjiang model in the Wuding River basin".

27) Page 26~27 Line 522~523, "The underestimation mainly derives from the deficiencies of the model structure" was added.

28) Page 27 Line 523, "Models" was changed into "Methods".

29) Page 27 Line 527~528, "It can be inferred the 12-SSC-DP is more applicable to the simulation of streamflow in semi-arid regions" was modified as "It can be inferred the 12-SSC-DP is more applicable to the simulation of streamflow in the Wuding River basin".

30) Page 27~28 Line 529~545, "The estimated time-varying parameters estimated by 12-SSC-DP are plotted in Fig. 11…The variations in WLM and IMP slowed down after the turning point, similar to the results of" was changed into "The estimated time-varying parameters estimated by 12-SSC-DP are plotted in Fig.14…The variations in WLM and IMP slowed down after the turning point, similar to the results of".

31) Page 28 Line 547, "Figures 9(c) and (d)" was changed into "Figures 11(c) and (d)".

32) Page 28~29 Line 560~570, "All methods performed well…The superior performance in the Wuding River basin suggests that SSC-DP is more useful when simulating streamflow in dry regions (or periods)" was changed into "The simulation performance is presented in Figure 15…3-SSC-DP simulate better the flows ranging from 100 m3/s to 200 m3/s".

33) Page 31~32 Line 623~641, "As reported in Sect. 4…the preferred length is six months" was changed into "It is suggested that the determination of the sub-period length considers three factors…if the difference between the NSE values of 6-SSC-DP and 3-SSC-DP is small, the preferred length is 6-month".

34) Page 33 Line 656~673, "One synthetic experiment used the TMWB model with two parameters and eight scenarios … between the temporal variations of parameters and the changing environment in real catchments." was changed into "The proposed method with a suitable length not only produces better simulation performance … indicating the distinct impacts of seasonal climate variability".

35) Page 39, the units were added to Table 1.

36) Page 40, the units were added to Table 2

37) Page 44, Figure 1 was added.

38) Page 48, Figure 5 was modified.

39) Page 49, Table 6 was deleted and Figure 6 was modified.

40) Page 51, Table 7 was deleted and Figure 8 was added.

41) Page 52, Table 8 was deleted and Figure 9 was modified.

42) Page 54, Figure 11 was modified.

43) Page 55, Figure 12 was added.

44) Page 56, Figure 13 was added.

45) Page 58, Figure 15 was added.

44) Page 59, Figure 16 was added.

[revised manuscript text omitted]

---

## Author Response (AR2)

**Responses to Editor and Reviewers**

**Editor:**

Referees are quite satisfied with the manuscript, however one of them still asks to address several minor points, which is quite easy to do. The revision will be reviewed by editor only.

**Reply:**

We thank the Editor and the reviewers very much for the evaluation and constructive comments/suggestions. The issues raised in the reviews have been carefully considered, and the manuscript has been revised accordingly. Please see the point-by-point responses to Reviewer 2 shown below.

**Reviewer #2:**

The authors adequately addressed my previous comments. I still have minor comments:

1. Line 76: the term "continuity" must be defined in this context. At this point of the paper, it is unclear why this "continuity" is needed for hydrological modeling.

**Reply:**

Thanks for the comment. The term "continuity" is defined as "differences between the parameters in consecutive time steps should be small".

Some conceptual hydrological parameters reflect the catchment characteristics such as the soil water storage capacity. They can hardly change dramatically in a very quick time under climate change and human activities. On this basis, the parameter continuity is reasonable for hydrological modeling.

To avoid confusion, this point is clarified in the Revised Manuscript:

Some conceptual hydrological parameters reflect the catchment characteristics. While climate change and human activities exert influence on these catchment characteristics, they can hardly change dramatically in a very quick time, such as the soil water storage capacity. **Hence, parameter continuity, defined as differences between the parameters in consecutive time steps to be small, is required for hydrological modeling.** However, few reports have considered the continuity of parameters in the SSC method. **(Page 5, Lines 76-82)**

2. Line 193: the terms "feasible parameters" and "nearly optimal" must be defined in this context.

**Reply:**

Here, the original "feasible parameters" means "near-optimal parameters". To avoid confusion, this term has been removed. Besides, the "nearly optimal" is corrected to

"near-optimal" which means that the parameter sets have objective values close to the optimum.

In the Revised Manuscript, the sentence is modified as follows:

**(2) Generate an ensemble of near-optimal parameters**. **Multiple parameter sets having objective values close to the optimum** for each sub-period are obtained using Markov chain Monte Carlo (MCMC) sampling (Chib and Greenberg, 1995). **(Page 10, Lines 194-196)**

3. Line 220: the term "good model performance" must be defined in this context.

**Reply:**

Here, the "good model performance" is defined as "accurate streamflow simulations". The sentence is moderated in the Revised Manuscript as follows:

(3) Optimize by using Dynamic programming. The goal is to find parameters that provide both **accurate streamflow simulations** and continuity. **(Page 11, Lines 202-203)**

4. Line 303: the term "accuracy of the estimated parameters" must be defined in this context.

**Reply:**

Thanks for the comment. The "accuracy of the estimated parameters" means "the overall agreement between the pre-determined parameters and their estimation in the synthetic experiments (see details in section 3.1)".

To avoid confusion, the sentence is moderated in the Revised Manuscript as follows:

The estimated parameters are evaluated by the RMSE (Alvisi et al., 2006), MARE (Khalil et al., 2001) and $R^2$ (Kim et al., 2007) **in the synthetic experiments (see details in section 3.1)**. RMSE is more sensitive to high values than MARE, while $R^2$ is based on the linear assumption (Dawson et al., 2007). **(Page 16, Lines 303-306)**

Dawson, C.W., Abrahart, R.J., See, L.M., 2007. Hydrotest: A web-based toolbox of evaluation metrics for the standardised assessment of hydrological forecasts. Environmental Modelling & Software 22(7), 1034-1052.

5. Line 305: what is the definition of "true" and "estimated" parameters in this context?

**Reply:**

The "true" and "estimated" parameters are defined as "the pre-determined parameters

and their estimations in the synthetic experiments (see details in section 3.1)".

In the Revised Manuscript, the sentence is moderated as follows:

The estimated parameters are evaluated by the RMSE (Alvisi et al., 2006), MARE (Khalil et al., 2001) and $R^2$ (Kim et al., 2007) in the synthetic experiments (see details in section 3.1). RMSE is more sensitive to high values than MARE, while $R^2$ is based on the linear assumption (Dawson et al., 2007). **(Page 16, Lines 303-306)**

6. Figure 5(e) and lines 382 to 390: these results ("temporal variations in the soil and water conservation measures") concern only the Wuding River basin and not the Xun River basin: it needs to be clearly stated in the figure caption.

**Reply:**

Thanks for the reminder. This point has been added in the caption of Figure 5(e) in the Revised Manuscript:

[Figure]

Figure 5 Location of (a) Wuding River basin and (b) Xun River basin. The plots (c) and (d) show the average yearly and monthly variations of precipitation, pan evaporation and streamflow in the Wuding River basin and Xun River basin, respectively. The plot (e) shows the temporal variations in the soil and water conservation measures **undertaken in the Wuding River basin**.

7. Figure 9: please state in the figure caption that these results concern the Xinanjiang model.

**Reply:**

In the Revised Manuscript, this point has been added in the caption of Figure 9:

[Figure]

Figure 9 Comparison among the SSC, SSC-EnKF and SSC-DP methods **in the synthetic experiment with the Xinanjiang model** for (a) streamflow simulation and parameter identification in terms of (b) RMSE, (c) MARE and (d) $R^2$.

8. Figures 12 and 15: please state in the text/in the figure caption how "RE" bars are constructed?

**Reply:**

Thanks for the comment. The description of RE is presented in the text as follows:

The streamflow simulations from the proposed method are verified by using the NSE, relative error (RE) and NSE on logarithm of streamflow (NSE$_{ln}$) (Hock, 1999). RE evaluates the error of the total volume of streamflow, while NSE and NSE$_{ln}$ evaluate the agreement between the hydrograph of observations and simulations. NSE is more sensitive to high flows, but NSE$_{ln}$ focuses more on low flows. Higher values of NSE, NSE$_{ln}$ and lower absolute values of RE indicate better streamflow simulations. The NSE, RE and NSE$_{ln}$ are expressed as followed:

$$NSE = 1 - \frac{\sum_{t=1}^{m}(Q_t - \widehat{Q}_t)^2}{\sum_{t=1}^{m}(Q_t - \overline{Q}_t)^2} \tag{15}$$

$$RE = \frac{\sum_{t=1}^{m}(Q_t - \widehat{Q}_t)}{\sum_{t=1}^{m}Q_t} \tag{16}$$

$$NSE_{ln} = 1 - \frac{\sum_{t=1}^{m}(\ln(Q_t) - \ln(\widehat{Q}_t))^2}{\sum_{t=1}^{m}(\ln(Q_t) - \ln(\overline{Q}_t))^2} \tag{17}$$

Besides, we agree with the reviewer that the "RE" bars are difficult to read in Figures 12 and 15. Thus, the "RE" bars are constructed after "NSE/NSEln" bars, and the details are added in the caption of Figures 12 and 15 in the Revised Manuscript as follows:

[Figure]

Figure 12 Simulation performance for streamflow in the Wuding River basin. **The results of NSE and NSEln are shown on the primary axis, while the values of RE are shown on the secondary axis.**

[Figure]

Figure 15 Simulation performance for streamflow in the Xun River basin. **The results of NSE and NSEln are shown on the primary axis, while the values of RE are shown on the secondary axis.**